# Sparse Uncertainty Representation in Deep Learning with Inducing Weights

**Hippolyt Ritter**[1]*, **Martin Kukla**[2], **Cheng Zhang**[2] **& Yingzhen Li**[3]*
[1]University College London [2]Microsoft Research Cambridge, UK [3]Imperial College London
j.ritter@cs.ucl.ac.uk, {Martin.Kukla,Cheng.Zhang}@microsoft.com,
yingzhen.li@imperial.ac.uk

## Abstract

Bayesian Neural Networks and deep ensembles represent two modern paradigms of uncertainty quantification in deep learning. Yet these approaches struggle to scale mainly due to memory inefficiency, requiring parameter storage several times that of their deterministic counterparts. To address this, we augment each weight matrix with a small inducing weight matrix, projecting the uncertainty quantification into a lower dimensional space. We further extend Matheron's conditional Gaussian sampling rule to enable fast weight sampling, which enables our inference method to maintain reasonable run-time as compared with ensembles. Importantly, our approach achieves competitive performance to the state-of-the-art in prediction and uncertainty estimation tasks with fully connected neural networks and ResNets, while reducing the parameter size to $\leq 24.3\%$ of that of a *single* neural network.

## 1 Introduction

Deep learning models are becoming deeper and wider than ever before. From image recognition models such as ResNet-101 (He et al., 2016a) and DenseNet (Huang et al., 2017) to BERT (Xu et al., 2019) and GPT-3 (Brown et al., 2020) for language modelling, deep neural networks have found consistent success in fitting large-scale data. As these models are increasingly deployed in real-world applications, calibrated uncertainty estimates for their predictions become crucial, especially in safety-critical areas such as healthcare. In this regard, Bayesian Neural Networks (BNNs) (MacKay, 1995; Blundell et al., 2015; Gal & Ghahramani, 2016; Zhang et al., 2020) and deep ensembles (Lakshminarayanan et al., 2017) represent two popular paradigms for estimating uncertainty, which have shown promising results in applications such as (medical) image processing (Kendall & Gal, 2017; Tanno et al., 2017) and out-of-distribution detection (Ovadia et al., 2019).

Though progress has been made, one major obstacle to scaling up BNNs and deep ensembles is their high storage cost. Both approaches require the parameter counts to be several times higher than their deterministic counterparts. Although recent efforts have improved memory efficiency (Louizos & Welling, 2017; Świątkowski et al., 2020; Wen et al., 2020; Dusenberry et al., 2020), these still use more parameters than a deterministic neural network. This is particularly problematic in hardware-constrained edge devices, when on-device storage is required due to privacy regulations.

Meanwhile, an *infinitely wide* BNN becomes a Gaussian process (GP) that is known for good uncertainty estimates (Neal, 1995; Matthews et al., 2018; Lee et al., 2018). But perhaps surprisingly, this infinitely wide BNN is "parameter efficient", as its "parameters" are effectively the datapoints, which have a considerably smaller memory footprint than explicitly storing the network weights. In addition, sparse posterior approximations store a smaller number of *inducing points* instead (Snelson & Ghahramani, 2006; Titsias, 2009), making sparse GPs even more memory efficient.

---

*Work done at Microsoft Research Cambridge.

35th Conference on Neural Information Processing Systems (NeurIPS 2021).

Can we bring the advantages of sparse approximations in GPs — which are infinitely-wide neural networks — to finite width deep learning models? We provide an affirmative answer regarding memory efficiency, by proposing an uncertainty quantification framework based on *sparse uncertainty representations*. We present our approach in BNN context, but the proposed approach is also applicable to deep ensembles. In detail, our contributions are as follows:

- We introduce *inducing weights* — an auxiliary variable method with lower dimensional counterparts to the actual weight matrices — for variational inference in BNNs, as well as a memory efficient parameterisation and an extension to ensemble methods (Section 3.1).
- We extend Matheron's rule to facilitate efficient posterior sampling (Section 3.2).
- We provide an in-depth computation complexity analysis (Section 3.3), showing the significant advantage in terms of parameter efficiency.
- We show the connection to sparse (deep) GPs, in that inducing weights can be viewed as *projected noisy inducing outputs* in pre-activation output space (Section 5.1).
- We apply the proposed approach to BNNs and deep ensembles. Experiments in classification, model robustness and out-of-distribution detection tasks show that our inducing weight approaches achieve competitive performance to their counterparts in the original weight space on modern deep architectures for image classification, while reducing the parameter count to $\leq 24.3\%$ of that of a single network.

We open-source our proposed inducing weight approach, together with baseline methods reported in the experiments, as a PyTorch (Paszke et al., 2019) wrapper named `bayesianize`: `https://github.com/microsoft/bayesianize`. As demonstrated in Appendix I, our software makes the conversion of a deterministic neural network to a Bayesian one with a few lines of code:

```
import bnn # our pytorch wrapper package
net = torchvision.models.resnet18() # construct a deterministic ResNet18
bnn.bayesianize_(net, inference="inducing") # convert it into a Bayesian one
```

## 2 Inducing variables for variational inference

Our work is built on variational inference and inducing variables for posterior approximations. Given observations $\mathcal{D} = \{\mathbf{X}, \mathbf{Y}\}$ with $\mathbf{X} = [\boldsymbol{x}_1, ..., \boldsymbol{x}_N]$, $\mathbf{Y} = [\boldsymbol{y}_1, ..., \boldsymbol{y}_N]$, we would like to fit a neural network $p(\boldsymbol{y}|\boldsymbol{x}, \mathbf{W}_{1:L})$ with weights $\mathbf{W}_{1:L}$ to the data. BNNs posit a prior distribution $p(\mathbf{W}_{1:L})$ over the weights, and construct an approximate posterior $q(\mathbf{W}_{1:L})$ to the exact posterior $p(\mathbf{W}_{1:L}|\mathcal{D}) \propto p(\mathcal{D}|\mathbf{W}_{1:L})p(\mathbf{W}_{1:L})$, where $p(\mathcal{D}|\mathbf{W}_{1:L}) = p(\mathbf{Y}|\mathbf{X}, \mathbf{W}_{1:L}) = \prod_{n=1}^{N} p(\boldsymbol{y}_n|\boldsymbol{x}_n, \mathbf{W}_{1:L})$.

**Variational inference** Variational inference (Hinton & Van Camp, 1993; Jordan et al., 1999; Zhang et al., 2018a) constructs an approximation $q(\boldsymbol{\theta})$ to the posterior $p(\boldsymbol{\theta}|\mathcal{D}) \propto p(\boldsymbol{\theta})p(\mathcal{D}|\boldsymbol{\theta})$ by maximising a variational lower-bound:

$$\log p(\mathcal{D}) \geq \mathcal{L}(q(\boldsymbol{\theta})) := \mathbb{E}_{q(\boldsymbol{\theta})}\left[\log p(\mathcal{D}|\boldsymbol{\theta})\right] - \mathbb{KL}\left[q(\boldsymbol{\theta})||p(\boldsymbol{\theta})\right]. \quad (1)$$

For BNNs, $\boldsymbol{\theta} = \{\mathbf{W}_{1:L}\}$, and a simple choice of $q$ is a Fully-factorized Gaussian (FFG): $q(\mathbf{W}_{1:L}) = \prod_{l=1}^{L} \prod_{i=1}^{d_{out}^l} \prod_{j=1}^{d_{in}^l} \mathcal{N}(m_l^{(i,j)}, v_l^{(i,j)})$, with $m_l^{(i,j)}, v_l^{(i,j)}$ the mean and variance of $\mathbf{W}_l^{(i,j)}$ and $d_{in}^l, d_{out}^l$ the respective number of inputs and outputs to layer $l$. The variational parameters are then $\boldsymbol{\phi} = \{m_l^{(i,j)}, v_l^{(i,j)}\}_{l=1}^{L}$. Gradients of $\mathcal{L}$ w.r.t. $\boldsymbol{\phi}$ can be estimated with mini-batches of data (Hoffman et al., 2013) and with Monte Carlo sampling from the $q$ distribution (Titsias & Lázaro-Gredilla, 2014; Kingma & Welling, 2014). By setting $q$ to an BNN, a variational BNN can be trained with similar computational requirements as a deterministic network (Blundell et al., 2015).

**Improved posterior approximation with inducing variables** Auxiliary variable approaches (Agakov & Barber, 2004; Salimans et al., 2015; Ranganath et al., 2016) construct the $q(\boldsymbol{\theta})$ distribution with an auxiliary variable $\mathbf{a}$: $q(\boldsymbol{\theta}) = \int q(\boldsymbol{\theta}|\mathbf{a})q(\mathbf{a})d\mathbf{a}$, with the hope that a potentially richer mixture distribution $q(\boldsymbol{\theta})$ can achieve better approximations. As then $q(\boldsymbol{\theta})$ becomes intractable, an auxiliary variational lower-bound is used to optimise $q(\boldsymbol{\theta}, \mathbf{a})$ (see Appendix B):

$$\log p(\mathcal{D}) \geq \mathcal{L}(q(\boldsymbol{\theta}, \mathbf{a})) = \mathbb{E}_{q(\boldsymbol{\theta}, \mathbf{a})}[\log p(\mathcal{D}|\boldsymbol{\theta})] + \mathbb{E}_{q(\boldsymbol{\theta}, \mathbf{a})}\left[\log \frac{p(\boldsymbol{\theta})r(\mathbf{a}|\boldsymbol{\theta})}{q(\boldsymbol{\theta}|\mathbf{a})q(\mathbf{a})}\right]. \quad (2)$$

Here $r(\mathbf{a}|\boldsymbol{\theta})$ is an auxiliary distribution that needs to be specified, where existing approaches often use a "reverse model" for $r(\mathbf{a}|\boldsymbol{\theta})$. Instead, we define $r(\mathbf{a}|\boldsymbol{\theta})$ in a generative manner: $r(\mathbf{a}|\boldsymbol{\theta})$ is the "posterior" of the following "generative model", whose "evidence" is exactly the prior of $\boldsymbol{\theta}$:

$$r(\mathbf{a}|\boldsymbol{\theta}) = \tilde{p}(\mathbf{a}|\boldsymbol{\theta}) \propto \tilde{p}(\mathbf{a})\tilde{p}(\boldsymbol{\theta}|\mathbf{a}), \quad \text{such that } \tilde{p}(\boldsymbol{\theta}) := \int \tilde{p}(\mathbf{a})\tilde{p}(\boldsymbol{\theta}|\mathbf{a})d\mathbf{a} = p(\boldsymbol{\theta}). \qquad (3)$$

Plugging Eq. (3) into Eq. (2):

$$\mathcal{L}(q(\boldsymbol{\theta}, \mathbf{a})) = \mathbb{E}_{q(\boldsymbol{\theta})}[\log p(\mathcal{D}|\boldsymbol{\theta})] - \mathbb{E}_{q(\mathbf{a})}\left[\mathbb{KL}[q(\boldsymbol{\theta}|\mathbf{a})||\tilde{p}(\boldsymbol{\theta}|\mathbf{a})]\right] - \mathbb{KL}[q(\mathbf{a})||\tilde{p}(\mathbf{a})]. \qquad (4)$$

This approach yields an efficient approximate inference algorithm, translating the complexity of inference in $\boldsymbol{\theta}$ to $\mathbf{a}$. If $\dim(\mathbf{a}) < \dim(\boldsymbol{\theta})$ and $q(\boldsymbol{\theta}, \mathbf{a}) = q(\boldsymbol{\theta}|\mathbf{a})q(\mathbf{a})$ has the following properties:

1. A "pseudo prior" $\tilde{p}(\mathbf{a})\tilde{p}(\boldsymbol{\theta}|\mathbf{a})$ is defined such that $\int \tilde{p}(\mathbf{a})\tilde{p}(\boldsymbol{\theta}|\mathbf{a})d\mathbf{a} = p(\boldsymbol{\theta})$;
2. The conditionals $q(\boldsymbol{\theta}|\mathbf{a})$ and $\tilde{p}(\boldsymbol{\theta}|\mathbf{a})$ are in the same parametric family, so can share parameters;
3. Both sampling $\boldsymbol{\theta} \sim q(\boldsymbol{\theta})$ and computing $\mathbb{KL}[q(\boldsymbol{\theta}|\mathbf{a})||\tilde{p}(\boldsymbol{\theta}|\mathbf{a})]$ can be done efficiently;
4. The designs of $q(\mathbf{a})$ and $\tilde{p}(\mathbf{a})$ can potentially provide extra advantages (in time and space complexities and/or optimisation easiness).

We call $\mathbf{a}$ the *inducing variable* of $\boldsymbol{\theta}$, which is inspired by variationally sparse GP (SVGP) with inducing points (Snelson & Ghahramani, 2006; Titsias, 2009). Indeed SVGP is a special case (see Appendix C): $\boldsymbol{\theta} = \mathbf{f}$, $\mathbf{a} = \mathbf{u}$, the GP prior is $p(\mathbf{f}|\mathbf{X}) = \mathcal{GP}(\mathbf{0}, \mathbf{K}_{\mathbf{XX}})$, $p(\mathbf{u}) = \mathcal{GP}(\mathbf{0}, \mathbf{K}_{\mathbf{ZZ}})$, $\tilde{p}(\mathbf{f}, \mathbf{u}) = p(\mathbf{u})p(\mathbf{f}|\mathbf{X}, \mathbf{u})$, $q(\mathbf{f}|\mathbf{u}) = p(\mathbf{f}|\mathbf{X}, \mathbf{u})$, $q(\mathbf{f}, \mathbf{u}) = p(\mathbf{f}|\mathbf{X}, \mathbf{u})q(\mathbf{u})$, and $\mathbf{Z}$ are the optimisable inducing inputs. The variational lower-bound is $\mathcal{L}(q(\mathbf{f}, \mathbf{u})) = \mathbb{E}_{q(\mathbf{f})}[\log p(\mathbf{Y}|\mathbf{f})] - \mathbb{KL}[q(\mathbf{u})||p(\mathbf{u})]$, and the variational parameters are $\phi = \{\mathbf{Z}, \text{distribution parameters of } q(\mathbf{u})\}$. SVGP satisfies the marginalisation constraint Eq. (3) by definition, and it has $\mathbb{KL}[q(\mathbf{f}|\mathbf{u})||\tilde{p}(\mathbf{f}|\mathbf{u})] = 0$. Also by using small $M = \dim(\mathbf{u})$ and exploiting the $q$ distribution design, SVGP reduces run-time from $\mathcal{O}(N^3)$ to $\mathcal{O}(NM^2 + M^3)$ where $N$ is the number of inputs in $\mathbf{X}$, meanwhile it also makes storing a full Gaussian $q(\mathbf{u})$ affordable. Lastly, $\mathbf{u}$ can be whitened, leading to the "pseudo prior" $\tilde{p}(\mathbf{f}, \mathbf{v}) = p(\mathbf{f}|\mathbf{X}, \mathbf{u} = \mathbf{K}_{\mathbf{ZZ}}^{1/2}\mathbf{v})\tilde{p}(\mathbf{v})$, $\tilde{p}(\mathbf{v}) = \mathcal{N}(\mathbf{v}; \mathbf{0}, \mathbf{I})$ which could bring potential benefits in optimisation.

We emphasise that the introduction of "pseudo prior" does *not* change the *probabilistic model* as long as the marginalisation constraint Eq. (3) is satisfied. In the rest of the paper we assume the constraint Eq. (3) holds and write $p(\boldsymbol{\theta}, \mathbf{a}) := \tilde{p}(\boldsymbol{\theta}, \mathbf{a})$. It might seem unclear how to design such $\tilde{p}(\boldsymbol{\theta}, \mathbf{a})$ for an arbitrary probabilistic model, however, for a Gaussian prior on $\boldsymbol{\theta}$ the rules for computing conditional Gaussian distributions can be used to construct $\tilde{p}$. In Section 3 we exploit these rules to develop an efficient approximate inference method for Bayesian neural networks with inducing weights.

## 3 Sparse uncertainty representation with inducing weights

### 3.1 Inducing weights for neural network parameters

Following the above design principles, we introduce to each network layer $l$ a *smaller* inducing weight matrix $\mathbf{U}_l$ to assist approximate posterior inference in $\mathbf{W}_l$. Therefore in our context, $\boldsymbol{\theta} = \mathbf{W}_{1:L}$ and $\mathbf{a} = \mathbf{U}_{1:L}$. In the rest of the paper, we assume a factorised prior across layers $p(\mathbf{W}_{1:L}) = \prod_l p(\mathbf{W}_l)$, and drop the $l$ indices when the context is clear to ease notation.

**Augmenting network layers with inducing weights** Suppose the weight $\mathbf{W} \in \mathbb{R}^{d_{out} \times d_{in}}$ has a Gaussian prior $p(\mathbf{W}) = p(\text{vec}(\mathbf{W})) = \mathcal{N}(0, \sigma^2\boldsymbol{I})$ where $\text{vec}(\mathbf{W})$ concatenates the columns of the weight matrix into a vector. A first attempt to augment $p(\text{vec}(\mathbf{W}))$ with an inducing weight variable $\mathbf{U} \in \mathbb{R}^{M_{out} \times M_{in}}$ may be to construct a multivariate Gaussian $p(\text{vec}(\mathbf{W}), \text{vec}(\mathbf{U}))$, such that $\int p(\text{vec}(\mathbf{W}), \text{vec}(\mathbf{U}))d\mathbf{U} = \mathcal{N}(0, \sigma^2\boldsymbol{I})$. This means for the joint covariance matrix of $(\text{vec}(\mathbf{W}), \text{vec}(\mathbf{U}))$, it requires the block corresponding to the covariance of $\text{vec}(\mathbf{W})$ to match the prior covariance $\sigma^2\boldsymbol{I}$. We are then free to parameterise the rest of the entries in the joint covariance matrix, as long as this full matrix remains positive definite. Now the conditional distribution $p(\mathbf{W}|\mathbf{U})$ is a function of these parameters, and the conditional sampling from $p(\mathbf{W}|\mathbf{U})$ is further discussed in Appendix D.1. Unfortunately, as $\dim(\text{vec}(\mathbf{W}))$ is typically large (e.g. of the order of $10^7$), using a full covariance Gaussian for $p(\text{vec}(\mathbf{W}), \text{vec}(\mathbf{U}))$ becomes computationally intractable.

We address this issue with matrix normal distributions (Gupta & Nagar, 2018). The prior $p(\text{vec}(\mathbf{W})) = \mathcal{N}(\mathbf{0}, \sigma^2\boldsymbol{I})$ has an equivalent matrix normal distribution form as $p(\mathbf{W}) =$

$\mathcal{MN}(0, \sigma_r^2 \boldsymbol{I}, \sigma_c^2 \boldsymbol{I})$, with $\sigma_r, \sigma_c > 0$ the row and column standard deviations satisfying $\sigma = \sigma_r \sigma_c$. Now we introduce the inducing variable $\mathbf{U}$ in matrix space, as well as two auxiliary variables $\mathbf{U}_r \in \mathbb{R}^{M_{out} \times d_{in}}$, $\mathbf{U}_c \in \mathbb{R}^{d_{out} \times M_{in}}$, so that the full augmented prior is:

$$\begin{pmatrix} \mathbf{W} & \mathbf{U}_c \\ \mathbf{U}_r & \mathbf{U} \end{pmatrix} \sim p(\mathbf{W}, \mathbf{U}_c, \mathbf{U}_r, \mathbf{U}) := \mathcal{MN}(0, \boldsymbol{\Sigma}_r, \boldsymbol{\Sigma}_c), \tag{5}$$

$$\text{with} \quad \boldsymbol{L}_r = \begin{pmatrix} \sigma_r \boldsymbol{I} & 0 \\ \boldsymbol{Z}_r & \boldsymbol{D}_r \end{pmatrix} \quad \text{s.t.} \quad \boldsymbol{\Sigma}_r = \boldsymbol{L}_r \boldsymbol{L}_r^\top = \begin{pmatrix} \sigma_r^2 \boldsymbol{I} & \sigma_r \boldsymbol{Z}_r^\top \\ \sigma_r \boldsymbol{Z}_r & \boldsymbol{Z}_r \boldsymbol{Z}_r^\top + \boldsymbol{D}_r^2 \end{pmatrix}$$

$$\text{and} \quad \boldsymbol{L}_c = \begin{pmatrix} \sigma_c \boldsymbol{I} & 0 \\ \boldsymbol{Z}_c & \boldsymbol{D}_c \end{pmatrix} \quad \text{s.t.} \quad \boldsymbol{\Sigma}_c = \boldsymbol{L}_c \boldsymbol{L}_c^\top = \begin{pmatrix} \sigma_c^2 \boldsymbol{I} & \sigma_c \boldsymbol{Z}_c^\top \\ \sigma_c \boldsymbol{Z}_c & \boldsymbol{Z}_c \boldsymbol{Z}_c^\top + \boldsymbol{D}_c^2 \end{pmatrix}.$$

See Fig. 1(a) for a visualisation of the augmentation. Matrix normal distributions have similar marginalisation and conditioning rules as multivariate Gaussian distributions, for which we provide further examples in Appendix D.2. Therefore the marginalisation constraint Eq. (3) is satisfied for any $\boldsymbol{Z}_c \in \mathbb{R}^{M_{in} \times d_{in}}$, $\boldsymbol{Z}_r \in \mathbb{R}^{M_{out} \times d_{out}}$ and diagonal matrices $\boldsymbol{D}_c, \boldsymbol{D}_r$. For the inducing weight $\mathbf{U}$ we have $p(\mathbf{U}) = \mathcal{MN}(0, \boldsymbol{\Psi}_r, \boldsymbol{\Psi}_c)$ with $\boldsymbol{\Psi}_r = \boldsymbol{Z}_r \boldsymbol{Z}_r^\top + \boldsymbol{D}_r^2$ and $\boldsymbol{\Psi}_c = \boldsymbol{Z}_c \boldsymbol{Z}_c^\top + \boldsymbol{D}_c^2$. In the experiments we use *whitened* inducing weights which transforms $\mathbf{U}$ so that $p(\mathbf{U}) = \mathcal{MN}(0, \boldsymbol{I}, \boldsymbol{I})$ (Appendix H), but for clarity we continue with the above formulas in the main text.

The matrix normal parameterisation introduces two additional variables $\mathbf{U}_r, \mathbf{U}_c$ without providing additional expressiveness. Hence it is desirable to integrate them out, leading to a joint multivariate normal with Khatri-Rao product structure for the covariance:

$$p(\text{vec}(\mathbf{W}), \text{vec}(\mathbf{U})) = \mathcal{N}\left(0, \begin{pmatrix} \sigma_c^2 \boldsymbol{I} \otimes \sigma_r^2 \boldsymbol{I} & \sigma_c \boldsymbol{Z}_c^\top \otimes \sigma_r \boldsymbol{Z}_r^\top \\ \sigma_c \boldsymbol{Z}_c \otimes \sigma_r \boldsymbol{Z}_r & \boldsymbol{\Psi}_c \otimes \boldsymbol{\Psi}_r \end{pmatrix}\right). \tag{6}$$

As the dominating memory complexity here is $\mathcal{O}(d_{out} M_{out} + d_{in} M_{in})$ which comes from storing $\boldsymbol{Z}_r$ and $\boldsymbol{Z}_c$, we see that the matrix normal parameterisation of the augmented prior is memory efficient.

**Posterior approximation in the joint space** We construct a factorised posterior approximation across the layers: $q(\mathbf{W}_{1:L}, \mathbf{U}_{1:L}) = \prod_l q(\mathbf{W}_l | \mathbf{U}_l) q(\mathbf{U}_l)$. Below we discuss options for $q(\mathbf{W}|\mathbf{U})$.

The simplest option is $q(\mathbf{W}|\mathbf{U}) = p(\text{vec}(\mathbf{W})| \text{vec}(\mathbf{U})) = \mathcal{N}(\boldsymbol{\mu}_{\mathbf{W}|\mathbf{U}}, \boldsymbol{\Sigma}_{\mathbf{W}|\mathbf{U}})$, similar to sparse GPs. A slightly more flexible variant adds a rescaling term $\lambda^2$ to the covariance matrix, which allows efficient KL computation (Appendix E):

$$q(\mathbf{W}|\mathbf{U}) = q(\text{vec}(\mathbf{W})| \text{vec}(\mathbf{U})) = \mathcal{N}(\boldsymbol{\mu}_{\mathbf{W}|\mathbf{U}}, \lambda^2 \boldsymbol{\Sigma}_{\mathbf{W}|\mathbf{U}}), \tag{7}$$

$$R(\lambda) := \mathbb{KL}\left[q(\mathbf{W}|\mathbf{U}) || p(\mathbf{W}|\mathbf{U})\right] = d_{in} d_{out}(0.5\lambda^2 - \log \lambda - 0.5). \tag{8}$$

Plugging $\boldsymbol{\theta} = \mathbf{W}_{1:L}, a = \mathbf{U}_{1:L}$ and Eq. (8) into Eq. (4) returns the following variational lower-bound

$$\mathcal{L}(q(\mathbf{W}_{1:L}, \mathbf{U}_{1:L})) = \mathbb{E}_{q(\mathbf{W}_{1:L})}[\log p(\mathcal{D}|\mathbf{W}_{1:L})] - \sum\nolimits_{l=1}^{L}(R(\lambda_l) + \mathbb{KL}[q(\mathbf{U}_l) || p(\mathbf{U}_l)]), \tag{9}$$

with $\lambda_l$ the associated scaling parameter for $q(\mathbf{W}_l | \mathbf{U}_l)$. Again as the choices of $\boldsymbol{Z}_c, \boldsymbol{Z}_r, \boldsymbol{D}_c, \boldsymbol{D}_r$ do not change the marginal prior $p(\mathbf{W})$, we are safe to optimise them as well. Therefore the variational parameters are now $\phi = \{\boldsymbol{Z}_c, \boldsymbol{Z}_r, \boldsymbol{D}_c, \boldsymbol{D}_r, \lambda, \text{dist. params. of } q(\mathbf{U})\}$ for each layer.

**Two choices of $q(\mathbf{U})$** A simple choice is FFG $q(\text{vec}(\mathbf{U})) = \mathcal{N}(\boldsymbol{m}_u, \text{diag}(\boldsymbol{v}_u))$, which performs mean-field inference in $\mathbf{U}$ space (c.f. Blundell et al., 2015), and here $\mathbb{KL}[q(\mathbf{U})||p(\mathbf{U})]$ has a closed-form solution. Another choice is a "mixture of delta measures" $q(\mathbf{U}) = \frac{1}{K} \sum_{k=1}^{K} \delta(\mathbf{U} = \mathbf{U}^{(k)})$, i.e. we keep $K$ distinct sets of parameters $\{U_{1:L}^{(k)}\}_{k=1}^{K}$ in inducing space that are projected back into the original parameter space via the *shared* conditionals $q(\mathbf{W}_l | \mathbf{U}_l)$ to obtain the weights. This approach can be viewed as constructing "deep ensembles" in $\mathbf{U}$ space, and we follow ensemble methods (e.g. Lakshminarayanan et al., 2017) to drop $\mathbb{KL}[q(\mathbf{U})||p(\mathbf{U})]$ in Eq. (9).

Often $\mathbf{U}$ is chosen to have significantly lower dimensions than $\mathbf{W}$, i.e. $M_{in} << d_{in}$ and $M_{out} << d_{out}$. As $q(\mathbf{W}|\mathbf{U})$ and $p(\mathbf{W}|\mathbf{U})$ only differ in the covariance scaling constant, $\mathbf{U}$ can be regarded as a *sparse representation of uncertainty* for the network layer, as the major updates in (approximate) posterior belief is quantified by $q(\mathbf{U})$.

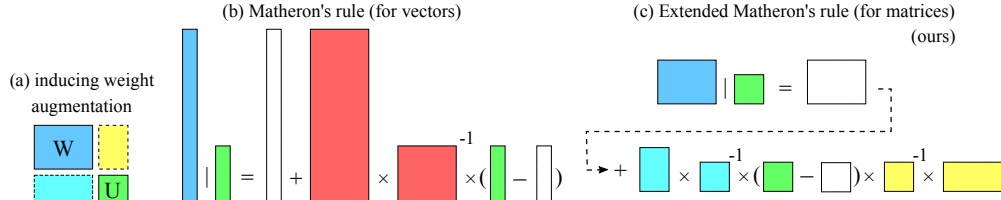

Figure 1: Visualisation of (a) the inducing weight augmentation, and compare (b) the original Matheron's rule to (c) our extended version. White blocks represent samples from the joint Gaussian.

### 3.2 Efficient sampling with extended Matheron's rule

Computing the variational lower-bound Eq. (9) requires samples from $q(\mathbf{W})$, which requires an efficient sampling procedure for $q(\mathbf{W}|\mathbf{U})$. Unfortunately, $q(\mathbf{W}|\mathbf{U})$ derived from Eq. (6) & Eq. (7) is not a matrix normal, so *direct* sampling is prohibitive. To address this challenge, we extend Matheron's rule (Journel & Huijbregts, 1978; Hoffman & Ribak, 1991; Doucet, 2010) to efficiently sample from $q(\mathbf{W}|\mathbf{U})$, with derivations provided in Appendix F.

The original Matheron's rule applies to multivariate Gaussian distributions. As a running example, consider two vector-valued random variables $\boldsymbol{w}, \boldsymbol{u}$ with joint distribution $p(\boldsymbol{w}, \boldsymbol{u}) = \mathcal{N}(\mathbf{0}, \boldsymbol{\Sigma})$. Then the conditional distribution $p(\boldsymbol{w}|\boldsymbol{u}) = \mathcal{N}(\boldsymbol{\mu}_{\boldsymbol{w}|\boldsymbol{u}}, \boldsymbol{\Sigma}_{\boldsymbol{w}|\boldsymbol{u}})$ is also Gaussian, and direct sampling from it requires decomposing the conditional covariance matrix $\boldsymbol{\Sigma}_{\boldsymbol{w}|\boldsymbol{u}}$ which can be costly. The main idea of Matheron's rule is that we can transform a sample from the joint Gaussian to obtain a sample from the conditional distribution $p(\boldsymbol{w}|\boldsymbol{u})$ as follows:

$$\boldsymbol{w} = \bar{\boldsymbol{w}} + \boldsymbol{\Sigma}_{\boldsymbol{wu}}\boldsymbol{\Sigma}_{\boldsymbol{uu}}^{-1}(\boldsymbol{u} - \bar{\boldsymbol{u}}), \quad \bar{\boldsymbol{w}}, \bar{\boldsymbol{u}} \sim \mathcal{N}(\mathbf{0}, \boldsymbol{\Sigma}), \quad \boldsymbol{\Sigma} = \begin{pmatrix} \boldsymbol{\Sigma}_{\boldsymbol{ww}} & \boldsymbol{\Sigma}_{\boldsymbol{wu}} \\ \boldsymbol{\Sigma}_{\boldsymbol{uw}} & \boldsymbol{\Sigma}_{\boldsymbol{uu}} \end{pmatrix}. \quad (10)$$

One can check the validity of Matheron's rule by computing the mean and variance of $\boldsymbol{w}$ above:

$$\mathbb{E}_{\bar{\boldsymbol{w}}, \bar{\boldsymbol{u}}}[\boldsymbol{w}] = \boldsymbol{\Sigma}_{\boldsymbol{wu}}\boldsymbol{\Sigma}_{\boldsymbol{uu}}^{-1}\boldsymbol{u} = \boldsymbol{\mu}_{\boldsymbol{w}|\boldsymbol{u}}, \quad \mathbb{V}_{\bar{\boldsymbol{w}}, \bar{\boldsymbol{u}}}[\boldsymbol{w}] = \boldsymbol{\Sigma}_{\boldsymbol{ww}} - \boldsymbol{\Sigma}_{\boldsymbol{wu}}\boldsymbol{\Sigma}_{\boldsymbol{uu}}^{-1}\boldsymbol{\Sigma}_{\boldsymbol{uw}} = \boldsymbol{\Sigma}_{\boldsymbol{w}|\boldsymbol{u}}.$$

It might seem counter-intuitive at first sight in that this rule requires samples from a higher dimensional space. However, in the case where decomposition/inversion of $\boldsymbol{\Sigma}$ and $\boldsymbol{\Sigma}_{\boldsymbol{uu}}$ can be done efficiently, sampling from the joint Gaussian $p(\boldsymbol{w}, \boldsymbol{u})$ can be significantly cheaper than directly sampling from the conditional Gaussian $p(\boldsymbol{w}|\boldsymbol{u})$. This happens e.g. when $\boldsymbol{\Sigma}$ is directly parameterised by its Cholesky decomposition and $\dim(\boldsymbol{u}) << \dim(\boldsymbol{w})$, so that sampling $\bar{\boldsymbol{w}}, \bar{\boldsymbol{u}} \sim \mathcal{N}(\mathbf{0}, \boldsymbol{\Sigma})$ is straight-forward, and computing $\boldsymbol{\Sigma}_{\boldsymbol{uu}}^{-1}$ is significantly cheaper than decomposing $\boldsymbol{\Sigma}_{\boldsymbol{w}|\boldsymbol{u}}$.

Unfortunately, the original Matheron's rule cannot be applied directly to sample from $q(\mathbf{W}|\mathbf{U})$. This is because $q(\mathbf{W}|\mathbf{U}) = q(\text{vec}(\mathbf{W})|\text{vec}(\mathbf{U}))$ differs from $p(\text{vec}(\mathbf{W})|\text{vec}(\mathbf{U}))$ only in the variance scaling $\lambda$, and for $p(\text{vec}(\mathbf{W})|\text{vec}(\mathbf{U}))$, its joint distribution counter-part Eq. (6) does not have an efficient representation for the covariance matrix. Therefore a naive application of Matheron's rule requires decomposing the covariance matrix of $p(\text{vec}(\mathbf{W}), \text{vec}(\mathbf{U}))$ which is even more expensive than direct conditional sampling. However, notice that for the joint distribution $p(\mathbf{W}, \mathbf{U}_c, \mathbf{U}_r, \mathbf{U})$ in an even higher dimensional space, the row and column covariance matrices $\boldsymbol{\Sigma}_r$ and $\boldsymbol{\Sigma}_c$ are parameterised by their Cholesky decompositions, so that sampling from this joint distribution can be done efficiently. This inspire us to extend the original Matheron's rule for efficient sampling from $q(\mathbf{W}|\mathbf{U})$ (details in Appendix F, when $\lambda = 1$ it also applies to sampling from $p(\mathbf{W}|\mathbf{U})$):

$$\mathbf{W} = \lambda\bar{\mathbf{W}} + \sigma\boldsymbol{Z}_r^\top\boldsymbol{\Psi}_r^{-1}(\mathbf{U} - \lambda\bar{\mathbf{U}})\boldsymbol{\Psi}_c^{-1}\boldsymbol{Z}_c; \quad \bar{\mathbf{W}}, \bar{\mathbf{U}} \sim p(\bar{\mathbf{W}}, \bar{\mathbf{U}}_c, \bar{\mathbf{U}}_r, \bar{\mathbf{U}}) = \mathcal{MN}(0, \boldsymbol{\Sigma}_r, \boldsymbol{\Sigma}_c). \quad (11)$$

Here $\bar{\mathbf{W}}, \bar{\mathbf{U}} \sim p(\bar{\mathbf{W}}, \bar{\mathbf{U}}_c, \bar{\mathbf{U}}_r, \bar{\mathbf{U}})$ means we first sample $\bar{\mathbf{W}}, \bar{\mathbf{U}}_c, \bar{\mathbf{U}}_r, \bar{\mathbf{U}}$ from the joint then drop $\bar{\mathbf{U}}_c, \bar{\mathbf{U}}_r$; in fact $\bar{\mathbf{U}}_c, \bar{\mathbf{U}}_r$ are never computed, and the other samples $\bar{\mathbf{W}}, \bar{\mathbf{U}}$ can be obtained by:

$$\bar{\mathbf{W}} = \sigma\mathbf{E}_1, \quad \bar{\mathbf{U}} = \boldsymbol{Z}_r\mathbf{E}_1\boldsymbol{Z}_c^\top + \hat{\boldsymbol{L}}_r\tilde{\mathbf{E}}_2\boldsymbol{D}_c + \boldsymbol{D}_r\tilde{\mathbf{E}}_3\hat{\boldsymbol{L}}_c^\top + \boldsymbol{D}_r\mathbf{E}_4\boldsymbol{D}_c,$$

$$\mathbf{E}_1 \sim \mathcal{MN}(0, \boldsymbol{I}_{d_{out}}, \boldsymbol{I}_{d_{in}}); \quad \tilde{\mathbf{E}}_2, \tilde{\mathbf{E}}_3, \mathbf{E}_4 \sim \mathcal{MN}(0, \boldsymbol{I}_{M_{out}}, \boldsymbol{I}_{M_{in}}), \quad (12)$$

$$\hat{\boldsymbol{L}}_r = \text{chol}(\boldsymbol{Z}_r\boldsymbol{Z}_r^\top), \quad \hat{\boldsymbol{L}}_c = \text{chol}(\boldsymbol{Z}_c\boldsymbol{Z}_c^\top).$$

The run-time cost is $\mathcal{O}(2M_{out}^3 + 2M_{in}^3 + d_{out}M_{out}M_{in} + M_{in}d_{out}d_{in})$ required by inverting $\boldsymbol{\Psi}_r, \boldsymbol{\Psi}_c$, computing $\hat{\boldsymbol{L}}_r, \hat{\boldsymbol{L}}_c$, and the matrix products. The extended Matheron's rule is visualised in Fig. 1

Table 1: Computational complexity per layer. We assume $\mathbf{W} \in \mathbb{R}^{d_{out} \times d_{in}}$, $\mathbf{U} \in \mathbb{R}^{M_{out} \times M_{in}}$, and $K$ forward passes for each of the $N$ inputs. (* uses a parallel computing friendly vectorisation technique (Wen et al., 2020) for further speed-up.)

| Method | Time complexity | Storage complexity |
|---|---|---|
| Deterministic-$\mathbf{W}$ | $\mathcal{O}(Nd_{in}d_{out})$ | $\mathcal{O}(d_{in}d_{out})$ |
| FFG-$\mathbf{W}$ | $\mathcal{O}(NKd_{in}d_{out})$ | $\mathcal{O}(2d_{in}d_{out})$ |
| Ensemble-$\mathbf{W}$ | $\mathcal{O}(NKd_{in}d_{out})$ | $\mathcal{O}(Kd_{in}d_{out})$ |
| Matrix-normal-$\mathbf{W}$ | $\mathcal{O}(NKd_{in}d_{out})$ | $\mathcal{O}(d_{in}d_{out} + d_{in} + d_{out})$ |
| $k$-tied FFG-$\mathbf{W}$ | $\mathcal{O}(NKd_{in}d_{out})$ | $\mathcal{O}(d_{in}d_{out} + k(d_{in} + d_{out}))$ |
| rank-1 BNN | $\mathcal{O}(NKd_{in}d_{out})^*$ | $\mathcal{O}(d_{in}d_{out} + 2(d_{in} + d_{out}))$ |
| FFG-$\mathbf{U}$ | $\mathcal{O}(NKd_{in}d_{out} + 2M_{in}^3 + 2M_{out}^3 + K(d_{out}M_{out}M_{in} + M_{in}d_{out}d_{in}))$ | $\mathcal{O}(d_{in}M_{in} + d_{out}M_{out} + 2M_{in}M_{out})$ |
| Ensemble-$\mathbf{U}$ | same as above | $\mathcal{O}(d_{in}M_{in} + d_{out}M_{out} + KM_{in}M_{out})$ |

with a comparison to the original Matheron's rule for sampling from $q(\text{vec}(\mathbf{W}) | \text{vec}(\mathbf{U}))$. Note that the original rule requires joint sampling from Eq. (6) (i.e. sampling the white blocks in Fig. 1(b)) which has $\mathcal{O}((d_{out}d_{in} + M_{out}M_{in})^3)$ cost. Therefore our recipe avoids inverting and multiplying big matrices, resulting in a significant speed-up for conditional sampling.

### 3.3 Computational complexities

In Table 1 we report the complexity figures for two types of inducing weight approaches: FFG $q(\mathbf{U})$ (FFG-$\mathbf{U}$) and Delta mixture $q(\mathbf{U})$ (Ensemble-$\mathbf{U}$). Baseline approaches include: Deterministic-$\mathbf{W}$, variational inference with FFG $q(\mathbf{W})$ (FFG-$\mathbf{W}$, Blundell et al., 2015), deep ensemble in $\mathbf{W}$ (Ensemble-$\mathbf{W}$, Lakshminarayanan et al., 2017), as well as parameter efficient approaches such as matrix-normal $q(\mathbf{W})$ (Matrix-normal-$\mathbf{W}$, Louizos & Welling (2017)), variational inference with $k$-tied FFG $q(\mathbf{W})$ ($k$-tied FFG-$\mathbf{W}$, Świątkowski et al. (2020)), and rank-1 BNN (Dusenberry et al., 2020). The gain in memory is significant for the inducing weight approaches, in fact with $M_{in} < d_{in}$ and $M_{out} < d_{out}$ the parameter storage requirement is smaller than a single deterministic neural network. The major overhead in run-time comes from the extended Matheron's rule for sampling $q(\mathbf{W}|\mathbf{U})$. Some of the computations there are performed only once, and in our experiments we show that by using a relatively low-dimensional $\mathbf{U}$ and large batch-sizes, the overhead is acceptable.

## 4 Experiments

We evaluate the inducing weight approaches on regression, classification and related uncertainty estimation tasks. The goal is to demonstrate competitive performance to popular $\mathbf{W}$-space uncertainty estimation methods while using significantly fewer parameters. We acknowledge that existing parameter efficient approaches for uncertainty estimation (e.g. k-tied or rank-1 BNNs) have achieved close performance to deep ensembles. However, *none* of them reduces the parameter count to be smaller than that of a *single* network. Therefore we decide *not* to include these baselines and instead focus on comparing: (1) variational inference with FFG $q(\mathbf{W})$ (FFG-$\mathbf{W}$, Blundell et al., 2015) v.s. FFG $q(\mathbf{U})$ (FFG-$\mathbf{U}$, ours); (2) deep ensemble in $\mathbf{W}$ space (Ensemble-$\mathbf{W}$, Lakshminarayanan et al., 2017) v.s. ensemble in $\mathbf{U}$ space (Ensemble-$\mathbf{U}$, ours). Another baseline is training a deterministic neural network with maximum likelihood. Details and additional results are in Appendices J and K.

### 4.1 Synthetic 1-D regression

The regression task follows Foong et al. (2019), which has two input clusters $x_1 \sim \mathcal{U}[-1, -0.7]$, $x_2 \sim \mathcal{U}[0.5, 1]$, and targets $y \sim \mathcal{N}(\cos(4x + 0.8), 0.01)$. For reference we show the exact posterior results using the NUTS sampler (Hoffman & Gelman, 2014). The results are visualised in Fig. 2 with predictive mean in blue, and up to three standard deviations as shaded area. Similar to historical results, FFG-$\mathbf{W}$ fails to represent the increased uncertainty away from the data and in between clusters. While underestimating predictive uncertainty overall, FFG-$\mathbf{U}$ shows a small increase in predictive uncertainty away from the data. In contrast, a per-layer Full-covariance Gaussian (FCG) in both weight (FCG-$\mathbf{W}$) and inducing space (FCG-$\mathbf{U}$) as well as Ensemble-$\mathbf{U}$ better capture the increased predictive variance, although the mean function is more similar to that of FFG-$\mathbf{W}$.

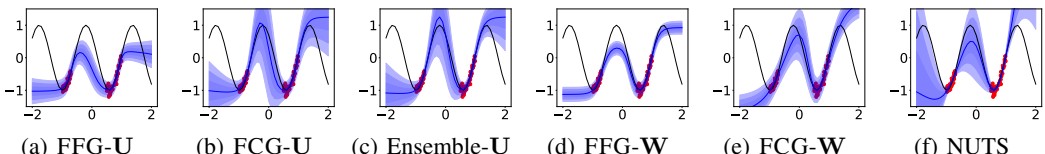

| (a) FFG-**U** | (b) FCG-**U** | (c) Ensemble-**U** | (d) FFG-**W** | (e) FCG-**W** | (f) NUTS |

Figure 2: Toy regression results, with observations in red dots and the ground truth function in black.

Table 2: CIFAR in-distribution metrics (in %).

| Method | CIFAR10 | | CIFAR100 | |
|---|---|---|---|---|
| | Acc. ↑ | ECE ↓ | Acc. ↑ | ECE ↓ |
| Deterministic | 94.72 | 4.46 | 75.73 | 19.69 |
| Ensemble-**W** | 95.90 | 1.08 | 79.33 | 6.51 |
| FFG-**W** | 94.13 | 0.50 | 74.44 | 4.24 |
| FFG-**U** | 94.40 | 0.64 | 75.37 | 2.29 |
| Ensemble-**U** | 94.94 | 0.45 | 75.97 | 1.12 |

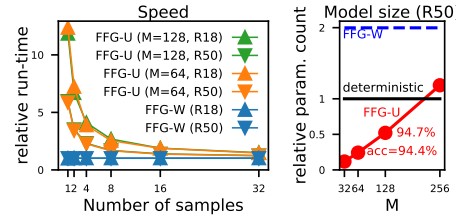

Figure 3: Resnet run-times & model sizes.

## 4.2 Classification and in-distribution calibration

As the core empirical evaluation, we train Resnet-50 models (He et al., 2016b) on CIFAR-10 and CIFAR-100 (Krizhevsky et al., 2009). To avoid underfitting issues with FFG-**W**, a useful trick is to set an upper limit $\sigma_{max}^2$ on the variance of $q(\mathbf{W})$ (Louizos & Welling, 2017). This trick is similarly applied to the **U**-space methods, where we cap $\lambda \leq \lambda_{max}$ for $q(\mathbf{W}|\mathbf{U})$, and for FFG-**U** we also set $\sigma_{max}^2$ for the variance of $q(\mathbf{U})$. In convolution layers, we treat the 4D weight tensor $\mathbf{W}$ of shape $(c_{out}, c_{in}, h, w)$ as a $c_{out} \times c_{in}hw$ matrix. We use **U** matrices of shape $64 \times 64$ for all layers (i.e. $M = M_{in} = M_{out} = 64$), except that for CIFAR-10 we set $M_{out} = 10$ for the last layer.

In Table 2 we report test accuracy and test expected calibration error (ECE) (Guo et al., 2017) as a first evaluation of the uncertainty estimates. Overall, Ensemble-**W** achieves the highest accuracy, but is not as well-calibrated as variational methods. For the inducing weight approaches, Ensemble-**U** outperforms FFG-**U** on both datasets; overall it performs the best on the more challenging CIFAR-100 dataset (close-to-Ensemble-**W** accuracy and lowest ECE). Tables 5 and 6 in Appendix K show that increasing the **U** dimensions to $M = 128$ improves accuracy but leads to slightly worse calibration.

In Fig. 3 we show prediction run-times for batch-size $= 500$ on an NVIDIA Tesla V100 GPU, relative to those of an ensemble of deterministic nets, as well as relative parameter sizes to a single ResNet-50. The extra run-times for the inducing methods come from computing the extended Matheron's rule. However, as they can be calculated once and cached for drawing multiple samples, the overhead reduces to a small factor when using larger number of samples $K$, especially for the bigger Resnet-50. More importantly, when compared to a *deterministic* ResNet-50, the inducing weight models reduce the parameter count by over 75% (5, 710, 902 vs. 23, 520, 842) for $M = 64$.

**Hyper-parameter choices**  We visualise in Fig. 4 the accuracy and ECE results for computation-ally lighter inducing weight ResNet-18 models with different hyper-parameters (see Appendix J).

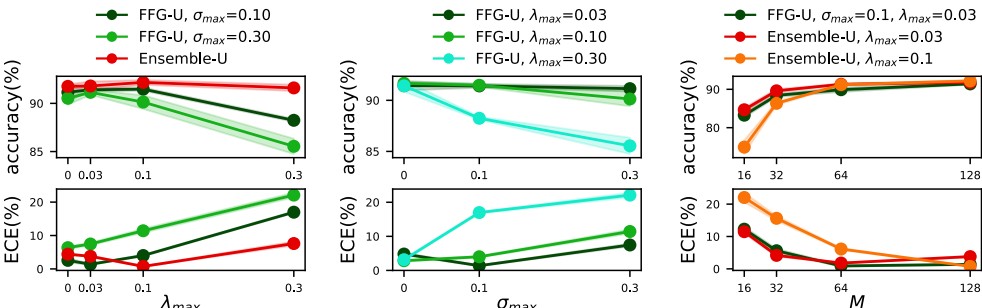

Figure 4: Ablation study: average CIFAR-10 accuracy (↑) and ECE (↓) for the inducing weight methods on ResNet-18. In the first two columns $M = 128$ for **U** dimensions. For $\lambda_{max}, \sigma_{max} = 0$ we use point estimates for **U**, **W** respectively.

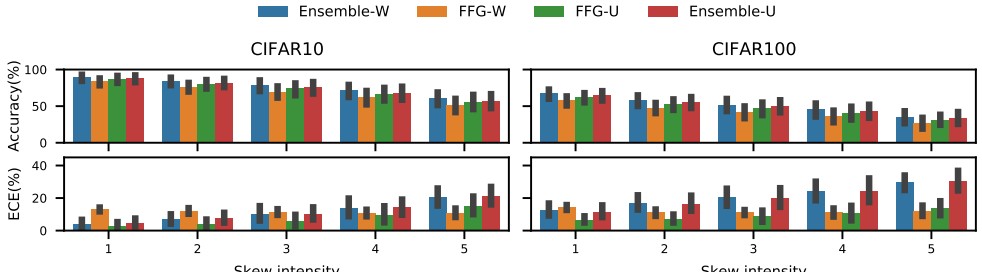

Figure 5: Mean±two errs. for Acc↑ and ECE↓ on corrupted CIFAR (Hendrycks & Dietterich, 2019).

Table 3: OOD detection metrics for Resnet-50 trained on CIFAR10/100.

| In-dist → OOD | C10 → C100 | | C10 → SVHN | | C100 → C10 | | C100 → SVHN | |
| Method / Metric | AUROC | AUPR | AUROC | AUPR | AUROC | AUPR | AUROC | AUPR |
|---|---|---|---|---|---|---|---|---|
| Deterministic | .84±.00 | .80±.00 | .93±.01 | .85±.01 | .74±.00 | .74±.00 | .81±.01 | .72±.02 |
| Ensemble-W | .89 | .89 | .95 | .92 | .78 | .79 | .86 | .78 |
| FFG-W | .88±.00 | .90±.00 | .90±.01 | .86±.01 | .76±.00 | .79±.00 | .80±.01 | .69±.01 |
| FFG-U | .89±.00 | .91±.00 | .94±.01 | .91±.01 | .77±.00 | .79±.00 | .83±.01 | .74±.01 |
| Ensemble-U | .90±.00 | .91±.00 | .93±.00 | .91±.00 | .77±.00 | .79±.00 | .82±.01 | .72±.02 |

Performance in both metrics improves as the $\mathbf{U}$ matrix size $M$ is increased (right-most panels), and the results for $M = 64$ and $M = 128$ are fairly similar. Also setting proper values for $\lambda_{max}, \sigma_{max}$ is key to the improved results. The left-most panels show that with fixed $\sigma_{max}$ values (or Ensemble-$\mathbf{U}$), the preferred conditional variance cap values $\lambda_{max}$ are fairly small (but still larger than 0 which corresponds to a point estimate for $\mathbf{W}$ given $\mathbf{U}$). For $\sigma_{max}$ which controls variance in $\mathbf{U}$ space, we see from the top middle panel that the accuracy metric is fairly robust to $\sigma_{max}$ as long as $\lambda_{max}$ is not too large. But for ECE, a careful selection of $\sigma_{max}$ is required (bottom middle panel).

### 4.3 Robustness, out-of-distribution detection and pruning

To investigate the models' robustness to distribution shift, we compute predictions on corrupted CIFAR datasets (Hendrycks & Dietterich, 2019) after training on clean data. Fig. 5 shows accuracy and ECE results for the ResNet-50 models. Ensemble-$\mathbf{W}$ is the most accurate model across skew intensities, while FFG-$\mathbf{W}$, though performing well on clean data, returns the worst accuracy under perturbation. The inducing weight methods perform competitively to Ensemble-$\mathbf{W}$ with Ensemble-$\mathbf{U}$ being slightly more accurate than FFG-$\mathbf{U}$ as on the clean data. For ECE, FFG-$\mathbf{U}$ outperforms Ensemble-$\mathbf{U}$ and Ensemble-$\mathbf{W}$, which are similarly calibrated. Interestingly, while the accuracy of FFG-$\mathbf{W}$ decays quickly as the data is perturbed more strongly, its ECE remains roughly constant.

Table 3 further presents the utility of the maximum predicted probability for out-of-distribution (OOD) detection. The metrics are the area under the receiver operator characteristic (AUROC) and the precision-recall curve (AUPR). The inducing-weight methods perform similarly to Ensemble-$\mathbf{W}$; all three outperform FFG-$\mathbf{W}$ and deterministic networks across the board.

**Parameter pruning** We further investigate pruning as a pragmatic alternative for more parameter-efficient inference. For FFG-$\mathbf{U}$, we prune entries of the $Z$ matrices, which contribute the largest number of parameters to the inducing methods, with the smallest magnitude. For FFG-$\mathbf{W}$ we follow Graves (2011) in setting different fractions of $\mathbf{W}$ to 0 depending on their variational mean-to-variance ratio and repeat the previous experiments after fine-tuning the distributions on the remaining variables. We stress that, unlike FFG-$\mathbf{U}$, the FFG-$\mathbf{W}$ pruning corresponds to a post-hoc change of the probabilistic model and no longer performs inference in the original weight-space.

For FFG-$\mathbf{W}$, pruning 90% of the parameters (leaving 20% of parameters as compared to its deterministic counterpart) worsens the ECE, in particular on CIFAR100, see Fig. 6. Further pruning to 1%

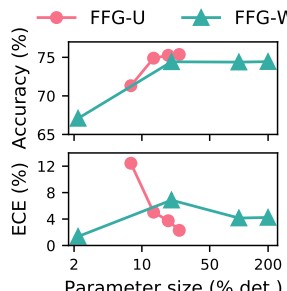

Figure 6: CIFAR100 pruning accuracy(↑) & ECE(↓). Rightmost points are w/out pruning.

worsens the accuracy and the OOD detection results as well. On the other hand, pruning $50\%$ of the $\boldsymbol{Z}$ matrices for FFG-U reduces the parameter count to $13.2\%$ of a deterministic net, at the cost of only slightly worse calibration. See Appendix K for the full results.

## 5 Discussions

### 5.1 A function-space perspective on inducing weights

Although the inducing weight approach performs approximate inference in weight space, we present in Appendix G a function-space inference perspective of the proposed method, showing its deep connections to sparse GPs. Our analysis considers the function-space behaviour of each network layer's output and discusses the corresponding interpretations of the $\mathbf{U}$ variables and $\boldsymbol{Z}$ parameters.

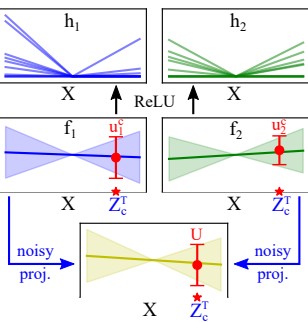

The interpretations are visualised in Fig. 7. Similar to sparse GPs, in each layer, the $\boldsymbol{Z}_c$ parameters can be viewed as the (transposed) inducing input locations which lie in the same space as the layer's input. The $\mathbf{U}_c$ variables can also be viewed as the corresponding (noisy) inducing outputs that lie in the pre-activation space. Given that the output dimension $d_{out}$ can still be high (e.g. $> 1000$ in a fully connected layer), our approach performs further dimension reduction in a similar spirit as probabilistic PCA (Tipping & Bishop, 1999), which projects the *column vectors* of $\mathbf{U}_c$ to a lower-dimensional space. This returns the inducing weight variables $\mathbf{U}$, and the projection parameters are $\{\boldsymbol{Z}_r, \boldsymbol{D}_r\}$. Combining the two steps, it means the column vectors of $\mathbf{U}$ can be viewed as collecting the "noisy projected inducing outputs" whose corresponding "inducing inputs" are row vectors of $\boldsymbol{Z}_c$ (see the red bars in Fig. 7).

Figure 7: Visualising $\mathbf{U}$ variables in pre-activation spaces.

In Appendix G we further derive the resulting variational objective from the function-space view, which is almost identical to Eq. (9), except for scaling coefficients on the $R(\lambda_l)$ terms to account for the change in dimensionality from weight space to function space. This result nicely connects posterior inference in weight- and function-space.

### 5.2 Related work

**Parameter-efficient uncertainty quantification methods** Recent research has proposed Gaussian posterior approximations for BNNs with efficient covariance structure (Ritter et al., 2018; Zhang et al., 2018b; Mishkin et al., 2018). The inducing weight approach differs from these in introducing structure via a hierarchical posterior with low-dimensional auxiliary variables. Another line of work reduces the memory overhead via efficient parameter sharing (Louizos & Welling, 2017; Wen et al., 2020; Świątkowski et al., 2020; Dusenberry et al., 2020). The third category of work considers a hybrid approach, where only a selective part of the neural network receives Bayesian treatments, and the other weights remain deterministic (Bradshaw et al., 2017; Daxberger et al., 2021). However, both types of approaches maintain a "mean parameter" for the weights, making the memory footprint at least that of storing a deterministic neural network. Instead, our approach shares parameters via the augmented prior with efficient low-rank structure, *reducing* the memory use compared to a deterministic network. In a similar spirit, Izmailov et al. (2019) perform inference in a $d$-dimensional sub-space obtained from PCA on weights collected from an SGD trajectory. But this approach does not leverage the layer-structure of neural networks and requires $d\times$ memory of a single network.

**Network pruning in uncertainty estimation context** There is a large amount of existing research advocating network pruning approaches for parameter-efficient deep learning, e.g. see Han et al. (2016); Frankle & Carbin (2018); Lee et al. (2019). In this regard, mean-field VI approaches have also shown success in network pruning, but only in terms of maintaining a minimum accuracy level (Graves, 2011; Louizos et al., 2017; Havasi et al., 2019). To the best of our knowledge, our empirical study presents the first evaluation for VI-based pruning methods in maintaining uncertainty estimation quality. Deng et al. (2019) considers pruning BNNs with stochastic gradient Langevin dynamics (Welling & Teh, 2011) as the inference engine. The inducing weight approach is orthogonal to these BNN pruning approaches, as it leaves the prior on the network parameters intact, while the pruning

approaches correspond to a post-hoc change of the probabilistic model to using a sparse weight prior. Indeed our parameter pruning experiments showed that our approach can be combined with network pruning to achieve further parameter efficiency improvements.

**Sparse GP and function-space inference**   As BNNs and GPs are closely related (Neal, 1995; Matthews et al., 2018; Lee et al., 2018), recent efforts have introduced GP-inspired techniques to BNNs (Ma et al., 2019; Sun et al., 2019; Khan et al., 2019; Ober & Aitchison, 2021). Compared to weight-space inference, function-space inference is appealing as its uncertainty is more directly relevant for predictive uncertainty estimation. While the inducing weight approach performs computations in weight-space, Section 5.1 establishes the connection to function-space posteriors. Our approach is related to sparse deep GP methods with $\mathbf{U}_c$ having similar interpretations as inducing outputs in e.g. Salimbeni & Deisenroth (2017). The major difference is that $\mathbf{U}$ lies in a low-dimensional space, projected from the pre-activation output space of a network layer.

The original Matheron's rule (Journel & Huijbregts, 1978; Hoffman & Ribak, 1991; Doucet, 2010) for sampling from conditional multivariate Gaussian distributions has recently been applied to speed-up sparse GP inference (Wilson et al., 2020, 2021). As explained in Section 3.2, direct application of the original rule to sampling $\mathbf{W}$ conditioned on $\mathbf{U}$ still incurs prohibitive cost as $p(\text{vec}(\mathbf{W}), \text{vec}(\mathbf{U}))$ does not have a convenient factorisation form. Our extended Matheron's rule addresses this issue by exploiting the efficient factorisation structure of the joint matrix normal distribution $p(\mathbf{W}, \mathbf{U}_c, \mathbf{U}_r, \mathbf{U})$, reducing the dominating factor of computation cost from cubic $(\mathcal{O}(d_{out}^3 d_{in}^3))$ to linear $(\mathcal{O}(d_{out} d_{in}))$. We expect this new rule to be useful for a wide range of models/applications beyond BNNs, such as matrix-variate Gaussian processes (Stegle et al., 2011).

**Priors on neural network weights**   Hierarchical priors for weights has also been explored (Louizos et al., 2017; Krueger et al., 2017; Atanov et al., 2019; Ghosh et al., 2019; Karaletsos & Bui, 2020). However, we emphasise that $\tilde{p}(\mathbf{W}, \mathbf{U})$ is a *pseudo prior* that is constructed to *assist posterior inference* rather than to *improve model design*. Indeed, parameters associated with the inducing weights are optimisable for improving posterior approximations. Our approach can be adapted to other priors, e.g. for a Horseshoe prior $p(\theta, \nu) = p(\theta|\nu)p(\nu) = \mathcal{N}(\theta; 0, \nu^2)C^+(\nu; 0, 1)$, the pseudo prior can be defined as $\tilde{p}(\theta, \nu, a) = \tilde{p}(\theta|\nu, a)\tilde{p}(a)p(\nu)$ such that $\int \tilde{p}(\theta|\nu, a)\tilde{p}(a)da = p(\theta|\nu)$. In general, pseudo priors have found broader success in Bayesian computation (Carlin & Chib, 1995).

# 6   Conclusion

We have proposed a parameter-efficient uncertainty quantification framework for neural networks. It augments each of the network layer weights with a small matrix of inducing weights, and by extending Matheron's rule to matrix-normal related distributions, maintains a relatively small run-time overhead compared with ensemble methods. Critically, experiments on prediction and uncertainty estimation tasks show the competence of the inducing weight methods to the state-of-the-art, while reducing the parameter count to under a quarter of a deterministic ResNet-50 before pruning. This represents a significant improvement over prior Bayesian and deep ensemble techniques, which so far have not managed to go below this threshold despite various attempts of matching it closely.

Several directions are to be explored in the future. First, modelling correlations across layers might further improve the inference quality. We outline an initial approach leveraging inducing variables in Appendix H. Second, based on the function-space interpretation of inducing weights, better initialisation techniques can be inspired from the sparse GP and dimension reduction literature. Similarly, this interpretation might suggest other innovative pruning approaches for the inducing weight method, thereby achieving further memory savings. Lastly, the run-time overhead of our approach can be mitigated by a better design of the inducing weight structure as well as vectorisation techniques amenable to parallelised computation. Designing hardware-specific implementations of the inducing weight approach is also a viable alternative for such purposes.

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
