## A  Notation

Generally, we use regular font letters $a$, $A$ to denote scalars, lower-case bold symbols $\mathbf{a}$ to denote vectors and upper-case bold symbols $\mathbf{A}$ for matrices.

Table 4: Overview of notation and matrix shapes

| | |
|---|---|
| **Random variables:** | |
| $\mathbf{W} : d_{out} \times d_{in}$ | Weight matrix of a layer |
| $\mathbf{U} : M_{out} \times M_{in}$ | Inducing weight matrix |
| $\mathbf{U}_r : M_{out} \times d_{in}$ | Inducing row matrix |
| $\mathbf{U}_c : d_{out} \times M_{in}$ | Inducing column matrix |
| | |
| **Parameters:** | |
| $\boldsymbol{Z}_r : M_{out} \times d_{out}, \boldsymbol{Z}_c : M_{in} \times d_{in}$ | Inducing covariance parameters |
| $\boldsymbol{D}_r : M_{out} \times M_{out}, \boldsymbol{D}_c : M_{in} \times M_{in}$ | Inducing precision parameters (we use diagonal matrices in our approach) |
| $\lambda^2$ | Conditional rescaling factor ($< 1$ reduces variance of weight distribution) |
| | |
| **Hyperparameters:** | |
| $\sigma^2$ | Prior variance |
| $\sigma^2_{max}$ | Maximum approximate posterior variance |
| $\lambda^2_{max}$ | Maximum conditional rescaling |
| | |
| **Other variables:** | |
| $\boldsymbol{\Sigma}_r : (d_{out} + M_{out}) \times (d_{out} + M_{out})$ | Joint row covariance of $\mathbf{W}$ and $\mathbf{U}$ |
| $\boldsymbol{\Sigma}_c : (d_{in} + M_{in}) \times (d_{in} + M_{in})$ | Joint column covariance of $\mathbf{W}$ and $\mathbf{U}$ |
| $\boldsymbol{L}_r, \boldsymbol{L}_c$ | Lower Cholesky factor of the joint row/column covariance |
| $\boldsymbol{\Psi}_r : M_{out} \times M_{out}, \boldsymbol{\Psi}_c : M_{in} \times M_{in}$ | Marginal row/column inducing covariance |

## B  Derivations of the auxiliary variational objective

When the variational distribution is constructed as a mixture distribution $q(\boldsymbol{\theta}) = \int q(\boldsymbol{\theta}|\mathbf{a})q(\mathbf{a})d\mathbf{a}$, the original variational lower-bound becomes intractable:

$$\log p(\mathcal{D}) \geq \mathcal{L}(q(\boldsymbol{\theta})) = \mathbb{E}_{q(\boldsymbol{\theta})}[\log p(\mathcal{D}|\boldsymbol{\theta})] + \mathbb{E}_{q(\boldsymbol{\theta})}\left[\log \frac{p(\boldsymbol{\theta})}{\int q(\boldsymbol{\theta}|\mathbf{a})q(\mathbf{a})d\mathbf{a}}\right]. \quad (13)$$

However, notice that we can also rewrite the "marginal" $q(\boldsymbol{\theta})$ using Bayes' rule: $q(\boldsymbol{\theta}) = \frac{q(\boldsymbol{\theta}|\mathbf{a})q(\mathbf{a})}{q(\mathbf{a}|\boldsymbol{\theta})}$, meaning that the variational lower-bound can be re-formulated as

$$\mathcal{L}(q(\boldsymbol{\theta})) = \mathbb{E}_{q(\boldsymbol{\theta},\mathbf{a})}[\log p(\mathcal{D}|\boldsymbol{\theta})] + \mathbb{E}_{q(\boldsymbol{\theta},\mathbf{a})}\left[\log \frac{p(\boldsymbol{\theta})q(\mathbf{a}|\boldsymbol{\theta})}{q(\boldsymbol{\theta}|\mathbf{a})q(\mathbf{a})}\right]. \quad (14)$$

For many flexible mixture distributions, $q(\mathbf{a}|\boldsymbol{\theta})$ remains intractable. Fortunately, notice that for *any* distribution $r(\mathbf{a}|\boldsymbol{\theta})$ with the same support as $q(\mathbf{a}|\boldsymbol{\theta})$, we have:

$$\mathbb{E}_{q(\boldsymbol{\theta},\mathbf{a})}\left[\log \frac{p(\boldsymbol{\theta})q(\mathbf{a}|\boldsymbol{\theta})}{q(\boldsymbol{\theta}|\mathbf{a})q(\mathbf{a})}\right] = \mathbb{E}_{q(\boldsymbol{\theta},\mathbf{a})}\left[\log \frac{p(\boldsymbol{\theta})r(\mathbf{a}|\boldsymbol{\theta})}{q(\boldsymbol{\theta}|\mathbf{a})q(\mathbf{a})}\right] + \mathbb{E}_{q(\boldsymbol{\theta},\mathbf{a})}\left[\log \frac{q(\mathbf{a}|\boldsymbol{\theta})}{r(\mathbf{a}|\boldsymbol{\theta})}\right], \quad (15)$$

and more importantly, the second term on the RHS of the above equation satisfies

$$\mathbb{E}_{q(\boldsymbol{\theta},\mathbf{a})}\left[\log \frac{q(\mathbf{a}|\boldsymbol{\theta})}{r(\mathbf{a}|\boldsymbol{\theta})}\right] = \mathbb{E}_{q(\boldsymbol{\theta})}[\mathbb{KL}[q(\mathbf{a}|\boldsymbol{\theta})||r(\mathbf{a}|\boldsymbol{\theta})]] \geq 0. \quad (16)$$

This means we can remove this KL term and construct a lower-bound to the variational lower-bound:

$$\log p(\mathcal{D}) \geq \mathcal{L}(q(\boldsymbol{\theta})) \geq \mathbb{E}_{q(\boldsymbol{\theta},\mathbf{a})}[\log p(\mathcal{D}|\boldsymbol{\theta})] + \mathbb{E}_{q(\boldsymbol{\theta},\mathbf{a})}\left[\log \frac{p(\boldsymbol{\theta})r(\mathbf{a}|\boldsymbol{\theta})}{q(\boldsymbol{\theta}|\mathbf{a})q(\mathbf{a})}\right] := \mathcal{L}(q(\boldsymbol{\theta},\mathbf{a})), \quad (17)$$

which corresponds to the auxiliary variational lower-bound Eq. (2) presented in the main text. This auxiliary bound can be improved by optimising $r(\mathbf{a}|\boldsymbol{\theta})$ towards better approximating $q(\mathbf{a}|\boldsymbol{\theta})$, and it recovers the original variational lower-bound iff. $r(\mathbf{a}|\boldsymbol{\theta}) = q(\mathbf{a}|\boldsymbol{\theta})$. Still we emphasise that the auxiliary bound is valid for *any* $r(\mathbf{a}|\boldsymbol{\theta})$ with the same support as $q(\mathbf{a}|\boldsymbol{\theta})$, which enables our design presented in the main text to improve memory efficiency.

## C   An introduction to SVGP

This section provides a brief introduction to sparse variational approximation for variationally sparse GP (SVGP). We use regression as a running example, but the principles of SVGP also apply to other supervised learning tasks such as classification. Readers are also referred to e.g. Leibfried et al. (2020) for a modern tutorial.

Assume we have a regression dataset $\mathcal{D} = \{\mathbf{X}, \mathbf{Y}\}$ where $\mathbf{X} = [\boldsymbol{x}_1, ..., \boldsymbol{x}_N]$ and $\mathbf{Y} = [\boldsymbol{y}_1, ..., \boldsymbol{y}_N]$. In GP regression we build the following probabilistic model to address this regression task:

$$\boldsymbol{y}_n = f(\boldsymbol{x}_n) + \boldsymbol{\epsilon}_n, \quad \boldsymbol{\epsilon}_n \sim \mathcal{N}(\mathbf{0}, \sigma^2 \mathbf{I}), \quad f(\cdot) \sim \mathcal{GP}(0, \mathcal{K}(\cdot, \cdot)). \tag{18}$$

Here we put on the regression function $f(\cdot)$ a GP prior with zero mean function and covariance function defined by kernel $\mathcal{K}(\cdot, \cdot)$. In practice we can only evaluate the function $f(\cdot)$ on finite number of inputs, but fortunately by construction, GP allows sampling of function values from a joint Gaussian distribution. In detail, we can sample from the following Gaussian distribution to get the function value samples $\mathbf{f} = [f(\boldsymbol{x}_1), ..., f(\boldsymbol{x}_N)]$ given the input locations $\mathbf{X}$:

$$\mathbf{f} \sim \mathcal{N}(\mathbf{0}, \mathbf{K_{XX}}), \quad \mathbf{K_{XX}}(m, n) = \mathcal{K}(\boldsymbol{x}_m, \boldsymbol{x}_n). \tag{19}$$

Therefore we can rewrite the probabilistic model in finite dimension as

$$p(\mathbf{Y}|\mathbf{X}, f(\cdot)) = p(\mathbf{Y}|\mathbf{f}) = \prod_{n=1}^{N} \mathcal{N}(\boldsymbol{y}_n; f(\boldsymbol{x}_n), \sigma^2 \mathbf{I}), \quad p(\mathbf{f}|\mathbf{X}) = \mathcal{N}(\mathbf{f}; \mathbf{0}, \mathbf{K_{XX}}). \tag{20}$$

Note that the GP prior can be extended to a larger set of inputs $\mathbf{X} \cup \mathbf{X}^*$ as $p(\mathbf{f}, \mathbf{f}^*|\mathbf{X}, \mathbf{X}^*) = \mathcal{N}([\mathbf{f}, \mathbf{f}^*]; \mathbf{0}, \mathbf{K}_{[\mathbf{X}, \mathbf{X}^*], [\mathbf{X}, \mathbf{X}^*]})$, where $\mathbf{f}^*$ denotes the function values given the inputs $\mathbf{X}^*$, and

$$\mathbf{K}_{[\mathbf{X}, \mathbf{X}^*], [\mathbf{X}, \mathbf{X}^*]} = \begin{pmatrix} \mathbf{K_{XX}} & \mathbf{K_{XX^*}} \\ \mathbf{K_{X^*X}} & \mathbf{K_{X^*X^*}} \end{pmatrix}.$$

Importantly, this definition leaves the marginal prior unchanged ($\int p(\mathbf{f}, \mathbf{f}^*|\mathbf{X}, \mathbf{X}^*)d\mathbf{f}^* = p(\mathbf{f}|\mathbf{X})$), and the conditional distribution $p(\mathbf{f}|\mathbf{f}^*, \mathbf{X}, \mathbf{X}^*)$ is also Gaussian. This means predictive inference can be done in the following way: given test inputs $\mathbf{X}^*$, the posterior predictive for $\mathbf{f}^*$ is

$$p(\mathbf{Y}|\mathbf{X}, \mathbf{X}^*) = \int p(\mathbf{Y}|\mathbf{f})p(\mathbf{f}|\mathbf{X})p(\mathbf{f}^*|\mathbf{f}, \mathbf{X}, \mathbf{X}^*)d\mathbf{f}d\mathbf{f}^* = \int p(\mathbf{Y}|\mathbf{f})p(\mathbf{f}|\mathbf{X})d\mathbf{f} = p(\mathbf{Y}|\mathbf{X}), \tag{21}$$

$$\begin{aligned} p(\mathbf{f}^*|\mathbf{X}, \mathbf{Y}, \mathbf{X}^*) &= \int \frac{p(\mathbf{Y}|\mathbf{f})p(\mathbf{f}, \mathbf{f}^*|\mathbf{X}, \mathbf{X}^*)}{p(\mathbf{Y}|\mathbf{X}, \mathbf{X}^*)} d\mathbf{f} \\ &= \int \frac{p(\mathbf{Y}|\mathbf{f})p(\mathbf{f}|\mathbf{X})p(\mathbf{f}^*|\mathbf{f}, \mathbf{X}, \mathbf{X}^*)}{p(\mathbf{Y}|\mathbf{X})} d\mathbf{f} \\ &= \int \underbrace{p(\mathbf{f}|\mathbf{X}, \mathbf{Y})p(\mathbf{f}^*|\mathbf{f}, \mathbf{X}, \mathbf{X}^*)}_{=p(\mathbf{f}, \mathbf{f}^*|\mathbf{X}, \mathbf{Y}, \mathbf{X}^*)} d\mathbf{f}. \end{aligned} \tag{22}$$

Unfortunately the exact posterior $p(\mathbf{f}|\mathbf{X}, \mathbf{Y})$ is intractable even for GP regression. Although in such case $p(\mathbf{f}|\mathbf{X}, \mathbf{Y})$ is Gaussian, evaluating this posterior requires inverting/decomposing an $N \times N$ covariance matrix which has time complexity $\mathcal{O}(N^3)$. Also the storage cost of $\mathcal{O}(N^2)$ for the posterior covariance is prohibitively expensive when $N$ is large.

To address the intractability issue, we seek to define an *approximate posterior* $q(f(\cdot))$, so that we can evaluate it on $\mathbf{X} \cup \mathbf{X}^*$ and approximate the posterior as $p(\mathbf{f}, \mathbf{f}^*|\mathbf{X}, \mathbf{Y}, \mathbf{X}^*) \approx q(\mathbf{f}, \mathbf{f}^*)$. SVGP (Snelson & Ghahramani, 2006; Titsias, 2009) defines such approximate posterior by introducing *inducing* inputs and outputs. Again this is done by noticing that we can extend the GP prior to an

even larger set of inputs $\mathbf{X} \cup \mathbf{X}^* \cup \mathbf{Z}$, where $\mathbf{Z} = [\mathbf{z}_1, ..., \mathbf{z}_M]$ are called *inducing inputs*, and the corresponding function values $\mathbf{u} = [f(\mathbf{z}_1), ..., f(\mathbf{z}_M)]$ are named *inducing outputs*:

$$p(\mathbf{f}, \mathbf{f}^*, \mathbf{u} | \mathbf{X}, \mathbf{X}^*, \mathbf{Z}) = \mathcal{N}([\mathbf{f}, \mathbf{f}^*, \mathbf{u}]; \mathbf{0}, \mathbf{K}_{[\mathbf{X}, \mathbf{X}^*, \mathbf{Z}], [\mathbf{X}, \mathbf{X}^*, \mathbf{Z}]})$$
$$= p(\mathbf{u} | \mathbf{Z}) p(\mathbf{f}, \mathbf{f}^* | \mathbf{u}, \mathbf{X}, \mathbf{X}^*, \mathbf{Z}). \tag{23}$$

Importantly, marginal consistency still holds for any $\mathbf{Z}$: $\int p(\mathbf{f}, \mathbf{f}^*, \mathbf{u} | \mathbf{X}, \mathbf{X}^*, \mathbf{Z}) d\mathbf{u} = p(\mathbf{f}, \mathbf{f}^* | \mathbf{X}, \mathbf{X}^*)$ (c.f. eq.(3) in the main text). Furthermore, the conditional distribution $p(\mathbf{f}, \mathbf{f}^* | \mathbf{u}, \mathbf{X}, \mathbf{X}^*, \mathbf{Z})$ is a Gaussian distribution. Observing these, the SVGP approach defines the approximate posterior as

$$q(\mathbf{f}, \mathbf{f}^*, \mathbf{u}) = p(\mathbf{f}, \mathbf{f}^* | \mathbf{X}, \mathbf{X}^*, \mathbf{Z}^*) q(\mathbf{u}), \quad q(\mathbf{f}, \mathbf{f}^*) = \int q(\mathbf{f}, \mathbf{f}^*, \mathbf{u}) d\mathbf{u}, \tag{24}$$

and minimises an upper-bound of the KL divergence to find the optimal $q(\mathbf{f}, \mathbf{f}^*)$:

$$\mathbb{KL}[q(\mathbf{f}, \mathbf{f}^*) || p(\mathbf{f}, \mathbf{f}^* | \mathbf{X}, \mathbf{Y}, \mathbf{X}^*)] = \mathbb{E}_{q(\mathbf{f}, \mathbf{f}^*)} \left[ \log \frac{q(\mathbf{f}, \mathbf{f}^*) p(\mathbf{Y} | \mathbf{X})}{p(\mathbf{f}, \mathbf{f}^*, \mathbf{Y} | \mathbf{X}, \mathbf{X}^*)} \right]$$

$$= \log p(\mathbf{Y} | \mathbf{X}) + \mathbb{E}_{q(\mathbf{f}, \mathbf{f}^*, \mathbf{u})} \left[ \log \frac{q(\mathbf{f}, \mathbf{f}^*, \mathbf{u})}{p(\mathbf{Y} | \mathbf{f}) p(\mathbf{f}, \mathbf{f}^*, \mathbf{u} | \mathbf{X}, \mathbf{X}^*, \mathbf{Z})} + \frac{p(\mathbf{u} | \mathbf{f}, \mathbf{f}^*, \mathbf{X}, \mathbf{X}^*, \mathbf{Z})}{q(\mathbf{u} | \mathbf{f}, \mathbf{f}^*)} \right]$$

$$= \log p(\mathbf{Y} | \mathbf{X}) + \mathbb{E}_{q(\mathbf{f}, \mathbf{f}^*, \mathbf{u})} \left[ \log \frac{\cancel{p(\mathbf{f}, \mathbf{f}^* | \mathbf{u}, \mathbf{X}, \mathbf{X}^*, \mathbf{Z})} q(\mathbf{u})}{p(\mathbf{Y} | \mathbf{f}) \cancel{p(\mathbf{f}, \mathbf{f}^* | \mathbf{u}, \mathbf{X}, \mathbf{X}^*, \mathbf{Z})} p(\mathbf{u} | \mathbf{Z})} \right]$$

$$\quad - \mathbb{E}_{q(\mathbf{f}, \mathbf{f}^*)} \mathbb{KL}[q(\mathbf{u} | \mathbf{f}, \mathbf{f}^*) || p(\mathbf{u} | \mathbf{f}, \mathbf{f}^*, \mathbf{X}, \mathbf{X}^*, \mathbf{Z})]$$

$$\leq \log p(\mathbf{Y} | \mathbf{X}) - \underbrace{\{ \mathbb{E}_{q(\mathbf{f})}[\log p(\mathbf{Y} | \mathbf{f})] - \mathbb{KL}[q(\mathbf{u}) || p(\mathbf{u} | \mathbf{Z})] \}}_{:= \mathcal{L}(q(\mathbf{f}, \mathbf{u})}.$$

$$\tag{25}$$

As KL divergences are non-negative, the above derivations also means $\log p(\mathbf{Y} | \mathbf{X}) \geq \mathcal{L}(q(\mathbf{f}, \mathbf{u}))$, and we can optimise the parameters in $q$ to tighten the lower-bound. These variational parameters include the distributional parameters of $q(\mathbf{u})$ (e.g. mean and covariance if $q(\mathbf{u})$ is Gaussian), as well as the inducing inputs $\mathbf{Z}$, since the variational lower-bound $\mathcal{L}(q(\mathbf{f}, \mathbf{u}))$ is valid for any settings of $\mathbf{Z}$. The lower-bound requires evaluating

$$q(\mathbf{f}) = \int p(\mathbf{f}, \mathbf{f}^* | \mathbf{u}, \mathbf{X}, \mathbf{X}^*, \mathbf{Z}) q(\mathbf{u}) d\mathbf{f}^* d\mathbf{u} = \int p(\mathbf{f} | \mathbf{u}, \mathbf{X}, \mathbf{Z}) q(\mathbf{u}) d\mathbf{u},$$

which can be done efficiently when $M << N$, as evaluating the conditional Gaussian $p(\mathbf{f} | \mathbf{u}, \mathbf{X}, \mathbf{Z})$ has $\mathcal{O}(NM^2 + M^3)$ run-time cost. Similarly, once $q$ is optimised, in prediction time one can directly sample from $q(\mathbf{f}^*)$ by computing $q(\mathbf{f}^*) = \int p(\mathbf{f}^* | \mathbf{u}, \mathbf{X}^*, \mathbf{Z}) q(\mathbf{u}) d\mathbf{u}$, and by caching the inverse/decomposition of both the covariacne matrix of $q(\mathbf{u})$ and $\mathbf{K}_{\mathbf{ZZ}}$, predictive inference can be approximated efficiently.

Using shorthand notations by dropping $\mathbf{Z}$, e.g. $p(\mathbf{u}) = p(\mathbf{u} | \mathbf{Z})$ and $p(\mathbf{f} | \mathbf{X}, \mathbf{u}) = p(\mathbf{f} | \mathbf{u}, \mathbf{X}, \mathbf{Z})$, returns the desired results discussed in Section 2 of the main text, if we set $a = \mathbf{u}$ and $\theta = \mathbf{f}$. In fact from the above discussions, we see that $\theta$ in GP inference is infinite dimensional: $\theta = f_{\neq \mathbf{u}}$, as we can extend the finite collection of function values to include both $\mathbf{f}$ and $\mathbf{f}^*$ for any $\mathbf{X}^* \neq \mathbf{Z}$. By tying the conditional distribution given $\mathbf{u}$ in both the GP prior and the approximate posterior $q(f(\cdot))$, the posterior belief updates are "compressed" into $\mathbf{u}$ space, which is also reflected by the name "sparse approximation" of the approach.

# D   Derivations of the augmented (pseudo) prior

## D.1   Inducing auxiliary variables: multivariate Gaussian case

Suppose each weight matrix has an isotropic Gaussian prior with zero mean, i.e. $\text{vec}(\mathbf{W}) \sim \mathcal{N}(0, \sigma^2 \mathbf{I})$ where vec concatenates the columns of a matrix into a vector and $\sigma$ is the standard deviation. Augmenting this Gaussian with an auxiliary variable $\mathbf{U}$ that also has a mean of zero and some covariance that we are free to parameterise, the joint distribution is

$$\begin{pmatrix} \text{vec}(\mathbf{W}) \\ \text{vec}(\mathbf{U}) \end{pmatrix} \sim \mathcal{N}(0, \boldsymbol{\Sigma}) \quad \text{with} \quad L = \begin{pmatrix} \sigma \mathbf{I} & 0 \\ \mathbf{Z} & \mathbf{D} \end{pmatrix}$$

$$\text{s.t.} \quad \boldsymbol{\Sigma} = \boldsymbol{L} \boldsymbol{L}^\top = \begin{pmatrix} \sigma^2 \mathbf{I} & \sigma \mathbf{Z}^\top \\ \sigma \mathbf{Z} & \mathbf{Z} \mathbf{Z}^\top + \mathbf{D}^2 \end{pmatrix}$$

where $D$ is a positive diagonal matrix and $Z$ a matrix with arbitrary entries. Through defining the Cholesky decomposition of $\Sigma$ we ensure its positive definiteness. By the usual rules of Gaussian marginalisation, the augmented model leaves the marginal prior on $\mathbf{W}$ unchanged. Further, we can analytically derive the conditional distribution on the weights given the inducing weights:

$$p(\operatorname{vec}(\mathbf{W})|\operatorname{vec}(\mathbf{U})) = \mathcal{N}(\boldsymbol{\mu}_{\mathbf{W}|\mathbf{U}}, \boldsymbol{\Sigma}_{\mathbf{W}|\mathbf{U}}), \tag{26}$$

$$\boldsymbol{\mu}_{\mathbf{W}|\mathbf{U}} = \sigma \boldsymbol{Z}^\top \boldsymbol{\Psi}^{-1} \operatorname{vec}(U), \tag{27}$$

$$\boldsymbol{\Sigma}_{\mathbf{W}|\mathbf{U}} = \sigma^2 (\boldsymbol{I} - \boldsymbol{Z}^\top \boldsymbol{\Psi}^{-1} \boldsymbol{Z}), \tag{28}$$

$$\boldsymbol{\Psi} = \boldsymbol{Z}\boldsymbol{Z}^\top + \boldsymbol{D}^2.$$

For inference, we now need to define an approximate posterior over the joint space $q(\mathbf{W}, \mathbf{U})$. We will do so by factorising it as $q(\mathbf{W}, \mathbf{U}) = q(\mathbf{W}|\mathbf{U})q(\mathbf{U})$. Factorising the prior in the same way leads to the following KL term in the ELBO:

$$\mathbb{KL}\left[q(\mathbf{W}, \mathbf{U})||p(\mathbf{W}, \mathbf{U})\right] = \mathbb{E}_{q(\mathbf{U})}\left[\mathbb{KL}\left[q(\mathbf{W}|\mathbf{U})||p(\mathbf{W}|\mathbf{U})\right] + \mathbb{KL}\left[q(\mathbf{U})||p(\mathbf{U})\right] \tag{29}$$

### D.2 Inducing auxiliary variables: matrix normal case

Now we introduce the inducing variables in matrix space, and, in addition to the inducing weight $\mathbf{U}$, we pad in two inducing matrices $\mathbf{U}_r$, $\mathbf{U}_c$, such that the full augmented prior is:

$$\begin{pmatrix} \mathbf{W} & \mathbf{U}_c \\ \mathbf{U}_r & \mathbf{U} \end{pmatrix} \sim p(\mathbf{W}, \mathbf{U}_c, \mathbf{U}_r, \mathbf{U}) := \mathcal{MN}(0, \boldsymbol{\Sigma}_r, \boldsymbol{\Sigma}_c), \tag{30}$$

$$\text{with} \quad \boldsymbol{L}_r = \begin{pmatrix} \sigma_r \boldsymbol{I} & 0 \\ \boldsymbol{Z}_r & \boldsymbol{D}_r \end{pmatrix} \quad \text{and} \quad \boldsymbol{L}_c = \begin{pmatrix} \sigma_c \boldsymbol{I} & 0 \\ \boldsymbol{Z}_c & \boldsymbol{D}_c \end{pmatrix}$$

$$\text{s.t.} \quad \boldsymbol{\Sigma}_r = \boldsymbol{L}_r \boldsymbol{L}_r^\top = \begin{pmatrix} \sigma_r^2 \boldsymbol{I} & \sigma_r \boldsymbol{Z}_r^\top \\ \sigma_r \boldsymbol{Z}_r & \boldsymbol{Z}_r \boldsymbol{Z}_r^\top + \boldsymbol{D}_r^2 \end{pmatrix},$$

$$\text{and} \quad \boldsymbol{\Sigma}_c = \boldsymbol{L}_c \boldsymbol{L}_c^\top = \begin{pmatrix} \sigma_c^2 \boldsymbol{I} & \sigma_c \boldsymbol{Z}_c^\top \\ \sigma_c \boldsymbol{Z}_c & \boldsymbol{Z}_c \boldsymbol{Z}_c^\top + \boldsymbol{D}_c^2 \end{pmatrix}.$$

Matrix normal distributions have similar marginalisation and conditioning properties as multivariate Gaussians. As such, the marginal both over some set of rows and some set of columns is still a matrix normal. Hence, $p(\mathbf{W}) = \mathcal{MN}(0, \sigma_r^2 \boldsymbol{I}, \sigma_c^2 \boldsymbol{I})$, and by choosing $\sigma_r \sigma_c = \sigma$ this matrix normal distribution is equivalent to the multivariate normal $p(\operatorname{vec}(\mathbf{W})) = \mathcal{N}(0, \sigma^2 \boldsymbol{I})$. Also $p(\mathbf{U}) = \mathcal{MN}(0, \boldsymbol{\Psi}_r, \boldsymbol{\Psi}_c)$, where again $\boldsymbol{\Psi}_r = \boldsymbol{Z}_r \boldsymbol{Z}_r^\top + \boldsymbol{D}_r^2$ and $\boldsymbol{\Psi}_c = \boldsymbol{Z}_c \boldsymbol{Z}_c^\top + \boldsymbol{D}_c^2$. Similarly, the conditionals on some rows or columns are matrix normal distributed:

$$\mathbf{U}_c|\mathbf{U} \sim \mathcal{MN}(\sigma_r \boldsymbol{Z}_r^\top \boldsymbol{\Psi}_r^{-1} \mathbf{U}, \sigma_r^2 (\boldsymbol{I} - \boldsymbol{Z}_r^\top \boldsymbol{\Psi}_r^{-1} \boldsymbol{Z}_r), \boldsymbol{\Psi}_c),$$

$$\mathbf{U}_r|\mathbf{U} \sim \mathcal{MN}(\mathbf{U}\boldsymbol{\Psi}_c^{-1}\sigma_c \boldsymbol{Z}_c, \boldsymbol{\Psi}_r, \sigma_c^2 (\boldsymbol{I} - \boldsymbol{Z}_c^\top \boldsymbol{\Psi}_c^{-1} \boldsymbol{Z}_c)),$$

$$\mathbf{W}|\mathbf{U}_c \sim \mathcal{MN}\left(\mathbf{U}_c \boldsymbol{\Psi}_c^{-1} \sigma_c \boldsymbol{Z}_c, \sigma_r^2 \boldsymbol{I}, \sigma_c^2 (\boldsymbol{I} - \boldsymbol{Z}_c^\top \boldsymbol{\Psi}_c^{-1} \boldsymbol{Z}_c)\right),$$

$$\mathbf{W}, \mathbf{U}_r|\mathbf{U}_c, \mathbf{U} \sim \mathcal{MN}(\begin{pmatrix} \mathbf{U}_c \\ \mathbf{U} \end{pmatrix} \boldsymbol{\Psi}_c^{-1} \sigma_c \boldsymbol{Z}_c, \boldsymbol{\Sigma}_r, \sigma_c^2 (\boldsymbol{I} - \boldsymbol{Z}_c^\top \boldsymbol{\Psi}_c^{-1} \boldsymbol{Z}_c)), \tag{31}$$

$$\mathbf{W}|\mathbf{U}_r, \mathbf{U}_c, \mathbf{U} \sim \mathcal{MN}(M_{\mathbf{W}}, \sigma_r^2 (\boldsymbol{I} - \boldsymbol{Z}_r^\top \boldsymbol{\Psi}_r^{-1} \boldsymbol{Z}_r), \sigma_c^2 (\boldsymbol{I} - \boldsymbol{Z}_c^\top \boldsymbol{\Psi}_c^{-1} \boldsymbol{Z}_c)),$$

$$M_{\mathbf{W}} = \sigma(\boldsymbol{Z}_r^\top \boldsymbol{\Psi}_r^{-1} \mathbf{U}_r + \mathbf{U}_c \boldsymbol{\Psi}_c^{-1} \boldsymbol{Z}_c - \boldsymbol{Z}_r^\top \boldsymbol{\Psi}_r^{-1} \mathbf{U} \boldsymbol{\Psi}_c^{-1} \boldsymbol{Z}_c).$$

## E   KL divergence for rescaled conditional weight distributions

For the conditional distribution on the weights, in the simplest case we set $q(\mathbf{W}|\mathbf{U}) = p(\mathbf{W}|\mathbf{U})$, hence the KL divergence would be zero. For the most general case of arbitrary Gaussian distributions with $q = \mathcal{N}(\boldsymbol{\mu}_q, \boldsymbol{\Sigma}_q)$ and $p = \mathcal{N}(\boldsymbol{\mu}_p, \boldsymbol{\Sigma}_p)$, the KL divergence is:

$$\mathbb{KL}\left[q||p\right] = \frac{1}{2}(\log \frac{\det \boldsymbol{\Sigma}_p}{\det \boldsymbol{\Sigma}_q} - d + \operatorname{tr}(\boldsymbol{\Sigma}_p^{-1} \boldsymbol{\Sigma}_q) + (\boldsymbol{\mu}_p - \boldsymbol{\mu}_q)^\top \boldsymbol{\Sigma}_p^{-1} (\boldsymbol{\mu}_p - \boldsymbol{\mu}_q)), \tag{32}$$

where $d$ is the number of elements of $\boldsymbol{\mu}$. As motivated, it is desirable to make $q(\mathbf{W}|\mathbf{U})$ similar to $p(\mathbf{W}|\mathbf{U})$. We consider a scalar rescaling of the covariance, i.e. for $p = \mathcal{N}(\boldsymbol{\mu}, \boldsymbol{\Sigma})$ we set

$q = \mathcal{N}(\boldsymbol{\mu}, \lambda^2 \boldsymbol{\Sigma})$. This leads to the final term, which is the Mahalanobis distance between the means under $p$, cancelling out entirely and the log determinant and trace terms becoming a function of $\lambda$ only: with $d = \dim(\text{vec}(\mathbf{W}))$,

$$
\begin{aligned}
\mathbb{KL}\left[q||p\right] &= \frac{1}{2}(\log \frac{\det \boldsymbol{\Sigma}}{\det \lambda^2 \boldsymbol{\Sigma}} - d + \text{tr}(\boldsymbol{\Sigma}^{-1}\lambda^2\boldsymbol{\Sigma})) \\
&= \frac{1}{2}(\log \frac{\det \boldsymbol{\Sigma}}{\lambda^{2d}\det\boldsymbol{\Sigma}} - d + \text{tr}(\lambda^2\boldsymbol{I})) \\
&= \frac{1}{2}(-2d\log\lambda - d + d\lambda^2) \\
&= d(\frac{1}{2}\lambda^2 - \log\lambda - \frac{1}{2}).
\end{aligned}
$$

# F    The extended Matheron's rule to matrix normal distributions

The original Matheron's rule (Journel & Huijbregts, 1978; Hoffman & Ribak, 1991; Doucet, 2010) for sampling conditional Gaussian variables states the following. If the joint multivariate Gaussian distribution is

$$
\begin{pmatrix} \text{vec}(\mathbf{W}) \\ \text{vec}(\mathbf{U}) \end{pmatrix} \sim p(\text{vec}(\mathbf{W}), \text{vec}(\mathbf{U})) := \mathcal{N}(\mathbf{0}, \boldsymbol{\Sigma}),
$$

$$
\boldsymbol{\Sigma} = \begin{pmatrix} \boldsymbol{\Sigma_{WW}} & \boldsymbol{\Sigma_{WU}} \\ \boldsymbol{\Sigma_{UW}} & \boldsymbol{\Sigma_{UU}} \end{pmatrix},
$$

then, conditioned on $\mathbf{U}$, sampling $\mathbf{W} \sim p(\text{vec}(\mathbf{W}), \text{vec}(\mathbf{U}))$ can be done as

$$
\text{vec}(\mathbf{W}) = \text{vec}(\bar{\mathbf{W}}) + \boldsymbol{\Sigma_{WU}}\boldsymbol{\Sigma_{UU}^{-1}}(\text{vec}(\mathbf{U}) - \text{vec}(\bar{\mathbf{U}})),
$$

$$
\text{vec}(\bar{\mathbf{W}}), \text{vec}(\bar{\mathbf{U}}) \sim \mathcal{N}(\mathbf{0}, \boldsymbol{\Sigma}).
$$

Matheron's rule can provide significant speed-ups if $\text{vec}(\mathbf{U})$ has significantly smaller dimensions than that of $\text{vec}(\mathbf{W})$, and the Cholesky decomposition of $\boldsymbol{\Sigma}$ can be computed with low costs (e.g. due to the specific structure in $\boldsymbol{\Sigma}$). Recall from the main text that the augmented prior is

$$
p(\text{vec}(\mathbf{W}), \text{vec}(\mathbf{U})) = \mathcal{N}\left(0, \begin{pmatrix} \sigma_c^2\boldsymbol{I} \otimes \sigma_r^2\boldsymbol{I} & \sigma_c\boldsymbol{Z}_c^\top \otimes \sigma_r\boldsymbol{Z}_r^\top \\ \sigma_c\boldsymbol{Z}_c \otimes \sigma_r\boldsymbol{Z}_r & \boldsymbol{\Psi}_c \otimes \boldsymbol{\Psi}_r \end{pmatrix}\right), \tag{33}
$$

and the corresponding conditional distribution is:

$$
p(\text{vec}(\mathbf{W})|\text{vec}(\mathbf{U})) = \mathcal{N}(\sigma_c\sigma_r\text{vec}(\boldsymbol{Z}_r\boldsymbol{\Psi}_r^{-1}\mathbf{U}\boldsymbol{\Psi}_c^{-1}\boldsymbol{Z}_c^\top), \sigma_c^2\sigma_r^2(\boldsymbol{I} - \boldsymbol{Z}_c^\top\boldsymbol{\Psi}_c^{-1}\boldsymbol{Z}_c \otimes \boldsymbol{Z}_r^\top\boldsymbol{\Psi}_r^{-1}\boldsymbol{Z}_r)). \tag{34}
$$

Therefore, while $\dim(\text{vec}(\mathbf{U}))$ is indeed significantly smaller than of $\dim(\text{vec}(\mathbf{W}))$ by construction, the joint covariance matrix does not support fast Cholesky decompositions, meaning that Matheron's rule for efficient sampling does not directly apply here.

However, in the full augmented space, the joint distribution does have an efficient matrix normal form: $p(\mathbf{W}, \mathbf{U}_c, \mathbf{U}_r, \mathbf{U}_c) = \mathcal{MN}(0, \boldsymbol{\Sigma}_r, \boldsymbol{\Sigma}_c)$. Furthermore, the row and column covariance matrices $\boldsymbol{\Sigma}_r$ and $\boldsymbol{\Sigma}_c$ are parameterised by their Cholesky decompositions, meaning that sampling from the joint distribution $p(\mathbf{W}, \mathbf{U}_c, \mathbf{U}_r, \mathbf{U})$ can be done in a fast way. Importantly, Cholesky decompositions for $p(\mathbf{U})$'s row and column covariance matrices $\boldsymbol{\Psi}_r$ and $\boldsymbol{\Psi}_c$ can be computed in $\mathcal{O}(M_{out}^3)$ and $\mathcal{O}(M_{in}^3)$ time, respectively, which are much faster than the multi-variate Gaussian case that requires $\mathcal{O}(M_{in}^3 M_{out}^3)$ time. Observing these, we extend Matheron's rule to sample $p(\mathbf{W}|\mathbf{U})$ where $p(\mathbf{W}, \mathbf{U})$ is the marginal distribution of $p(\mathbf{W}, \mathbf{U}_c, \mathbf{U}_r, \mathbf{U}_c) = \mathcal{MN}(0, \boldsymbol{\Sigma}_r, \boldsymbol{\Sigma}_c)$.

In detail, for drawing a sample from $p(\mathbf{W}|\mathbf{U})$ we need to draw a sample from the joint $p(\mathbf{W}, \mathbf{U})$. To do so, we sample from the augmented prior $\bar{\mathbf{W}}, \bar{\mathbf{U}}_c, \bar{\mathbf{U}}_r, \bar{\mathbf{U}} \sim p(\bar{\mathbf{W}}, \bar{\mathbf{U}}_c, \bar{\mathbf{U}}_r, \bar{\mathbf{U}}) = \mathcal{MN}(0, \boldsymbol{\Sigma}_r, \boldsymbol{\Sigma}_c)$, computed using the Cholesky decompositions of $\boldsymbol{\Sigma}_r$ and $\boldsymbol{\Sigma}_c$:

$$
\begin{pmatrix} \bar{\mathbf{W}} & \bar{\mathbf{U}}_c \\ \bar{\mathbf{U}}_r & \bar{\mathbf{U}} \end{pmatrix} = \begin{pmatrix} \sigma_r\boldsymbol{I} & 0 \\ \boldsymbol{Z}_r & \boldsymbol{D}_r \end{pmatrix} \begin{pmatrix} \mathbf{E}_1 & \mathbf{E}_2 \\ \mathbf{E}_3 & \mathbf{E}_4 \end{pmatrix} \begin{pmatrix} \sigma_c\boldsymbol{I} & \boldsymbol{Z}_c^\top \\ 0 & \boldsymbol{D}_c \end{pmatrix},
$$

where $\mathbf{E}_1 \in \mathbb{R}^{d_{out}\times d_{in}}$, $\mathbf{E}_2 \in \mathbb{R}^{d_{out}\times M_{in}}$, $\mathbf{E}_3 \in \mathbb{R}^{M_{out}\times d_{in}}$, $\mathbf{E}_4 \in \mathbb{R}^{M_{out}\times M_{in}}$ are standard Gaussian noise samples, and $\bar{\mathbf{W}} \in \mathbb{R}^{d_{out}\times d_{in}}$ and $\bar{\mathbf{U}} \in \mathbb{R}^{M_{out}\times M_{in}}$. Then we construct the

conditional sample $\mathbf{W} \sim p(\mathbf{W}|\mathbf{U})$ as follows, similar to Matheron's rule in the multivariate Gaussian case:

$$\mathbf{W} = \bar{\mathbf{W}} + \sigma_r \sigma_c \mathbf{Z}_r^\top \mathbf{\Psi}_r^{-1}(U - \bar{U})\mathbf{\Psi}_c^{-1}\mathbf{Z}_c. \tag{35}$$

From the above equations we see that $\bar{\mathbf{U}}_r$ and $\bar{\mathbf{U}}_c$ do not contribute to the final $\mathbf{W}$ sample. Therefore we do not need to compute $\bar{\mathbf{U}}_r$ and $\bar{\mathbf{U}}_c$, and we write the separate expressions for $\bar{\mathbf{W}}$ and $\bar{\mathbf{U}}$ as:

$$\bar{\mathbf{W}} = \sigma_r \sigma_c \mathbf{E}_1, \quad \bar{\mathbf{U}} = \underbrace{\mathbf{Z}_r \mathbf{E}_1 \mathbf{Z}_c^\top}_{\bar{\mathbf{U}}_1} + \underbrace{\mathbf{Z}_r \mathbf{E}_2 \mathbf{D}_c}_{\bar{\mathbf{U}}_2} + \underbrace{\mathbf{D}_r \mathbf{E}_3 \mathbf{Z}_c^\top}_{\bar{\mathbf{U}}_3} + \underbrace{\mathbf{D}_r \mathbf{E}_4 \mathbf{D}_c}_{\bar{\mathbf{U}}_4}. \tag{36}$$

Note that $\bar{\mathbf{U}}$ is a sum of four samples from matrix normal distributions. In particular, we have that:

$$\bar{\mathbf{U}}_2 \overset{d}{\sim} \mathcal{MN}(0, \mathbf{Z}_r \mathbf{Z}_r^\top, \mathbf{D}_c^2) \quad \text{and} \quad \bar{\mathbf{U}}_3 \overset{d}{\sim} \mathcal{MN}(0, \mathbf{D}_r^2, \mathbf{Z}_c \mathbf{Z}_c^\top).$$

Hence instead of sampling the "long and thin" Gaussian noise matrices $\mathbf{E}_2$ and $\mathbf{E}_3$, we can reduce variance by sampling standard Gaussian noise matrices $\tilde{E}_2, \tilde{E}_3 \in \mathbb{R}^{M_{out} \times M_{in}}$, and calculate $\bar{\mathbf{U}}$ as

$$\bar{\mathbf{U}} = \mathbf{Z}_r \mathbf{E}_1 \mathbf{Z}_c^\top + \hat{L}_r \tilde{E}_2 \mathbf{D}_c + \mathbf{D}_r \tilde{E}_3 \hat{L}_c^\top + \mathbf{D}_r \mathbf{E}_4 \mathbf{D}_c. \tag{37}$$

This is enabled by calculating the Cholesky decompositions $\hat{L}_r \hat{L}_r^\top = \mathbf{Z}_r \mathbf{Z}_r^\top$ and $\hat{L}_c \hat{L}_c^\top = \mathbf{Z}_c \mathbf{Z}_c^\top$, which have $\mathcal{O}(M_{out}^3)$ and $\mathcal{O}(M_{in}^3)$ run-time costs, respectively. As a reminder, the Cholesky factors are *square* matrices, i.e. $\hat{L}_r \in \mathbb{R}^{M_{out} \times M_{out}}, \hat{L}_c \in \mathbb{R}^{M_{in} \times M_{in}}$). We name the approach the *extended Matheron's rule* for sampling conditional Gaussians when the full joint has a matrix normal form.

To verify the proposed approach, we compute the mean and the variance of the random variable $\mathbf{W}$ defined in Eq. (35), and check if they match the mean and variance of Eq. (34). First as $\bar{\mathbf{W}}, \bar{\mathbf{U}}$ have zero mean, it is straightforward to verify that $\mathbb{E}[W] = \sigma_r \sigma_c \mathbf{Z}_r^\top \mathbf{\Psi}_r^{-1} U \mathbf{\Psi}_c^{-1} \mathbf{Z}_c$ which matches the mean of Eq. (34). For the variance of $\text{vec}(\mathbf{W})$, it requires computing the following terms:

$$\begin{aligned} \mathbb{V}(\text{vec}(\mathbf{W})) =& \mathbb{V}(\text{vec}(\bar{\mathbf{W}})) + \mathbb{V}(\text{vec}(\sigma_r \sigma_c \mathbf{Z}_r^\top \mathbf{\Psi}_r^{-1} \bar{U} \mathbf{\Psi}_c^{-1} \mathbf{Z}_c)) \\ & - 2\text{Cov}[\text{vec}(\mathbf{W}), \text{vec}(\sigma_r \sigma_c \mathbf{Z}_r^\top \mathbf{\Psi}_r^{-1} \bar{U} \mathbf{\Psi}_c^{-1} \mathbf{Z}_c)] \\ =:& \mathbf{A}_1 + \mathbf{A}_2 - 2\mathbf{A}_3. \end{aligned} \tag{38}$$

First it can be shown that

$$\begin{aligned} \mathbf{A}_1 =& \sigma_r^2 \sigma_c^2 \mathbf{I} \quad \text{since } \bar{\mathbf{W}} \sim \mathcal{MN}(0, \sigma_r^2 \mathbf{I}, \sigma_c^2 \mathbf{I}), \\ \mathbf{A}_2 =& \sigma_r^2 \sigma_c^2 \mathbf{Z}_c^\top \mathbf{\Psi}_c^{-1} \mathbf{Z}_c \otimes \mathbf{Z}_r^\top \mathbf{\Psi}_r^{-1} \mathbf{Z}_r \\ \text{as} \quad \mathbf{Z}_r^\top \mathbf{\Psi}_r^{-1} \bar{U} \mathbf{\Psi}_c^{-1} \mathbf{Z}_c \sim& \mathcal{MN}(0, \mathbf{Z}_r^\top \mathbf{\Psi}_r^{-1} \mathbf{Z}_r, \mathbf{Z}_c^\top \mathbf{\Psi}_c^{-1} \mathbf{Z}_c). \end{aligned}$$

For the correlation term $\mathbf{A}_3$, we notice that $\bar{\mathbf{W}}$ and $\bar{\mathbf{U}}$ only share the noise matrix $\mathbf{E}_1$ in the joint sampling procedure Eq. (36). This also means

$$\begin{aligned} \mathbf{A}_3 =& \sigma_r^2 \sigma_c^2 \text{Cov}[\text{vec}(\mathbf{E}_1), \text{vec}(\mathbf{Z}_r^\top \mathbf{\Psi}_r^{-1} \mathbf{Z}_r \mathbf{E}_1 \mathbf{Z}_c^\top \mathbf{\Psi}_c^{-1} \mathbf{Z}_c)] \\ =& \sigma_r^2 \sigma_c^2 \mathbf{Z}_c^\top \mathbf{\Psi}_c^{-1} \mathbf{Z}_c \otimes \mathbf{Z}_r^\top \mathbf{\Psi}_r^{-1} \mathbf{Z}_r. \end{aligned}$$

Plugging in $\mathbf{A}_1, \mathbf{A}_2, \mathbf{A}_3$ into Eq. (38) verifies that $\mathbb{V}(\text{vec}(\mathbf{W}))$ matches the variance of the conditional distribution $p(\text{vec}(\mathbf{W})| \text{vec}(\mathbf{U}))$, showing that the proposed extended Matheron's rule indeed draws samples from the conditional distribution.

As for sampling $\mathbf{W}$ from $q(\mathbf{W}|\mathbf{U})$, since it has the same mean but a rescaled covariance as compared with $p(\mathbf{W}|\mathbf{U})$, we can compute the samples by adapting the extend Matheron's rule as follows. Notice that the mean of $\mathbf{W}$ in Eq. (35) is $\mathbb{E}[\mathbf{W}|\mathbf{U}] = \sigma_r \sigma_c \mathbf{Z}_r^\top \mathbf{\Psi}_r^{-1} U \mathbf{\Psi}_c^{-1} \mathbf{Z}_c$, therefore by rearranging terms, Eq. (35) can be re-written as

$$\begin{aligned} \mathbf{W} =& \mathbf{Z}_r^\top \mathbf{\Psi}_r^{-1} \mathbf{U} \mathbf{\Psi}_c^{-1} \mathbf{Z}_c + [\bar{\mathbf{W}} - \sigma_r \sigma_c \mathbf{Z}_r^\top \mathbf{\Psi}_r^{-1} \bar{\mathbf{U}} \mathbf{\Psi}_c^{-1} \mathbf{Z}_c] \\ :=& \text{mean} + \text{noise}. \end{aligned}$$

So sampling from $q(\mathbf{W}|\mathbf{U})$ can be done by rescaling the noise term in the above equation with the scale parameter $\lambda$. In summary, the extended Matheron's rule for sampling $q(\mathbf{W}|\mathbf{U})$ is as follows:

$$\mathbf{W} = \lambda\bar{\mathbf{W}} + \sigma_r\sigma_c\mathbf{Z}_r^\top\mathbf{\Psi}_r^{-1}(U - \lambda\bar{\mathbf{U}})\mathbf{\Psi}_c^{-1}\mathbf{Z}_c,$$
$$\bar{\mathbf{W}}, \bar{\mathbf{U}} \sim p(\bar{\mathbf{W}}, \bar{\mathbf{U}}_c, \bar{\mathbf{U}}_r, \bar{\mathbf{U}}). \tag{39}$$

Plugging in $\sigma_r\sigma_c = \sigma$ here returns the conditional sampling rule Eq. (12) in the main text.

## G  Function-space view of inducing weights

Here we present the detailed derivations of Section 5.1. Assume a neural network layer with weight $\mathbf{W}$ computes the following transformation of the input $\mathbf{X} = [\boldsymbol{x}_1, ..., \boldsymbol{x}_N], \boldsymbol{x}_i \in \mathbb{R}^{d_{in}\times 1}$:

$$\mathbf{F} = \mathbf{WX}, \ \mathbf{H} = g(\mathbf{F}), \quad \mathbf{W} \in \mathbb{R}^{d_{out}\times d_{in}}, \mathbf{X} \in \mathbb{R}^{d_{in}\times N},$$

where $g(\cdot)$ is the non-linearity. Therefore the Gaussian prior $p(\mathbf{W}) = \mathcal{N}(0, \sigma^2\boldsymbol{I})$ induces a Gaussian distribution on the linear transformation output $\mathbf{F}$, in fact each of the rows in $\mathbf{F} = [\mathbf{f}_1, ..., \mathbf{f}_{d_{out}}]^\top, \mathbf{f}_i \in \mathbb{R}^{N\times 1}$ has a Gaussian process form with linear kernel:

$$\mathbf{f}_i|\mathbf{X} \sim \mathcal{GP}(\mathbf{0}, \mathbf{K}_{\mathbf{XX}}), \quad \mathbf{K}_{\mathbf{XX}}(m, n) = \sigma^2\boldsymbol{x}_m^\top\boldsymbol{x}_n. \tag{40}$$

Performing inference on $\mathbf{F}$ directly has $\mathcal{O}(N^3 + d_{out}N^2)$ cost, so a sparse approximation is needed. Slightly different from the usual approach, we introduce "scaled noisy inducing outputs" $\mathbf{U}_c = [\mathbf{u}_1^c, ..., \mathbf{u}_{d_{out}}^c]^\top \in \mathbb{R}^{d_{out}\times M_{in}}$ in the following way, using shared inducing inputs $\boldsymbol{Z}_c^\top \in \mathbb{R}^{d_{in}\times M_{in}}$:

$$p(\mathbf{f}_i, \hat{\mathbf{u}}_i|\mathbf{X}) = \mathcal{GP}\left(\mathbf{0}, \begin{pmatrix} \mathbf{K}_{\mathbf{XX}} & \mathbf{K}_{\mathbf{XZ}_c} \\ \mathbf{K}_{\mathbf{Z}_c\mathbf{X}} & \mathbf{K}_{\mathbf{Z}_c\mathbf{Z}_c} \end{pmatrix}\right),$$

$$p(\mathbf{u}_i^c|\hat{\mathbf{u}}_i) = \mathcal{N}\left(\frac{\hat{\mathbf{u}}_i}{\sigma_c}, \sigma_r^2\boldsymbol{D}_c^2\right),$$

with $\mathbf{K}_{\mathbf{Z}_c\mathbf{X}} = \sigma^2\boldsymbol{Z}_c\mathbf{X}$ and $\mathbf{K}_{\mathbf{Z}_c\mathbf{Z}_c} = \sigma^2\boldsymbol{Z}_c\boldsymbol{Z}_c^\top$. By marginalising out the "noiseless inducing outputs" $\hat{\mathbf{u}}_i$, we have the joint distribution $p(\mathbf{f}_i, \mathbf{u}_i)$ as

$$p(\mathbf{u}_i^c) = \mathcal{N}(\mathbf{0}, \sigma_r^2\mathbf{\Psi}_c), \ \mathbf{\Psi}_c = \boldsymbol{Z}_c\boldsymbol{Z}_c^\top + \boldsymbol{D}_c^2,$$
$$p(\mathbf{f}_i|\mathbf{X}, \mathbf{u}_i^c) = \mathcal{N}(\sigma_c\sigma^{-2}\mathbf{K}_{\mathbf{XZ}_c}\mathbf{\Psi}_c^{-1}\mathbf{u}_i^c, \mathbf{K}_{\mathbf{XX}} - \sigma^{-2}\mathbf{K}_{\mathbf{XZ}_c}\mathbf{\Psi}_c^{-1}\mathbf{K}_{\mathbf{Z}_c\mathbf{X}}).$$

Collecting all the random variables in matrix forms, this leads to

$$p(\mathbf{U}_c) = \mathcal{MN}(\mathbf{0}, \sigma_r^2\boldsymbol{I}, \mathbf{\Psi}_c),$$
$$p(\mathbf{F}|\mathbf{X}, \mathbf{U}_c) = \mathcal{MN}(\sigma_c\sigma^{-2}\mathbf{U}_c\mathbf{\Psi}_c^{-1}\mathbf{K}_{\mathbf{Z}_c\mathbf{X}}, \sigma_r^2\boldsymbol{I}, \sigma_r^{-2}(\mathbf{K}_{\mathbf{XX}} - \sigma^{-2}\mathbf{K}_{\mathbf{XZ}_c}\mathbf{\Psi}_c^{-1}\mathbf{K}_{\mathbf{Z}_c\mathbf{X}})) \tag{41}$$
$$= \mathcal{MN}(\mathbf{U}_c\mathbf{\Psi}_c^{-1}\sigma_c\boldsymbol{Z}_c\mathbf{X}, \sigma_r^2\boldsymbol{I}, \mathbf{X}^\top\sigma_c^2(\boldsymbol{I} - \boldsymbol{Z}_c^\top\mathbf{\Psi}_c^{-1}\boldsymbol{Z}_c)\mathbf{X}).$$

Also recall from conditioning rules of matrix normal distributions, we have that

$$p(\mathbf{W}|\mathbf{U}_c) = \mathcal{MN}\left(\mathbf{U}_c\mathbf{\Psi}_c^{-1}\sigma_c\boldsymbol{Z}_c, \sigma_r^2\boldsymbol{I}, \sigma_c^2(\boldsymbol{I} - \boldsymbol{Z}_c^\top\mathbf{\Psi}_c^{-1}\boldsymbol{Z}_c)\right).$$

Since for $\mathbf{W} \sim \mathcal{MN}(\mathbf{M}, \mathbf{\Sigma}_1, \mathbf{\Sigma}_2)$ we have $\mathbf{WX} \stackrel{d}{\sim} \mathcal{MN}(\mathbf{MX}, \mathbf{\Sigma}_1, \mathbf{X}^\top\mathbf{\Sigma}_2\mathbf{X})$, this immediately shows that $p(\mathbf{F}|\mathbf{X}, \mathbf{U}_c)$ is the push-forward distribution of $p(\mathbf{W}|\mathbf{U}_c)$ for the operation $\mathbf{F} = \mathbf{WX}$. In other words:

$$\mathbf{F} \sim p(\mathbf{F}|\mathbf{X}, \mathbf{U}_c) \quad \Leftrightarrow \quad \mathbf{W} \sim p(\mathbf{W}|\mathbf{U}_c), \ \mathbf{F} = \mathbf{WX}.$$

This confirms the interpretation of $\mathbf{U}_c$ as "scaled noisy inducing outputs" that lie in the same space as $\mathbf{F}$. Notice that in the main text we provide a pictorial visualisation of $\mathbf{U}_c$ by selecting $\sigma_c = 1$. As the inducing weights $\mathbf{U}$ are the focus of our analysis here, we conclude that this specific choice of $\sigma_c$ is without loss of generality.

So far the $\mathbf{U}_c$ variables assist the posterior inference to capture correlations across functions values of different inputs. Up to now the function values remain independent across output dimensions, which is also reflected by the diagonal row covariance matrices in the above matrix normal distributions. As in neural networks the output dimension can be fairly large (e.g. $d_{out} = 1000$), to further improve memory efficiency, we proceed to project the *column vectors* of $\mathbf{U}_c$ to an $M_{out}$ dimensional space

with $M_{out} << d_{out}$. This dimension reduction step is done with a generative approach, similar to probabilistic PCA (Tipping & Bishop, 1999):

$$\mathbf{U} \sim \mathcal{MN}(0, \mathbf{\Psi}_r, \mathbf{\Psi}_c),$$
$$\mathbf{U}_c|\mathbf{U} \sim \mathcal{MN}(\sigma_r \mathbf{Z}_r^\top \mathbf{\Psi}_r^{-1}\mathbf{U}, \sigma_r^2(\mathbf{I} - \mathbf{Z}_r^\top \mathbf{\Psi}_r^{-1}\mathbf{Z}_r), \mathbf{\Psi}_c). \tag{42}$$

Note that the column covariance matrices in the above two matrix normal distributions are the same, and the conditional sampling procedure is done by a linear transformation of the columns in $\mathbf{U}$ plus noise terms. Again from the marginalisation and conditioning rules of matrix normal distributions, we have that the full joint distribution Eq. (5), after proper marginalisation and conditioning, returns

$$p(\mathbf{U}) = \mathcal{MN}(0, \mathbf{\Psi}_r, \mathbf{\Psi}_c),$$
$$p(\mathbf{U}_c|\mathbf{U}) = \mathcal{MN}(\sigma_r \mathbf{Z}_r^\top \mathbf{\Psi}_r^{-1}\mathbf{U}, \sigma_r^2(\mathbf{I} - \mathbf{Z}_r^\top \mathbf{\Psi}_r^{-1}\mathbf{Z}_r), \mathbf{\Psi}_c).$$

This means $\mathbf{U}$ can be viewed as "projected noisy inducing points" for the GPs $p(\mathbf{F})$, whose corresponding "inducing inputs" are row vectors in $\mathbf{Z}_c$. Similarly, column vectors in $\mathbf{U}_r\mathbf{X}$ can be viewed as the noisy projections of the column vectors in $\mathbf{F}$, in other words $\mathbf{U}_r$ can also be viewed as "neural network weights" connecting the data inputs $\mathbf{X}$ to the projected output space that $\mathbf{U}$ lives in.

As for the variational objective, since $q(\mathbf{W}|\mathbf{U})$ and $p(\mathbf{W}|\mathbf{U})$ only differ in the scale of the covariance matrices, it is straightforward to show that the push-forward distribution $q(\mathbf{W}|\mathbf{U}) \rightarrow q(\mathbf{F}|\mathbf{X}, \mathbf{U})$ has the same mean as $p(\mathbf{F}|\mathbf{X}, \mathbf{U})$, but with a different covariance matrix that scales $p(\mathbf{F}|\mathbf{X}, \mathbf{U})$'s covariance matrix by $\lambda^2$. As the operation $\mathbf{F} = \mathbf{W}\mathbf{X}$ maps $\mathbf{W} \in \mathbb{R}^{d_{out} \times d_{in}}$ to $\mathbf{F} \in \mathbb{R}^{d_{out} \times N}$, this means the conditional KL is scaled up/down, depending on whether $N \geq d_{in}$ or not:

$$\mathbb{KL}[q(\mathbf{F}|\mathbf{X}, \mathbf{U})||p(\mathbf{F}|\mathbf{X}, \mathbf{U})] = \frac{N}{d_{in}}R(\lambda),$$
$$R(\lambda) := \mathbb{KL}[q(\mathbf{W}|\mathbf{U})||p(\mathbf{W}|\mathbf{U})].$$

In summary, the push-forward distribution of $q(\mathbf{W}_{1:L}, \mathbf{U}_{1:L}) \rightarrow q(\mathbf{F}_{1:L}, \mathbf{U}_{1:L})$ is

$$q(\mathbf{F}_{1:L}, \mathbf{U}_{1:L}) = \prod_{l=1}^{L} q(\mathbf{F}_l|\mathbf{F}_{l-1}, \mathbf{U}_l)q(\mathbf{U}_l), \quad \mathbf{F}_0 := \mathbf{X},$$

and the corresponding variational lower-bound for $q(\mathbf{F}_{1:L}, \mathbf{U}_{1:L})$ becomes (with $\mathcal{D} = (\mathbf{X}, \mathbf{Y})$)

$$\mathcal{L}(q(\mathbf{F}_{1:L}, \mathbf{U}_{1:L})) = \mathbb{E}_{q(\mathbf{F}_{1:L})}[\log p(\mathbf{Y}|\mathbf{F}_L)] - \sum_{l=1}^{L} \left( \frac{N}{d_{in}^l}R(\lambda_l) + \mathbb{KL}[q(\mathbf{U}_l)||p(\mathbf{U}_l)] \right), \tag{43}$$

with $d_{in}^l$ the input dimension of layer $l$.

Note that

$$\mathbb{E}_{q(\mathbf{F}_{1:L})}[\log p(\mathbf{Y}|\mathbf{F}_L)] = \mathbb{E}_{q(\mathbf{W}_{1:L})}[\log p(\mathbf{Y}|\mathbf{X}, \mathbf{W}_{1:L})] = \mathbb{E}_{q(\mathbf{W}_{1:L})}[\log p(\mathcal{D}|\mathbf{W}_{1:L})]. \tag{44}$$

Comparing equations eqs. (9) and (43), we see that the only difference between weight-space and function-space variational objectives comes in the scale of the conditional KL term. Though not investigated in the experiments, we conjecture that it could bring potential advantage to optimise the following variational lower-bound:

$$\tilde{\mathcal{L}}(q(\mathbf{F}_{1:L}, \mathbf{U}_{1:L})) = \mathbb{E}_{q(\mathbf{F}_{1:L})}[\log p(\mathbf{Y}|\mathbf{F}_L)] - \sum_{l=1}^{L} \left( \beta_l R(\lambda_l) + \mathbb{KL}[q(\mathbf{U}_l)||p(\mathbf{U}_l)] \right), \tag{45}$$
$$\beta_l = \min(1, \frac{N}{d_{in}^l}).$$

The intuition is that, as uncertainty is expected to be lower when $N \geq d_{in}$, it makes sense to use $\beta = 1 \leq N/d_{in}$ to reduce the regularisation effect introduced by the KL term. In other words, this allows the variational posterior to focus more on fitting the data, and in this "large-data" regime over-fitting is less likely to appear. On the other hand, function-space inference approaches (such as GPs) often return better uncertainty estimates when trained on small data ($N < d_{in}$). So choosing $\beta = N/d_{in} < 1$ in this case would switch to function-space inference and thereby improving uncertainty quality potentially. In the CIFAR experiments, the usage of convolutional filters leads to the fact that $N \geq d_{in}^l$ for all ResNet layers. Therefore in those experiments $\beta_l = 1$ for all layers, which effectively falls back to the weight-space objective Eq. (9).

# H  Whitening and hierarchical inducing variables

The inducing weights $\mathbf{U}_{1:L}$ further allow for introducing a single inducing weight matrix $\mathbf{U}$ that is shared across the network. By doing so, correlations of weights between layers in the approximate posterior are introduced. The inducing weights are then sampled jointly conditioned on the global inducing weights. This requires that all inducing weight matrices are of the same size along at least one axis, such that they can be concatenated along the other one.

The easiest way of introducing a global inducing weight matrix is by proceeding similarly to the introduction of the per-layer inducing weights. As a pre-requisite, we need to work with "whitened" inducing weights, i.e. set the covariance of the marginal $p(\mathbf{U}_l)$ to the identity and pre-multiply the covariance block between $\mathbf{W}_l$ and $\mathbf{U}_l$ with the inverse Cholesky of $\mathbf{\Psi}_l$. In this whitened model, the full augmented prior per-layer is:

$$\begin{pmatrix} \mathbf{W} & \mathbf{U}_c \\ \mathbf{U}_r & \mathbf{U} \end{pmatrix} \sim p(\mathbf{W}, \mathbf{U}_c, \mathbf{U}_r, \mathbf{U}) := \mathcal{MN}(0, \tilde{\mathbf{\Sigma}}_r, \tilde{\mathbf{\Sigma}}_r), \tag{46}$$

$$\text{with} \quad \tilde{\boldsymbol{L}}_r = \begin{pmatrix} \sigma_r \boldsymbol{I} & 0 \\ \boldsymbol{L}_r^{-1} \boldsymbol{Z}_r & \boldsymbol{L}_r^{-1} \boldsymbol{D}_r \end{pmatrix} \tag{47}$$

$$\text{s.t.} \quad \tilde{\mathbf{\Sigma}}_r = \tilde{\boldsymbol{L}}_r \tilde{\boldsymbol{L}}_r^\top = \begin{pmatrix} \sigma_r^2 \boldsymbol{I} & \sigma_r \boldsymbol{Z}_r^\top \boldsymbol{L}_r^{-\top} \\ \sigma_r \boldsymbol{L}_r^{-1} \boldsymbol{Z}_r & \boldsymbol{I} \end{pmatrix}$$

$$\text{and} \quad \tilde{\boldsymbol{L}}_c = \begin{pmatrix} \sigma_c \boldsymbol{I} & 0 \\ \boldsymbol{L}_c^{-1} \boldsymbol{Z}_c & \boldsymbol{L}_c^{-1} \boldsymbol{D}_c \end{pmatrix} \tag{48}$$

$$\text{s.t.} \quad \tilde{\mathbf{\Sigma}}_c = \tilde{\boldsymbol{L}}_c \tilde{\boldsymbol{L}}_c^\top = \begin{pmatrix} \sigma_c^2 \boldsymbol{I} & \sigma_c \boldsymbol{Z}_c^\top \boldsymbol{L}_c^{-\top} \\ \sigma_c \boldsymbol{L}_c^{-1} \boldsymbol{Z}_c & \boldsymbol{I} \end{pmatrix}.$$

One can verify that this whitened model leads to the same conditional distribution $p(\mathbf{W}|\mathbf{U})$ as presented in the main text. After whitening, for each $\mathbf{U}_l$ we have that $p(\text{vec}(\mathbf{U}_l)) = \mathcal{N}(0, \boldsymbol{I})$, therefore we can also write their joint distribution as $p(\text{vec}(\mathbf{U}_{1:L})) = \mathcal{N}(0, \boldsymbol{I})$. In order to construct a matrix normal prior $p(\mathbf{U}_{1:L}) = \mathcal{MN}(0, \boldsymbol{I}, \boldsymbol{I})$, the inducing weight matrices $\mathbf{U}_{1:L}$ needs to be stacked either along the rows or columns, requiring the other dimension to be matching across all layers. Then, As the covariance is the identity with $\sigma = \sigma_r = \sigma_c = 1$, we can augment $p(\mathbf{U}_{1:L})$ in the exact same way as we previously augmented the prior $p(\mathbf{W}_l)$ with $\mathbf{U}_l$.

# I  Open-source code

We open-source our approach as a PyTorch wrapper `bayesianize`:
https://github.com/microsoft/bayesianize

`bayesianize` is a lightweight Bayesian neural network (BNN) wrapper, and the goal is to allow for easy conversion of neural networks in existing scripts to BNNs with minimal changes to the code. Currently the wrapper supports the following uncertainty estimation methods for feed-forward neural networks and convolutional neural networks:

- Mean-field variational inference (MFVI) with fully factorised Gaussian (FFG) approximation, i.e. FFG-$\mathbf{W}$ in abbreviation.
- Variational inference with full-covariance Gaussian approximation (FCG-$\mathbf{W}$).
- Inducing weight approaches, including FFG-$\mathbf{U}$, FCG-$\mathbf{U}$ and Ensemble-$\mathbf{U}$.

The main workhorse of our library is the `bayesianize_` function, which turns deterministic `nn.Linear` and `nn.Conv` layers into their Bayesian counterparts. For example, to construct a Bayesian ResNet-18 that uses the variational inducing weight method, run:

```python
import bnn
net = torchvision.models.resnet18()
bnn.bayesianize_(net, inference="inducing", inducing_rows=64, inducing_cols=64)
```

In the above code, `inducing_rows` corresponds to $M_{out}$ and `inducing_cols` corresponds to $M_{in}$. In other words, they specify the dimensions of $\mathbf{U} \in \mathbb{R}^{M_{out} \times M_{in}}$. Then the converted BNN can be trained in almost identical way as one would train a deterministic net:

```
yhat = net(x_train)
nll = F.cross_entropy(yhat, y_train)
kl = sum(m.kl_divergence() for m in net.modules()
         if hasattr(m, "kl_divergence"))
loss = nll + kl / dataset_size
loss.backward()
optim.step()
```

Note that while the call to the forward method of the `net` looks the same, it is no longer deterministic because the weights are sampled, so subsequent calls will lead to different predictions. Therefore, when testing, an average of multiple predictions is needed. For example, in BNN classification:

```
net.eval()
with torch.no_grad():
    logits = torch.stack([net(x_test) for _ in range(num_samples)])
    probs = logits.softmax(-1).mean(0)
```

In the above code, `probs` computes

$$p(\boldsymbol{y}|\boldsymbol{x}, \mathcal{D}) \approx \frac{1}{K} \sum_{k=1}^{K} p(\boldsymbol{y}|\boldsymbol{x}, \mathbf{W}_{1:L}^{(k)}), \quad \mathbf{W}_{1:L}^{(k)} \sim q,$$

where $K$ is the number of Monte Carlo samples `num_samples`.

`bayesianize` also supports using different methods or arguments for different layers, by passing in a dictionary for the inference argument. This way we can, for example, take a *pre-trained* ResNet and only perform (approximate) Bayesian inference over the weights of the final, linear layer:

```
net = ... # load pre-trained network with net.fc as the last layer
bnn.bayesianize_(net, inference={
    net.fc: {"inference": "fcg"}
    })
optim = torch.optim.Adam(net.fc.parameters(), 1e-3)
# then train the last Bayesian layer accordingly
...
```

For more possible ways of configuring the BNN settings, see example config `json` files in the open-source repository.

## J  Experimental details

### J.1  Regression experiments

Following (Foong et al., 2019), we sample 50 inputs each from $\mathcal{U}[-1, -0.7]$ and $\mathcal{U}[0.5, 1]$ as inputs and targets $y \sim \mathcal{N}(\cos(0.4x + 0.8), 0.01)$. As a prior we use a zero-mean Gaussian with standard deviation $\frac{4}{\sqrt{d_{in}}}$ for the weights and biases of each layer. Our network architecture has a single hidden layer of 50 units and uses a $\tanh$-nonlinearity. All three variational methods are optimised using Adam (Kingma & Ba, 2015) for $20,000$ updates with an initial learning rate of $10^{-3}$. We average over 32 MC samples from the approximate posterior for every update. For Ensemble-U and FCG-U we decay the learning rate by a factor of $0.1$ after $10,000$ updates and the size of the inducing weight matrix is $2 \times 25$ for the input layer (accounting for the bias) and $25 \times 1$ for the output layer. Ensemble-U uses an ensemble size of $8$.

For NUTS we use the implementation provided in Pyro (Bingham et al., 2019). We draw a total of $25,000$ samples, discarding the first $5000$ as burn-in and using $1000$ randomly selected ones for prediction.

### J.2  Classification experiments

We base our implementation on the Resnet-18 class in torchvision (Paszke et al., 2019), replacing the input convolutional layer with a $3 \times 3$ kernel size and removing the max-pooling layer. We train the

deterministic network on CIFAR-10 using Adam with a learning rate of $3 \times 10^{-4}$ for 200 epochs. On CIFAR-100 we found SGD with a momentum of $0.9$ and initial learning rate of $0.1$ decayed by a factor of $0.1$ after 60, 120 and 160 epochs to lead to better accuracies. The ensemble is formed of the five deterministic networks trained with different random seeds.

For FFG-**W** we initialised the mean parameters using the default initialisation in pytorch for the corresponding deterministic layers. The initial standard deviations are set to $10^{-4}$. We train using Adam for 200 epochs on CIFAR-10 with a learning rate of $3 \times 10^{-4}$, and 300 epochs on CIFAR-100 with an initial learning rate of $10^{-3}$, decaying by a factor of $0.1$ after 200 epochs. On both datasets we only use the negative log likelihood (NLL) part of the variational lower bound for the first 100 epochs as initialisation to the maximum likelihood parameter and then anneal the weight of the kl term linearly over the following 50 epochs. For the prior we use a standard Gaussian on all weights and biases and restrict the standard deviation of the posterior to be at most $\sigma_{max} = 0.1$. We also experimented with a larger upper limit of $\sigma_{max} = 0.3$, but found this to negatively affect both accuracy and calibration.

All the **U**-space approaches use Gaussian priors $p(\text{vec}(\mathbf{W}_l)) = \mathcal{N}(0, 1/\sqrt{d_{in}})$, motivated by the connection to GPs. Hyperparameter and optimisation details for the inducing weight methods on CIFAR-10 are discussed below in the details on the ablation study. We train all methods using Adam for 300 epochs with a learning rate of $10^{-3}$ for the first 200 epochs and then decay by a factor of 10. For the initial 100 epochs we train without the KL-term of the ELBO and then anneal its weight linearly over the following 50 epochs. For the tables and figures in the main text, we set $\lambda_{max} = 0.1$ for Ensemble-**U** on both datasets, and $\sigma_{max} = 0.1, \lambda_{max} = 0.03$ on CIFAR-10 for FFG-**U**. We initialise the entries of the $Z$ matrices by sampling from a zero-mean Gaussian with variance $\frac{1}{M}$ and set the diagonal entries of the $D$ matrices to $10^{-3}$. For FFG-**U** we initialise the mean of the variational Gaussian posterior in **U**-space by sampling from a standard Gaussian and set the initial variances to $10^{-3}$. For Ensemble-**U** initialisation, we draw an $M \times M$ shaped sample from a standard Gaussian that is shared across ensemble members and add independent Gaussian noise with a standard deviation of $0.1$ for each member. We use an ensemble size of 5. During optimisation, we draw 1 MC samples per update step for both FFG-**U** and Ensemble-**U** (such that each ensemble member is used once). For testing we use 20 MC samples for all variational methods. We fit BatchNorm parameters by minimising the NLL.

**The study of hyper-parameter selection on CIFAR-10**   We run the inducing weight method with the following options:

- Row/column dimensions of $\mathbf{U}_l$ ($M$): $M \in \{16, 32, 64, 128\}$.
  We set $M = M_{in} = M_{out}$ except for the last layer, where we use $M_{in} = M$ and $M_{out} = 10$.
- $\lambda_{max}$ values for FFG-**U** and Ensemble-**U**: $\lambda_{max} \in \{0, 0.03, 0.1, 0.3\}$.
  When $\lambda_{max} = 0$ it means $q(\mathbf{W}|\mathbf{U})$ is a delta measure centered at the mean of $p(\mathbf{W}|\mathbf{U})$.
- $\sigma_{max}$ values for FFG-**U**: $\sigma_{max} = \{0, 0.1, 0.3\}$.
  When $\sigma_{max} = 0$ we use a MAP estimate for **U**.

Each experiment is repeated with 5 random seeds to collect the averaged results on a single NVIDIA RTX 2080TI. The models are trained with 100 epochs in total. We first run 50 epochs of maximum likelihood to initialise the model, then run 40 epochs training on the modified variational lower-bound with KL annealing (linear scaling schedule), finally we run 10 epochs of training with the variational lower-bound (i.e. no KL annealing). We use Adam optimiser with learning rate $3e-4$ and the default $\beta_1, \beta_2$ parameters in PyTorch's implementation.

# K   Additional Results

Below in Tables 5 and 6 we provide extended versions of Table 2. This table contains standard errors across the random seeds for the corresponding metrics and we additionally report NLLs and Brier scores. The error bar results are not available for Ensemble-**W**, as it is constructed from the 5 independently trained deterministic neural network with maximum likelihood.

The results of pruning different fractions of the weights can be found in Table 7 for the in-distribution uncertainty evaluation for Resnet-50 and the OOD detection in Table 8. For the pruning experiments,

Table 5: Complete in-distribution results for Resnet-50 on CIFAR10

| Method | Acc. ↑ | NLL ↓ | ECE ↓ | Brier ↓ |
|---|---|---|---|---|
| Deterministic | $94.72_{\pm0.08}$ | $0.43_{\pm0.01}$ | $4.46_{\pm0.08}$ | $0.10_{\pm0.00}$ |
| Ensemble-W | 95.90 | 0.20 | 1.08 | 0.06 |
| FFG-W | $94.13_{\pm0.08}$ | $0.18_{\pm0.00}$ | $0.50_{\pm0.06}$ | $0.09_{\pm0.00}$ |
| FFG-U (M=64) | $94.40_{\pm0.05}$ | $0.17_{\pm0.00}$ | $0.64_{\pm0.06}$ | $0.08_{\pm0.00}$ |
| FFG-U (M=128) | $94.66_{\pm0.09}$ | $0.17_{\pm0.00}$ | $1.59_{\pm0.06}$ | $0.08_{\pm0.00}$ |
| Ensemble-U (M=64) | $94.94_{\pm0.07}$ | $0.16_{\pm0.00}$ | $0.45_{\pm0.06}$ | $0.08_{\pm0.00}$ |
| Ensemble-U (M=128) | $95.34_{\pm0.05}$ | $0.17_{\pm0.00}$ | $1.29_{\pm0.05}$ | $0.07_{\pm0.00}$ |

Table 6: Complete in-distribution results for Resnet-50 on CIFAR100

| Method | Acc. ↑ | NLL ↓ | ECE ↓ | Brier ↓ |
|---|---|---|---|---|
| Deterministic | $75.73_{\pm0.16}$ | $2.14_{\pm0.01}$ | $19.69_{\pm0.15}$ | $0.43_{\pm0.00}$ |
| Ensemble-W | 79.33 | 1.23 | 6.51 | 0.31 |
| FFG-W | $74.44_{\pm0.27}$ | $1.01_{\pm0.01}$ | $4.24_{\pm0.10}$ | $0.35_{\pm0.00}$ |
| FFG-U (M=64) | $75.37_{\pm0.09}$ | $0.92_{\pm0.01}$ | $2.29_{\pm0.39}$ | $0.34_{\pm0.00}$ |
| FFG-U (M=128) | $75.88_{\pm0.13}$ | $0.91_{\pm0.00}$ | $6.66_{\pm0.15}$ | $0.34_{\pm0.00}$ |
| Ensemble-U (M=64) | $75.97_{\pm0.12}$ | $0.90_{\pm0.00}$ | $1.12_{\pm0.06}$ | $0.33_{\pm0.00}$ |
| Ensemble-U (M=128) | $77.61_{\pm0.11}$ | $0.94_{\pm0.00}$ | $6.00_{\pm0.12}$ | $0.32_{\pm0.00}$ |

we take the parameters from the corresponding full runs, set a fixed percentage of the weights to be deterministically 0 and fine-tune the remaining weights with a new optimizer for 50 epochs. We use Adam with a learning rate of $10^{-4}$. For FFG-**W** we select the weights with the smallest ratio of absolute mean to standard deviation in the approximate posterior, and for FFG-**U** the $Z$ parameters with the smallest absolute value.

For FFG-**W** we find that pruning up to 90% of the weights only worsens ECE and NLL on the more difficult CIFAR100 datasets. Pruning 99% of the weights worsens accuracy and OOD detection, but interestingly improves ECE on CIFAR100, where accuracy is noticeably worse.

Pruning 25% and 50% of the $Z$ parameters in FFG-**U** results in a total parameter count of $4,408,790$ and $3,106,678$, i.e. $18.7$ and $13.2\%$ of the deterministic parameters respectively on ResNet-50. Up to pruning 50% of the $Z$ parameters, we find that only ECE becomes slightly worse, although on CIFAR100 it is still better than the ECE for FFG-**W** at 100% of the weights. Other metrics are not affected neither on the in-distribution uncertainty or OOD detection, except for a minor drop in accuracy.

Table 7: Pruning in-distribution uncertainty results for Resnet-50. The percentage refers to the weights left for FFG-**W** and the number of $Z$ parameters for FFG-**U**.

| Method | CIFAR10 | | | | CIFAR100 | | | |
|---|---|---|---|---|---|---|---|---|
| | Acc. ↑ | NLL ↓ | ECE ↓ | Brier ↓ | Acc. ↑ | NLL ↓ | ECE ↓ | Brier ↓ |
| FFG-W (100%) | $94.13_{\pm0.08}$ | $0.18_{\pm0.00}$ | $0.50_{\pm0.06}$ | $0.09_{\pm0.00}$ | $74.44_{\pm0.27}$ | $1.01_{\pm0.01}$ | $4.24_{\pm0.10}$ | $0.35_{\pm0.00}$ |
| FFG-W (50%) | $94.07_{\pm0.03}$ | $0.18_{\pm0.00}$ | $0.40_{\pm0.04}$ | $0.09_{\pm0.00}$ | $74.38_{\pm0.21}$ | $1.02_{\pm0.01}$ | $4.15_{\pm0.09}$ | $0.36_{\pm0.00}$ |
| FFG-W (10%) | $94.17_{\pm0.06}$ | $0.18_{\pm0.00}$ | $0.58_{\pm0.02}$ | $0.09_{\pm0.00}$ | $74.42_{\pm0.23}$ | $1.10_{\pm0.01}$ | $6.86_{\pm0.08}$ | $0.36_{\pm0.00}$ |
| FFG-W (1%) | $93.60_{\pm0.09}$ | $0.19_{\pm0.00}$ | $0.80_{\pm0.06}$ | $0.09_{\pm0.00}$ | $67.08_{\pm0.33}$ | $1.19_{\pm0.01}$ | $1.36_{\pm0.18}$ | $0.44_{\pm0.00}$ |
| FFG-W (0.1%) | $58.59_{\pm0.75}$ | $1.15_{\pm0.02}$ | $7.33_{\pm0.23}$ | $0.55_{\pm0.01}$ | $10.66_{\pm0.43}$ | $3.97_{\pm0.03}$ | $2.80_{\pm0.15}$ | $0.96_{\pm0.00}$ |
| FFG-U (100%) | $94.40_{\pm0.05}$ | $0.17_{\pm0.00}$ | $0.64_{\pm0.06}$ | $0.08_{\pm0.00}$ | $75.37_{\pm0.09}$ | $0.92_{\pm0.01}$ | $2.29_{\pm0.39}$ | $0.34_{\pm0.00}$ |
| FFG-U (75%) | $94.45_{\pm0.05}$ | $0.18_{\pm0.00}$ | $2.19_{\pm0.11}$ | $0.09_{\pm0.00}$ | $75.26_{\pm0.07}$ | $0.93_{\pm0.00}$ | $3.74_{\pm0.67}$ | $0.35_{\pm0.00}$ |
| FFG-U (50%) | $94.31_{\pm0.07}$ | $0.18_{\pm0.00}$ | $2.31_{\pm0.09}$ | $0.09_{\pm0.00}$ | $74.89_{\pm0.07}$ | $0.94_{\pm0.00}$ | $5.04_{\pm0.77}$ | $0.35_{\pm0.00}$ |
| FFG-U (25%) | $93.34_{\pm0.03}$ | $0.22_{\pm0.00}$ | $4.83_{\pm0.09}$ | $0.11_{\pm0.00}$ | $71.32_{\pm0.16}$ | $1.09_{\pm0.01}$ | $12.44_{\pm0.90}$ | $0.42_{\pm0.00}$ |

The number of parameters for FFG-U and Ensemble-U with an ensemble size of 5 in the ResNet-50 experiments are reported in Table 9. The corresponding parameter counts for pruning FFG-**W** and FFG-**U** (M=64) are in Table 10.

In Tables 11 to 14 we report the numerical results for Fig. 5.

Table 8: Pruning OOD detection metrics for Resnet-50 trained on CIFAR10/100.

| In-dist → OOD | C10 → C100 | | C10 → SVHN | | C100 → C10 | | C100 → SVHN | |
|---|---|---|---|---|---|---|---|---|
| Method / Metric | AUROC | AUPR | AUROC | AUPR | AUROC | AUPR | AUROC | AUPR |
| FFG-W (100%) | $.88_{\pm.00}$ | $.90_{\pm.00}$ | $.90_{\pm.01}$ | $.86_{\pm.01}$ | $.76_{\pm.00}$ | $.79_{\pm.00}$ | $.80_{\pm.01}$ | $.69_{\pm.01}$ |
| FFG-W (50%) | $.88_{\pm.00}$ | $.90_{\pm.00}$ | $.90_{\pm.00}$ | $.86_{\pm.00}$ | $.76_{\pm.00}$ | $.79_{\pm.00}$ | $.79_{\pm.01}$ | $.69_{\pm.01}$ |
| FFG-W (10%) | $.88_{\pm.00}$ | $.90_{\pm.00}$ | $.90_{\pm.01}$ | $.87_{\pm.01}$ | $.76_{\pm.00}$ | $.79_{\pm.00}$ | $.79_{\pm.01}$ | $.69_{\pm.01}$ |
| FFG-W (1%) | $.88_{\pm.00}$ | $.89_{\pm.00}$ | $.90_{\pm.01}$ | $.86_{\pm.01}$ | $.72_{\pm.00}$ | $.75_{\pm.00}$ | $.71_{\pm.02}$ | $.57_{\pm.02}$ |
| FFG-W (0.1%) | $.65_{\pm.01}$ | $.65_{\pm.01}$ | $.38_{\pm.03}$ | $.24_{\pm.03}$ | $.56_{\pm.01}$ | $.59_{\pm.01}$ | $.31_{\pm.04}$ | $.21_{\pm.02}$ |
| FFG-U (100%) | $.89_{\pm.00}$ | $.91_{\pm.00}$ | $.94_{\pm.01}$ | $.91_{\pm.01}$ | $.77_{\pm.00}$ | $.79_{\pm.00}$ | $.83_{\pm.01}$ | $.74_{\pm.01}$ |
| FFG-U (75%) | $.89_{\pm.00}$ | $.91_{\pm.00}$ | $.94_{\pm.00}$ | $.91_{\pm.00}$ | $.77_{\pm.00}$ | $.79_{\pm.00}$ | $.82_{\pm.01}$ | $.72_{\pm.02}$ |
| FFG-U (50%) | $.89_{\pm.00}$ | $.91_{\pm.00}$ | $.93_{\pm.00}$ | $.91_{\pm.00}$ | $.77_{\pm.00}$ | $.79_{\pm.00}$ | $.82_{\pm.01}$ | $.72_{\pm.02}$ |
| FFG-U (25%) | $.88_{\pm.00}$ | $.90_{\pm.00}$ | $.92_{\pm.01}$ | $.88_{\pm.01}$ | $.75_{\pm.00}$ | $.77_{\pm.00}$ | $.82_{\pm.02}$ | $.72_{\pm.03}$ |

Table 9: Parameter counts for the inducing models with varying $\mathbf{U}$ size $M$.

| $M$ | Method | $M = 16$ | $M = 32$ | $M = 64$ | $M = 128$ | $M = 256$ | Deterministic |
|---|---|---|---|---|---|---|---|
| Abs. value | FFG-U | $1,384,662$ | $2,771,446$ | $5,710,902$ | $12,253,366$ | $27,992,502$ | $23,520,842$ |
| | Ensemble-U | $1,426,134$ | $2,937,334$ | $6,374,454$ | $14,907,574$ | $38,609,334$ | |
| rel. size (%) | FFG-U | 5.89 | 11.78 | 24.28 | 52.10 | 119.01 | 100 |
| | Ensemble-U | 6.06 | 12.49 | 27.10 | 63.38 | 164.15 | |

In Tables 15 to 18 we report the corresponding results for pruning FFG-W and FFG-U. See Fig. 8 for visualisation.

Table 10: Pruning parameter counts for keeping fractions of the weights in FFG-**W** and the $Z$ parameters in FFG-**U**.

| Method | Abs. param. count | rel. size (%) |
|--------|------------------|---------------|
| FFG-**W** (100%) | $46,988,564$ | 199.8 |
| FFG-**W** (50%) | $23,520,852$ | 100 |
| FFG-**W** (10%) | $4,746,682$ | 20.2 |
| FFG-**W** (1%) | $522,494$ | 2.2 |
| FFG-**W** (0.1%) | $100,075$ | 0.4 |
| FFG-**U** (100%) | $5,710,902$ | 24.28 |
| FFG-**U** (75%) | $4,408,790$ | 18.7 |
| FFG-**U** (50%) | $3,106,678$ | 13.2 |
| FFG-**U** (25%) | $1,804,566$ | 7.7 |

Table 11: Corrupted CIFAR-10 accuracy ($\uparrow$) values (in %).

| Method | Skew Intensity | | | | |
|--------|------|------|------|------|------|
| | 1 | 2 | 3 | 4 | 5 |
| Deterministic | $87.90_{\pm2.31}$ | $82.02_{\pm2.84}$ | $76.31_{\pm3.80}$ | $68.91_{\pm4.81}$ | $57.94_{\pm5.10}$ |
| Ensemble-W | $89.45_{\pm2.27}$ | $83.94_{\pm2.71}$ | $78.40_{\pm3.61}$ | $71.18_{\pm4.53}$ | $60.15_{\pm4.82}$ |
| FFG-W | $83.80_{\pm2.43}$ | $76.22_{\pm3.10}$ | $69.30_{\pm4.11}$ | $61.82_{\pm4.66}$ | $50.72_{\pm4.68}$ |
| FFG-U | $86.90_{\pm2.47}$ | $80.33_{\pm3.14}$ | $74.34_{\pm4.06}$ | $67.23_{\pm4.75}$ | $57.00_{\pm4.83}$ |
| Ensemble-U | $87.35_{\pm2.39}$ | $80.45_{\pm3.19}$ | $73.89_{\pm4.23}$ | $66.52_{\pm4.96}$ | $54.89_{\pm5.26}$ |

Table 12: Corrupted CIFAR-10 ECE ($\downarrow$) values (in %).

| Method | Skew Intensity | | | | |
|--------|------|------|------|------|------|
| | 1 | 2 | 3 | 4 | 5 |
| Deterministic | $10.41_{\pm2.03}$ | $15.58_{\pm2.52}$ | $20.56_{\pm3.37}$ | $27.06_{\pm4.23}$ | $37.26_{\pm4.65}$ |
| Ensemble-W | $4.12_{\pm1.31}$ | $7.01_{\pm1.64}$ | $10.10_{\pm2.33}$ | $14.12_{\pm2.84}$ | $20.48_{\pm2.95}$ |
| FFG-W | $13.05_{\pm0.64}$ | $12.14_{\pm0.89}$ | $11.56_{\pm0.93}$ | $10.77_{\pm1.05}$ | $10.86_{\pm1.31}$ |
| FFG-U | $2.47_{\pm1.04}$ | $4.93_{\pm1.66}$ | $7.77_{\pm2.44}$ | $11.27_{\pm2.81}$ | $16.16_{\pm2.92}$ |
| Ensemble-U | $2.77_{\pm1.04}$ | $5.86_{\pm1.69}$ | $9.06_{\pm2.40}$ | $12.54_{\pm2.66}$ | $19.66_{\pm3.12}$ |

Table 13: Corrupted CIFAR-100 accuracy ($\uparrow$) values (in %).

| Method | Skew Intensity | | | | |
|--------|------|------|------|------|------|
| | 1 | 2 | 3 | 4 | 5 |
| Deterministic | $63.18_{\pm3.06}$ | $54.23_{\pm3.67}$ | $48.47_{\pm4.27}$ | $41.84_{\pm4.59}$ | $31.96_{\pm3.96}$ |
| Ensemble-W | $67.10_{\pm3.19}$ | $57.92_{\pm3.82}$ | $51.83_{\pm4.50}$ | $45.16_{\pm4.91}$ | $34.93_{\pm4.27}$ |
| FFG-W | $57.49_{\pm3.17}$ | $47.62_{\pm3.64}$ | $41.99_{\pm4.15}$ | $35.61_{\pm4.24}$ | $26.59_{\pm3.66}$ |
| FFG-U | $61.71_{\pm3.35}$ | $52.61_{\pm3.84}$ | $47.08_{\pm4.36}$ | $40.56_{\pm4.54}$ | $30.88_{\pm3.93}$ |
| Ensemble-U | $61.87_{\pm3.36}$ | $52.69_{\pm3.97}$ | $46.96_{\pm4.52}$ | $40.59_{\pm4.72}$ | $30.85_{\pm3.98}$ |

Table 14: Corrupted CIFAR-100 ECE ($\downarrow$) values (in %).

| Method | Skew Intensity | | | | |
|--------|------|------|------|------|------|
| | 1 | 2 | 3 | 4 | 5 |
| Deterministic | $30.06_{\pm2.70}$ | $37.50_{\pm3.17}$ | $42.48_{\pm3.66}$ | $48.41_{\pm4.06}$ | $57.19_{\pm3.60}$ |
| Ensemble-W | $12.31_{\pm2.04}$ | $17.03_{\pm2.31}$ | $20.36_{\pm2.61}$ | $24.16_{\pm2.98}$ | $29.72_{\pm2.49}$ |
| FFG-W | $14.28_{\pm0.78}$ | $11.07_{\pm1.11}$ | $11.13_{\pm0.85}$ | $11.21_{\pm1.29}$ | $11.96_{\pm1.68}$ |
| FFG-U | $4.64_{\pm1.66}$ | $7.80_{\pm2.03}$ | $10.42_{\pm2.39}$ | $13.98_{\pm2.83}$ | $19.17_{\pm2.72}$ |
| Ensemble-U | $5.84_{\pm1.91}$ | $10.28_{\pm2.43}$ | $13.60_{\pm2.96}$ | $17.54_{\pm3.42}$ | $23.56_{\pm3.16}$ |

Table 15: Corrupted CIFAR-10 accuracy (↑) values (in %) for pruning FFG-W and FFG-U.

| Method | Skew Intensity | | | | |
| --- | --- | --- | --- | --- | --- |
| | 1 | 2 | 3 | 4 | 5 |
| FFG-W (100%) | $83.80_{\pm 2.43}$ | $76.22_{\pm 3.10}$ | $69.30_{\pm 4.11}$ | $61.82_{\pm 4.66}$ | $50.72_{\pm 4.68}$ |
| FFG-W (50%) | $83.39_{\pm 2.66}$ | $75.58_{\pm 3.23}$ | $68.43_{\pm 4.21}$ | $60.69_{\pm 4.75}$ | $49.75_{\pm 4.73}$ |
| FFG-W (10%) | $84.04_{\pm 2.61}$ | $76.21_{\pm 3.24}$ | $69.35_{\pm 4.18}$ | $61.65_{\pm 4.72}$ | $50.51_{\pm 4.74}$ |
| FFG-W (1%) | $84.16_{\pm 2.31}$ | $76.48_{\pm 3.02}$ | $69.72_{\pm 3.92}$ | $62.72_{\pm 4.47}$ | $51.26_{\pm 4.58}$ |
| FFG-W (0.1%) | $46.33_{\pm 1.30}$ | $41.92_{\pm 1.44}$ | $38.91_{\pm 1.59}$ | $36.03_{\pm 1.76}$ | $32.43_{\pm 1.88}$ |
| FFG-U (100%) | $86.90_{\pm 2.47}$ | $80.33_{\pm 3.14}$ | $74.34_{\pm 4.06}$ | $67.23_{\pm 4.75}$ | $57.00_{\pm 4.83}$ |
| FFG-U (75%) | $86.99_{\pm 2.37}$ | $80.64_{\pm 2.95}$ | $74.99_{\pm 3.81}$ | $67.87_{\pm 4.53}$ | $57.33_{\pm 4.65}$ |
| FFG-U (50%) | $86.93_{\pm 2.36}$ | $80.70_{\pm 2.93}$ | $75.10_{\pm 3.76}$ | $68.14_{\pm 4.44}$ | $57.66_{\pm 4.49}$ |
| FFG-U (25%) | $85.93_{\pm 2.39}$ | $79.41_{\pm 2.96}$ | $73.48_{\pm 3.82}$ | $66.52_{\pm 4.52}$ | $55.57_{\pm 4.59}$ |

Table 16: Corrupted CIFAR-10 ECE (↓) values (in %) for pruning FFG-W and FFG-U.

| Method | Skew Intensity | | | | |
| --- | --- | --- | --- | --- | --- |
| | 1 | 2 | 3 | 4 | 5 |
| FFG-W (100%) | $13.05_{\pm 0.64}$ | $12.14_{\pm 0.89}$ | $11.56_{\pm 0.93}$ | $10.77_{\pm 1.05}$ | $10.86_{\pm 1.31}$ |
| FFG-W (50%) | $14.27_{\pm 0.59}$ | $13.19_{\pm 0.88}$ | $12.36_{\pm 0.94}$ | $11.59_{\pm 1.07}$ | $11.43_{\pm 1.33}$ |
| FFG-W (10%) | $12.50_{\pm 0.52}$ | $11.66_{\pm 0.76}$ | $11.33_{\pm 0.90}$ | $10.86_{\pm 1.04}$ | $11.28_{\pm 1.38}$ |
| FFG-W (1%) | $9.86_{\pm 0.49}$ | $9.17_{\pm 0.62}$ | $8.85_{\pm 0.87}$ | $8.87_{\pm 0.93}$ | $11.10_{\pm 1.45}$ |
| FFG-W (0.1%) | $10.08_{\pm 0.87}$ | $7.82_{\pm 0.96}$ | $6.74_{\pm 0.81}$ | $7.34_{\pm 0.79}$ | $8.77_{\pm 0.93}$ |
| FFG-U (100%) | $2.47_{\pm 1.04}$ | $4.93_{\pm 1.66}$ | $7.77_{\pm 2.44}$ | $11.27_{\pm 2.81}$ | $16.16_{\pm 2.92}$ |
| FFG-U (75%) | $2.79_{\pm 0.60}$ | $3.92_{\pm 0.92}$ | $5.27_{\pm 1.59}$ | $7.63_{\pm 2.00}$ | $11.80_{\pm 2.39}$ |
| FFG-U (50%) | $3.11_{\pm 0.59}$ | $4.26_{\pm 0.92}$ | $5.44_{\pm 1.58}$ | $7.58_{\pm 1.94}$ | $11.38_{\pm 2.26}$ |
| FFG-U (25%) | $5.27_{\pm 0.35}$ | $5.33_{\pm 0.57}$ | $6.20_{\pm 1.19}$ | $7.95_{\pm 1.55}$ | $11.19_{\pm 2.21}$ |

Table 17: Corrupted CIFAR-100 accuracy (↑) values (in %) for pruning FFG-W and FFG-U.

| Method | Skew Intensity | | | | |
| --- | --- | --- | --- | --- | --- |
| | 1 | 2 | 3 | 4 | 5 |
| FFG-W (100%) | $57.49_{\pm 3.17}$ | $47.62_{\pm 3.64}$ | $41.99_{\pm 4.15}$ | $35.61_{\pm 4.24}$ | $26.59_{\pm 3.66}$ |
| FFG-W (50%) | $57.16_{\pm 3.16}$ | $47.33_{\pm 3.65}$ | $41.43_{\pm 4.18}$ | $35.02_{\pm 4.25}$ | $25.89_{\pm 3.58}$ |
| FFG-W (10%) | $58.77_{\pm 3.18}$ | $48.61_{\pm 3.69}$ | $42.89_{\pm 4.25}$ | $36.33_{\pm 4.35}$ | $26.97_{\pm 3.70}$ |
| FFG-W (1%) | $50.64_{\pm 2.89}$ | $41.70_{\pm 3.35}$ | $37.35_{\pm 3.69}$ | $31.56_{\pm 3.65}$ | $23.84_{\pm 3.05}$ |
| FFG-W (0.1%) | $6.72_{\pm 0.23}$ | $6.00_{\pm 0.24}$ | $5.77_{\pm 0.28}$ | $5.43_{\pm 0.33}$ | $4.82_{\pm 0.33}$ |
| FFG-U (100%) | $61.71_{\pm 3.35}$ | $52.61_{\pm 3.84}$ | $47.08_{\pm 4.36}$ | $40.56_{\pm 4.54}$ | $30.88_{\pm 3.93}$ |
| FFG-U (75%) | $61.47_{\pm 3.41}$ | $52.25_{\pm 3.95}$ | $46.84_{\pm 4.46}$ | $40.37_{\pm 4.65}$ | $30.48_{\pm 3.97}$ |
| FFG-U (50%) | $60.76_{\pm 3.46}$ | $51.68_{\pm 3.96}$ | $46.28_{\pm 4.43}$ | $39.84_{\pm 4.61}$ | $29.85_{\pm 3.92}$ |
| FFG-U (25%) | $58.11_{\pm 3.20}$ | $49.06_{\pm 3.64}$ | $43.62_{\pm 4.09}$ | $37.20_{\pm 4.23}$ | $27.93_{\pm 3.60}$ |

Table 18: Corrupted CIFAR-100 ECE (↓) values (in %) for pruning FFG-W and FFG-U.

| Method | Skew Intensity | | | | |
| --- | --- | --- | --- | --- | --- |
| | 1 | 2 | 3 | 4 | 5 |
| FFG-W (100%) | $14.28_{\pm 0.78}$ | $11.07_{\pm 1.11}$ | $11.13_{\pm 0.85}$ | $11.21_{\pm 1.29}$ | $11.96_{\pm 1.68}$ |
| FFG-W (50%) | $15.42_{\pm 0.84}$ | $12.04_{\pm 1.19}$ | $11.85_{\pm 0.91}$ | $11.77_{\pm 1.31}$ | $11.71_{\pm 1.61}$ |
| FFG-W (10%) | $10.85_{\pm 0.84}$ | $9.08_{\pm 0.95}$ | $9.96_{\pm 1.10}$ | $10.97_{\pm 1.74}$ | $13.35_{\pm 1.98}$ |
| FFG-W (1%) | $14.86_{\pm 1.10}$ | $11.99_{\pm 1.13}$ | $11.78_{\pm 1.20}$ | $11.48_{\pm 1.47}$ | $11.37_{\pm 1.64}$ |
| FFG-W (0.1%) | $3.21_{\pm 0.57}$ | $4.47_{\pm 0.55}$ | $5.28_{\pm 0.70}$ | $6.53_{\pm 0.96}$ | $7.79_{\pm 1.01}$ |
| FFG-U (100%) | $4.64_{\pm 1.66}$ | $7.80_{\pm 2.03}$ | $10.42_{\pm 2.39}$ | $13.98_{\pm 2.83}$ | $19.17_{\pm 2.72}$ |
| FFG-U (75%) | $4.78_{\pm 1.58}$ | $7.29_{\pm 1.91}$ | $9.59_{\pm 2.33}$ | $13.05_{\pm 2.91}$ | $18.39_{\pm 2.87}$ |
| FFG-U (50%) | $5.31_{\pm 1.55}$ | $7.10_{\pm 1.81}$ | $9.06_{\pm 2.18}$ | $12.35_{\pm 2.82}$ | $17.43_{\pm 2.83}$ |
| FFG-U (25%) | $9.74_{\pm 0.99}$ | $8.17_{\pm 1.01}$ | $8.67_{\pm 1.21}$ | $10.53_{\pm 1.85}$ | $13.22_{\pm 2.15}$ |

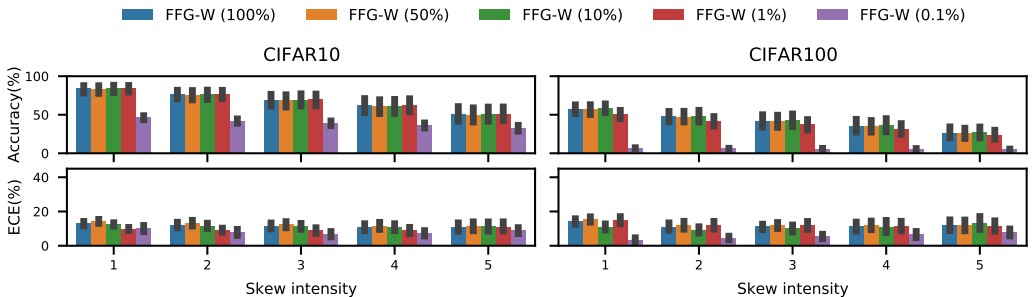

Figure 8: Accuracy (↑) and ECE (↓) on corrupted CIFAR for pruning FFG-W. We show the mean and two standard errors for each metric on the 19 perturbations provided in (Hendrycks & Dietterich, 2019).

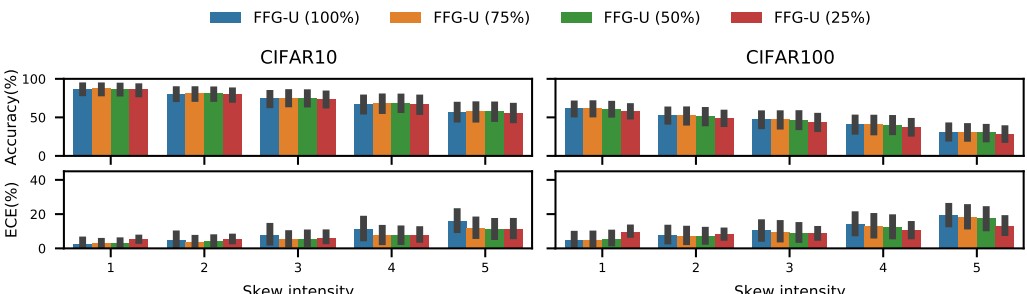

Figure 9: Accuracy (↑) and ECE (↓) on corrupted CIFAR for pruning FFG-U. We show the mean and two standard errors for each metric on the 19 perturbations provided in (Hendrycks & Dietterich, 2019).