# OpenReview forum: "Sparse Uncertainty Representation in Deep Learning with Inducing Weights"
_NeurIPS.cc/2021/Conference — NeurIPS 2021 Poster_

### Official Review · Reviewer_6QzC · 2021-07-10

**Rating:** 6
**Confidence:** 3

**Summary:**

This paper proposes a method for reducing the storage cost of BNNs. The idea is to apply the inducing point method and fast Matheron's sampling rule, commonly used in GPs, to standard variational BNNs. Experiments show that the resulting BNNs stay competitive to standard BNNs while enjoying up to 75% reduction in parameter size.

**Limitations And Societal Impact:**

I might have missed it, but I could not find any explicit discussion regarding the limitations of the method. This is especially crucial due to the first point in my "Cons" above.

**Main Review:**

**Pros:**

This paper is generally well written, though not without blemishes (see below). The method is solid and can be seen as a "natural" counterpart of the standard inducing weight method in function space (GPs) for weight space (BNNs).


**Cons:**

* While the proposed method significantly reduces the storage cost of BNNs, it introduces a large overhead during prediction. E.g. the proposed BNNs are around twice as slow as the vanilla counterparts for the commonly-used number of MC samples (e.g. 10, see Tab. 4). This makes me wonder whether the trade-off is worth it at all, considering ResNet-50 (the architecture that the authors use) for ImageNet only requires just under 100MB of storage space. In contrast, prediction latency has a real-world, tangible effect in any real-time predictive system, e.g. in self-driving cars. I hope to see a discussion from the authors during the rebuttal period.

* The paper can be hard to read at times. I think this is because of how dense the presentation of the paper is. Given how dense this paper is already, it is thus not good to see that the authors still tried to squeeze every little bit of space by using e.g. text style symbols (e.g. \textstyle\sum) in display equations. I think it would be much better presentation-wise if the authors move some least important sections to the appendix, and then be more judicious with whitespace.


**Minor comments/suggestions:**

* Please *do not* just copy-and-paste bibtex entries from Google Scholar. Many references are either inaccurate or inconsistent. For instance, Hendrycks & Dietterich, Kingma & Ba are in ICLR; "A tutorial on sparse gaussian processes ..." -> "A tutorial on sparse **G**aussian processes"; etc.
* L.36, L.41: please be consistent with the em dashes.
* Starting from L.57: The usage of the boldface in math should be consistent. The matrices $\mathbf{X}, \mathbf{Y}$ are in bold, but $W_l$ among other quantities are not.
* L.60: Hinton and Van Camp, 1993 should be cited.
* L.65: Missing iterates over $i, j$ in the definition of $\phi$
* L.78: "This approach returns ..." -> "This approach yields ..."
* L.78-79: Break the sentence up
* L.81: "... so can share parameters" what does it mean exactly?
* L.87-88: Too dense with the inline math, please use display math
* L.94-95: Inline math hard to read. Consider adding some words in between the two $\widetilde{p}(\mathbf{v})$'s
* L.134-135: More efficient compared to what complexity?
* Fig. 1: Use $\left( ... \right)$, esp. for the middle term consisting of green and white vertical bars
* L.175: "... the non-linearity ..." -> "... a non-linearity ..."
* L.184: What does "d" on top of \sim mean?
* Fig. 3: Need Ensemble-W result


---

**Post-rebuttal**

I increased my score. Nevertheless, I still think that the presentation of the paper needs to be improved:

* The application of parameter-efficient-but-slower BNNs should be discussed more explicitly in the introduction. This will help motivate the method more. Also, it is instructive to show the parameter efficiency of the method on larger networks (that take more than 100MB of storage).
* The exposition of the method should be improved more---there should be a balance between information content and density.

**Time Spent Reviewing:**

3

---

> ### Author Response · Authors · 2021-08-08
> **Response**
>
> Thank you for your review. We appreciate your comment on the paper as generally well-written.
>
> Currently our method is mainly relevant for edge computing (as also observed by R**8V4M**). There has been significant research effort invested in reducing parameter counts of deterministic networks via pruning methods. So clearly the community does consider the 100MB of storage for ResNet-50 problematic for certain applications, making the storage cost of such deterministic networks an important threshold to clear. Memory efficient Bayesian methods have been of interest recently ([rank-1 BNNs \(Dusenberry et al., ICML 2020\)](https://arxiv.org/abs/2005.07186), [k-tied Normal distribution \(Swiatkowski et al., ICML 2020\)](https://arxiv.org/abs/2002.02655)), so our paper is timely and relevant. Given that none of them has gone below the parameter count of a deterministic network, we present a unique advancement in that direction.
>
> Our method is not a free lunch and indeed comes with an overhead in computing time. However, we believe that we have described the advantages and drawbacks openly and the reviewer guidelines state that _“authors should be rewarded rather than punished for being up front about the limitations of their work”_. R**8V4M** commends this aspect of our work: _“The paper is careful and honest about evaluating the strengths (smaller number of parameters) and weakness (larger computational cost)”_. Further speeding up methods such as ours is a topic for future research, and we hope that our work can serve as inspiration and a promising direction. Yet, there are edge computing applications that are less time-sensitive but require good uncertainty estimates and/or protection of data privacy, e.g. AI assistants in healthcare-related mobile apps. We believe that our substantial leap in parameter efficiency is a significant contribution and fills a niche that has not been addressed by existing methods.
>
> We hope this clarifies the significance of our work and that you would like to see it at NeurIPS. Should you have any concerns or questions remaining, we are of course happy to address them.
>
> Thank you also for your more detailed comments and suggestions; such feedback is highly valuable and we will incorporate these into our next revision. In particular we will make use of the additional page for the camera-ready version to make the manuscript less dense. Knowing at which points specifically the paper is hard to read is particularly helpful.
>
> Regarding the questions from your minor comments:
>
> > L81 “... so can share parameters" what does it mean exactly?
>
> Tying the distributional parameters to the same values (e.g. mean and covariance of prior and posterior conditional Gaussian).
>
> > L.134-135: More efficient compared to what complexity?
>
> Compared to the full weight matrix, which would be $O(d_{out} d_{in})$ for a factorised Gaussian.
>
> > L.184: What does "d" on top of \sim mean?
>
> “Distributed as”. We will remove the "d" for consistency.

---

> > ### Comment · Reviewer_6QzC · 2021-08-18
> > **Thank you for your response**
> >
> > Thank you for your response, I increased my score. Nevertheless, I still think that the presentation of the paper needs to be improved:
> >
> > * The application of parameter-efficient-but-slower BNNs should be discussed more explicitly in the introduction. This will help motivate the method more. Also, it is instructive to show the parameter efficiency of the method on larger networks (that take more than 100MB of storage).
> >
> > * The exposition of the method should be improved more---there should be a balance between information content and density. Speaking about an additional page for the camera-ready (in case of acceptance), I'm not sure if there would be any (there is no mention of this in the guideline). But in any case, please try to make it less dense anyway.

---

### Official Review · Reviewer_pv3L · 2021-07-13

**Rating:** 4
**Confidence:** 4

**Summary:**

This paper proposes a layer-wise variational inference scheme for Bayesian neural networks that uses "inducing weights" for each layer before up-sampling to the full (layer-wise) weight distribution. The scheme is equivalent to a hierarchical prior formulation with an analogous hierarchical variational posterior. Experiments are performed on the standard BDL benchmarks of synthetic one dimensional regression, ResNets on CIFAR-100, and out of distribution detection.

**Ethical Concerns:**

n/a.

**Limitations And Societal Impact:**

maybe suggest why smaller neural networks are better or worse for society?

**Main Review:**

tldr: This method has a lot of promise, but the methodological writing is close to impenetrable currently, so I tend to reject it. Another round of revision to clean up the experiments and focus the writing presentation would help this paper have impact.

## Originality

*Are the tasks or methods new?* Yes, they seem to be a new version of VI for BNNs.

*Is the work a novel combination of well-known techniques?* Yes, I mostly followed along the Matheron's rule derivation, which is a nifty idea to reduce the computational complexity of prior sampling over the conditional distribution from the weights and the inducing weights.

*Is it clear how this work differs from previous contributions?* Yes, I think so. It bears a bit more resemblance to the GP style hierarchical priors and posteriors of [Karaletsos & Bui, NeurIPS, '20](https://arxiv.org/abs/2002.04033) than is mentioned in the paper. Variational matrix normal posteriors are also pretty closely related and probably deserve a bit of discussion as well (even though you do compare to them, which is helpful) [Louizos & Welling, ICML, '16](https://arxiv.org/pdf/1603.04733.pdf).

*Is related work adequately cited?*
- See above comment plus also the probabilistic meta-hypernetwork approach of [Karaletsos et al, '18](https://arxiv.org/abs/1810.00555)
- Hypernetworks and their Bayesian counterparts are also relevant as hypernetworks can reduce the parameter count of standard networks significantly, while Bayesian hyper-networks should contain some of the same advantages of this work (better uncertainty quantification, etc...) [Ha et al, '16](https://arxiv.org/abs/1609.09106) and [Krueger et al, '17](https://arxiv.org/abs/1710.04759).

## Quality

*Is the submission technically sound?* Mostly so, a few comments:
- line 148: from my understanding the "mixture of delta posterior" is not theoretically justified due to support mismatch in KL space, which isn't an issue at all if you're not going for the bayesian interpretation (just all it ensembling), but could prove problematic if you're trying to train through that KL.
    - Thus, is it okay to drop the KL term when training?

- line 631 (but throughout): My understanding for sampling this joint would be to use Kronecker algebra to have the cholesky factors of the covariance (both defined under eq 5) and then to compute the MVM $(L_r \otimes L_c)z$ where $z$ is a standard gaussian random vector (before reshaping into a matrix). But, what you're taking is instead a non-square matrix vector product and so it's really tough to parse if that's the correct set of algebra.
    - Also, from the Kronecker approach, I think you can just write down the algebra and simplify to get out $\bar W$ and $\bar U$ (although I haven't had the chance to do so), which would simplify the presentation a good deal.
    - Eq 31 (and following): I don't follow the shapes of these matrices any more because of how the matrices are being written down, as well as how these matrix matrix products are being written down.

- line 638: "we can reduce variance ..." this statement concerns me, are you actually doing the full Matheron's rule here? or some sort of variance reduction method? How does this play into the variance based proofs in the following lines?

- line 640: why do we need to take the choleskies of these matrices? my understanding is that given any $AA' = \Sigma$ we can use $A'z$ to produce valid samples.

- eq 36: it's a bit of a misnomer to call this a GP because its just a linear kernel (and this could be easily derived by marginalizing linear regression).

- 3.2 and 3.3: which version is actually implemented?
    - Do the variables $U_c$ play a role in the Matheron's rule based approach?
    - If not, are they in the implementation anywhere or are they just an object of theoretical interest due to their connection to the deep GP literature?
    - line 305: the scale of the KL term is probably important, and suggests that there's probably a missing constant or something there. After all, the scale determines if you're doing proper Bayesian inference or a generalized version (which isn't practically important, just you should be clear).

*Are claims well supported (e.g., by theoretical analysis or experimental results)?*
- The experimental work demonstrates that the hierarchical VI scheme does outperform SVI, but it definitely doesn't seem to outperform either ensembling approach or really a deterministic NN, so the claims of "significant improvement" are not really well supported.

*Are the methods used appropriate?*
- I'm not a fan of the comparisons and think that some other reduced space parameterization technique really ought to be compared to (even if the memory overheads are somewhat higher).
    - From what I can tell, MIMO networks [Havasi et al, ICLR, '20](https://arxiv.org/pdf/2010.06610.pdf), rank-1 BNNs [Dusenberry et a l, ICML, '20](https://arxiv.org/abs/2005.07186), and subspace inference [Izmailov et al, '19](https://arxiv.org/abs/1907.07504) are probably also relevant comparisons. At least one of these would be very helpful (probably MIMO or rank-1 BNNs b/c I think that they are using the same architectures that you are).
    - This is especially needed because FFG-W (aka SVI) is such a weak baseline and then your approach is often dominated by your own baseline of Ensemble-U.
        - Even the claim of "not including these baselines [because they increase the parameter count]" (lines 223-225) is insincere because FFG-W doubles the parameter count as well.
    - A comparison to the standard Bayesian approaches for BDL is also missing [VOGN, SGMCMC, SWAG, GG-MC, etc.], but may be less relevant if the authors wish to focus solely on the parameter counting aspects. But these should still probably be cited.

- Similarly, a comparison to a post facto pruning method (down to 25% of the weights) would also be helpful. Potentially, something like [Savarese et al, '20](https://arxiv.org/pdf/1912.04427v4.pdf) would be a good comparison or simple magnitude based pruning.
    - Pruning represents a similar reduction in model capacity that the proposed approach does as well.
    - Fig. 7 is a nice attempt but a deterministic comparison should be performed.
    - One could always just imagine doing post facto pruning, and then a cheap and cheerful bayesian approach on retraining in the sparse space, say SWAG ([Maddox et al, NeurIPS, '19](https://arxiv.org/abs/1902.02476)).

- An alternative comparison (and probably more relevant) would probably be the sparsity based SGMCMC procedure of [Deng et al, NeurIPS, '19](https://proceedings.neurips.cc/paper/2019/file/2d71b2ae158c7c5912cc0bbde2bb9d95-Paper.pdf), which is both designed for producing sparsity in deep networks and for producing strong uncertainty calibration.

*Please note that I'm not suggesting the authors compare to all of these, please just pick one or two in the rebuttal (but cite the possibility of the rest). Preferably the SGMCMC one and/or deterministic pruning.*

*Is this a complete piece of work or work in progress?* Complete work.

*Are the authors careful and honest about evaluating both the strengths and weaknesses of their work?* No weaknesses are discussed from my reading. The obvious one is what happens if $U$ is too small of a vector / matrix, in which the model will have too little capacity from the approximation algorithm. This might be seen in Fig. 4 right but I'm not sure.

## Significance

*Are the results important?* Possibly.

*Are others (researchers or practitioners) likely to use the ideas or build on them?* Probably, this is a popular field right now.

*Does the submission address a difficult task in a better way than previous work?* While the approach beats SVI, that's a very low bar. Both ensemble based approaches (one of which is your own baseline) seem to outperform it.

*Does it advance the state of the art in a demonstrable way?* Empirically maybe not, in terms of practical sampling mechanisms for matrix normals, probably so.

*Does it provide unique data, unique conclusions about existing data, or a unique theoretical or experimental approach?* A unique theoretical approach, same experiments as many papers in this space.

## Discussion

- The uncertainty bands in Fig. 3 are pretty nice overall.

- Fig 4 looks like a mixed bag, the accuracy of FFG-U (which I take to be your method) is reasonably worse in terms of accuracy than even a standard deterministic NN (although better calibrated).

- My overarching takeaway from the experiments is that the performance across all experiments goes *deep ensembles > ensembles-U > FFG-U > FFG-W* which suggests that FFG-U only outperforms FFG-W, but the hierarchical prior structure has some sort of performance limitations (as standard deep ensembles tend outperform ensembles over $U$).
    - This limits the practical usefulness of the results to be only that the approach reduces the parameter count, but then further comparisons to pruning would be helpful.
    - This ends up being hidden, as a result, but looking at the uncertainty quantification results in Tables 10+, it seems like generically the deterministic NN outperforms FFG-U and ensemble-U in terms of accuracy, but loses in terms of ECE, thus continuing to weaken the empirical evidence.
        - This type of result really shouldn't be hidden deep in the appendix.

- At its core, one of the weaknesses of SVI for NNs is that it tends to significantly underfit the data (see for example [Blier & Ollivier, NeurIPS, '18](https://papers.nips.cc/paper/2018/file/3b712de48137572f3849aabd5666a4e3-Paper.pdf)), hence, why the variance is clipped practically (lines 240-246), yet, your approach still seems like it'll underfit (given empirically that you still have to clamp the variances and the improved ECE, which is often an indicator of the amount of posterior variance).
    - Should your approach get around this under-fitting issue or is its success solely due to the increased amount of structure?
    - What happens if you don't clamp the variance in your approach?


### Questions

- More details on the *Ensemble-U* baseline would be very helpful. Is this just an ensemble of five models with the extra regularizer of $p(\text{vec}(W) | \text{vec}(U))$ as the prior?
    - If that's not quite what it is, then that should be the baseline.
    - Also, see the comment on line 148 above.

- What does the reduction in parameter count mean practically? Is there actually a memory reduction or is this negated by the requirement of forming the full layerwise matrices $W$ via sampling using $U$?
    - Does it induce some sort of sparsity in weight space or something along those lines?

- What is the say "accuracy vs model size" tradeoff required on CIFAR-10 / 100? It's hard to parse from the two small points in Fig. 4.

- Why does ECE skyrocket in Fig 7 as more pruning is performed?
    - This seems to be the case in Fig 5 bottom right for small $M$.

- What is training time like? In Fig. 4 you mention prediction time? It seems like it'll be a few times slower due to the extended Matheron's rule comments.

- Why should magnitude based pruning not change the probabilistic model in FFG-U?

- Overall, what happens if you ignore the layer-wise approximation and do the reduction on the full space (e.g. $\text{vec}(W)$ refers to the vector of all parameters of the NN)? Does the approach still work?

### Writing Comments

- Fig 4 left is too small and detailed, had to zoom into 400% to get it to be visible, as well as to be able to parse taht there's actually two separate architectures in there. I'd suggest displaying only one architecture and the two inference types to make this clearer.
    - Can this be shown on a log scale as well? Is there ever a "break-even point" where with enough samples, the Matheron's rule approach is as fast, so to speak?

- line 657: mean + noise isn't helpful.

- Personally, I'd suggest replacing most of the tables with plots using the x axis as the skew intensity level. The tables are really tough to parse as nothing is bolded or anything.

- throughout: define "push-forward distribution" is this just the transformed (matrix) normal distribution after some sort of linear transformation?

**Time Spent Reviewing:**

8

---

> ### Author Response · Authors · 2021-08-08
> **Initial high-level response**
>
> Thank you very much for your detailed review. We appreciate that this has been a significant time investment and are grateful for the many constructive suggestions, which we will incorporate into our next revision.
>
> Your largely positive comments about the novelty and technical soundness of the method, as well as the detail of your questions, suggest that you understand the paper rather well. Therefore we are surprised that you describe the paper as _“close to impenetrable”_ and are disappointed about the negative score. Note that the other reviewers also understand the overall approach and describe the achievement of memory efficiency as significant. We hope that we will be able to alleviate your concerns and that you will reconsider your overall assessment.
>
> We will answer your questions in two replies. In this first reply, we will focus our initial response around what we perceive as your core concerns: the clarity of the detailed derivations in the appendix and to a lesser degree the choice of baselines. We will address your detailed questions in a separate response before the end of the initial author feedback period.
>
> **Concerns regarding clarity**
>
> > line 631 (but throughout): My understanding for sampling this joint would be to use Kronecker algebra to have the cholesky factors of the covariance (...) to compute the MVM $(L_r \otimes L_c)z$ where $z$ is a standard gaussian random vector (...) you're taking instead a non-square matrix vector product and so it's really tough to parse if that's the correct set of algebra.
>
> - We use [standard equalities for Kronecker products](https://en.wikipedia.org/wiki/Kronecker_product#Matrix_equations) where possible to rewrite Kronecker matrix-vector products as two matrix-matrix products, which is [commonly used for expressing the sampling procedure for Matrix normal distributions](https://en.wikipedia.org/wiki/Matrix_normal_distribution#Drawing_values_from_the_distribution). Appendix B also provides an introduction for Matrix normal distributions and related conditioning rules and we are happy to further add the equalities we just linked. Does this clarify the derivations for you or did we misinterpret the source of confusion? We opted for this convention as in our opinion it significantly reduces visual clutter by removing the need for any $\textrm{vec}(\cdot)$ expressions and $\otimes$ symbols, instead denoting everything as matrix-matrix products (which also results in the equations matching the code more closely). We are of course aware that different readers may prefer different conventions.
>
> > I think you can just write down the algebra and simplify to get out $\bar{W}$ and $\bar{U}$
>
> - As far as we are aware that is not possible as the covariance of the joint of $\bar{W}, \bar{U}$ has Khatri-Rao product structure (i.e. blocks of Kronecker products), but is not Kronecker factored itself and does not have an efficient square root. See e.g. slides 8ff [here](https://people.cam.cornell.edu/~dme65/data/scan_talk.pdf) for work from other researchers.
>
> > Eq 31 (and following): I don't follow the shapes of these matrices any more
>
> - Matrix shapes other than the ones before Eq. 31 are defined in the main text around Eq. 5 (end of page 3, beginning of page 4). As suggested by R**tCFn**, we will add a reference table for the notation and will also include all matrix shapes there to improve the clarity.
>
> **Additional baselines**
>
> - As stated in the experimental section we aim _“to demonstrate competitive performance to popular W-space uncertainty estimation methods while using significantly fewer parameters”_. The baselines are chosen accordingly to demonstrate such improvement. The additional baselines you suggest, including rank-1 BNNs, would fill a similar role to deep ensembles in providing reference values for the ideal performance when more parameters than a deterministic network can be used. This is supported by [existing comparisons on CIFAR10 with a Wide-Resnet-28-10](https://github.com/google/uncertainty-baselines/tree/main/baselines/cifar) (35M parameters for deterministic net, deterministic ResNet-50 has 25M parameters) which shows many of your suggested methods perform similarly or slightly worse than deep ensembles.
>
> - You suggested pruning as another baseline. We indeed have this experiment in appendix H, with results summarised in the last paragraph of section 4.3 and Fig. 7. Our claim of the inducing weight methods achieving <24.3% memory of a deterministic net is **before pruning**, and after pruning the savings are even more significant. We will explain the details for this experiment in the next reply.
>
> - Our impression is that you have evaluated our experiments with an expectation of improving performance over SOTA while simultaneously achieving parameter efficiency. This is overall unrealistic: for network compression in general (beyond uncertainty estimation), the developed memory efficient networks typically do NOT outperform their full-size counterparts. Therefore we believe our experimental results are significant when evaluated from the viewpoint of parameter-efficient networks.
>
> - We can add the performance of additional baselines as further reference values to the camera-ready paper. However, under the space constraints for an already dense and technical paper, and for the reasons explained above, at submission time we did not find that they would add enough insight to justify the space.
>
> **Further comments on our chosen baselines & experiments**
>
> - We believe FFG-W is an important baseline to have. It is the natural equivalent in weight space to our FFG-U method with similar hyperparameter choices, and a comparison between FFG-W and FFG-U serves as an “ablation” for the usage of the inducing weights. It is also the sole baseline in recently published work on the [k-tied Normal distribution \(Swiatkowski et al., ICML 2020\)](https://arxiv.org/abs/2002.02655), so we do not share the sentiment that it is too weak or not relevant at all.
>
> - We also note that comparisons should focus on FFG-W vs FFG-U and Ensemble-W vs Ensemble-U, rather than FFG-U vs Ensemble-U. We include both FFG-VI and ensemble methods to demonstrate the wide applicability of the inducing weight approach. Indeed one can follow [variational MCMC approaches in sparse GP \(Hensman et al., 2015\)](https://arxiv.org/abs/1506.04000) and apply MCMC in U space; in that case it should be compared with weight-space MCMC approaches, e.g. [Cyclical SG-MCMC](https://arxiv.org/abs/1902.03932) or the SGMCMC-SA method (Deng et al., NeurIPS 2019) that you suggest. We leave this to future work.
>
> **Please see our next reply titled "Second response on the detailed questions" to address your other questions in detail.**

---

> ### Author Response · Authors · 2021-08-09
> **Second response on the detailed questions**
>
> In the second reply to your review, we organise your questions into different categories and answer them therein. Let us know if you need further clarifications on our answers.
>
>
> **Related work**:
>
> 1a. Choices of references:
>
> - We are aware that there is a large body of reference in BDL (e.g. VOGN, SGMCMC, SWAG, GG-MC, etc. as you mentioned). We choose to focus on discussing related work relevant to parameter efficiency.
> - R **8V4M** acknowledged that “The paper adequately cites and describes differences with related work”.
> - We still thank you for pointing out additional references, and we will add an extended related work section to the appendix with further in-depth discussion.
>
> 1b. Existing “parameter efficient approaches” are not even close to ours in terms of efficiency:
>
> - We presented in Table 1 the parameter count per layer for k-tied FFG-W and rank-1 BNNs -- they need parameter counts higher than Deterministic.
> - SG-MCMC and/or SWAG requires tracking the sampling/gradient descent trajectory and take intermediate results. This means it requires $K$ times the parameter count of Deterministic, if $K$ networks are ensembled.
> - Subspace inference considers the space spanned by the SGD trajectory and uses the first $K$ PCA components of this space. Therefore it requires $K$ times the parameter count of Deterministic.
> - MIMO is orthogonal to our work and all other works mentioned above. Even though it can process $K$ repeated inputs at the same time, it still requires the same parameter count as Deterministic.
>
> 1c. Why pruning FFG-W and your suggested approaches changes the underlying BNN model, and why pruning $Z$ parameters for FFG-U does not:
>
> - In our pruning baseline for FFG-W, the post-pruning network is fine-tuned with an ELBO different from the ELBO in the initial training stage. This is because of the change in the KL term: in the initial stage the ELBO contains $KL[q(W)||p(W)]$, but after pruning, the fine-tuning stage uses $KL[q(W_{pruned}) || p(W_{pruned})]$. This means if $W_{ij}$ is pruned out, then the corresponding prior is set to $p(W_{ij}) = \delta(W_{ij} = 0)$. Therefore it clearly changes the model (due to change in the prior).
> - Similarly for the suggested “prune first following e.g. Savarese et al. (2020), then run BDL” approaches, after pruning, (approximate) posterior inference will be performed only on the remaining weights. Again this means the resulting probabilistic model is different.
> - The “sparse SGMCMC” approach by Deng et al. (2019) uses a sparse prior, again meaning that a different probabilistic model is in use. Once more, the pruning step changes the prior in a similar way as explained in point (a) and (b).
> - On the contrary, pruning $Z$ parameters for FFG-U does not change the underlying model. This is because, the “pseudo prior” is always consistent with the actual prior, regardless of the $Z$ values: $p(W) = \int p(W|U) p(U) dU$. In the extreme case, setting all entries of $Z$ to 0, which results in the highest parameter efficiency (although not useful in performance), still maintains the W-space prior to stay the same.
>
>
> **Clarification on implementations**:
>
> 2a. Our implementation follows the weight-space perspective (section 3.2). The function-space perspective (section 3.3) is an equivalent theoretical view and it shows the connection to sparse GP more explicitly.
>
> 2b. The extended Matheron’s rule:
> - We are using the full Matheron’s rule, which requires drawing samples from the joint distribution. But we also reparameterise this sampling process to draw the Gaussian noise in a lower dimensional space akin to the [local reparameterisation trick \(Kingma et al., NeurIPS 2015\)](https://arxiv.org/abs/1506.02557). This trick can reduce variance because the number of Gaussian noise variables being sampled is significantly reduced.
> - We compute the cholesky decompositions to enable sampling fewer numbers of Gaussian noise variables. In general, if $AA^\top = \Sigma$ then one can use $Az, z \sim N(0, I)$ (instead of $A^\top z$) to produce valid Gaussian samples. But when $A$ has way more columns than rows, using $Az$ for sampling means $z$ will be a very long vector. Instead, using the Cholesky decomposition $\Sigma = BB^\top$ and getting samples with $Bz$ means the length of $z$ equals the number of rows in $A$. Note that even when sampling $K$ samples with the extended Matheron’s rule, we only pay the $O(M_{in}^3 + M_{out}^3)$ cost for Cholesky decomposition once. Therefore this additional cost is still acceptable when compared with matrix product costs $O(KM_{in}d_{in}d_{out})$, especially when $M << d$.
> - The $U_c$ and $U_r$ matrices are never instantiated or computed in practice. They are introduced to help formulate the expressions/equations.
>
> 2c. The “scaling” of the KL term (are you referring to line 705 instead of 305?):
> - We are aware of the change of W-space to F-space which brings in changes in variable dimensionality, and this explains the “scaling change” in the conditional KL term.
> - We note that the “scaling change to account for dimensionality change” in our derivation is different from the “scaling change from tempering”, i.e. if we write ELBO = data-LL + $\beta$ * KL with $\beta$ as a tempering hyperparameter, then we still use $\beta = 1$, regardless of derivations in W-space or F-space.
>
> 2d. Details of Ensemble-U implementation:
> - Ensemble-U with $K=5$ is an ensemble of five models in $U$ space with the extra regularizer of $KL[q(vec(W) | vec(U)) || p(vec(W) | vec(U))]$. In other words, it has 5 $U$ matrices but a shared single set of $Z$ matrices. As typically $M_{in} << d_{in}$ and $M_{out} << d_{out}$, the parameter efficiency is comparable to FFG-U.
> - The $KL[q(U) || p(U)]$ is dropped because it is not defined (q(U) is a mixture of delta measures while p(U) is Gaussian). We do not claim Ensemble-U to be Bayesian; rather Ensemble-U should be compared with Ensemble-W (also non-Bayesian) as for improvement in parameter efficiency.
>
>
> **More details on experiments**:
>
> 3a. We consider FFG-U and Ensemble-U as our proposed approaches, and others as baselines. We do not claim better accuracy/calibration when compared with Ensemble-W. The only place where we use the phrase “significant improvement” is in the conclusion referring to the parameter count.
>
> 3b. On your comment of "not including these baselines [because they increase the parameter count]" (lines 223-225)" being "insincere":
> - We are honest with our parameter efficiency claim and we disagree with the “insincere” comment. All our quantitative references in the paper are w.r.t. Deterministic net, e.g. in the abstract we clearly stated “**reducing the parameter size to ≤ 24.3% of that of a single neural network**”. If we were to go for a much more eye-catching statement then we could state our approach to achieve >20x compression as compared with Ensemble-W or use the post-pruning results for even bigger improvements.
> - Ensemble-W and FFG-W are included to provide reference in terms of accuracy/calibration, not for parameter efficiency.
>
> 3c. Comparisons to deterministic pruning:
> - We have pruning experiments in appendix H, with a summary in section 4.3. The comparisons are done not only for accuracy, but also for calibration and OOD robustness tasks as well which are not available in the literature. Therefore we don’t think it is fair to just call our effort “a nice attempt”.
> - We have explained in point (1c) of the “related work” part why your suggested approaches change the underlying probabilistic model while pruning FFG-U does not. In this regard, for a proper comparison of (approximate) Bayesian inference methods, one should make sure the probabilistic models are the same. In other words, we can also apply the inducing weight approach “post-hoc” after pruning, which can potentially improve parameter efficiency further.
> - We are happy to further compare pruned ensembles in the camera-ready paper, but as this presents a somewhat intricate baseline with two independent dimensions for pruning (ensemble size and percentage of weights pruned in each network). We felt that the explanation for the setup and additional discussion of the results would not fit into the space constraints at submission time.
>
> 3d. On your comments of "overarching takeaway from the experiments is that the performance across all experiments goes deep ensembles > ensembles-U > FFG-U > FFG-W":
> - As stated many times, our focus is NOT to improve accuracy/calibration performance and outperform SOTA (deep ensembles). Instead we focus on achieving parameter efficiency without big loss of performance. In this regard our experimental result is significant.
>
> 3e. On your comments of "hiding Deterministic results in appendix":
> - We strongly disagree that this result is “hidden deep in the appendix”. With Determinstic performing by far the worst in terms of ECE in the data corruption experiments, we found that it rescaled the y-axis of Fig. 6 enough to make the figure too difficult to read and therefore deferred the results to the appendix - where we have of course included them for completeness.
> - We are happy to highlight this particular aspect of the experiments more explicitly in the additional content page for the camera-ready paper as part of the discussion in the experiment section. But overall, we have run extensive empirical comparisons, and it is impossible to discuss all aspects of the results at length in the main text under the significant space constraints of a primarily technical conference paper.
>
> 3f. On your question about "worse ECE in Fig 7 and Fig 5 when network is severely pruned or having small $M$":
> - This is because using very small $M$ or pruning $Z$ too much make the number of parameters too small to be sufficient for performing well. The result is aligned with many network pruning results: if the pruning rates are too high, the resulting performance can be less satisfactory.

---

> > ### Author Response · Authors · 2021-08-09
> > **(Continued) Second response on the detailed questions**
> >
> > **Other questions**:
> >
> > 4a. On your comments of “No weakness is discussed”:
> > - We strongly disagree: we explicitly show the increased runtime cost in Fig. 4 and the limits in reducing the parameter count of our method in the ablation study (Fig. 5) and the pruning experiment (Fig. 7).
> > - All other reviewers have commented on this, in particular R**8V4M**, who describes our paper as “careful and honest about evaluating the strengths (...) and weaknesses”.
> > - The reviewer guidelines state that “In general, authors should be rewarded rather than punished for being up front about the limitations of their work” and we purposefully attempted to comply with this requirement.
> >
> > 4b. On your question "What happens if you ignore the layer-wise approximation and do the reduction on the full space (e.g. vec(W) refers to the vector of all parameters of the NN)?"
> > - Our method is designed for matrices, so applying it to a vector containing all weight parameters would not work (unless treating the vector as a Dx1 matrix). One possibility would be to concatenate the weight matrices along a dimension if they have the same size along the other. Alternatively, one could reshape the full parameter vector into some arbitrary matrix and apply our method, although this would break the row/column structure of the weight matrices.
> > - We also described a hierarchical approach in appendix F to introduce correlations between layers.
> >
> > 4c. On your question "What does the reduction in parameter count mean practically?"
> > - See response to R**8V4M**. In short: we instantiate $W$ in our code for simplicity, but memory efficient implementations are of course possible. Note that many existing works in network compression also instantiate full $W$ (by e.g. filling in the zeros) to allow efficient usage of GPU computing. In general, translating parameter efficiency to memory efficiency requires further design on implementations which should be device- and hardware-specific.
> >
> > 4d. On other hyper-priors on BNN weights:
> > - We are aware of other prior designs for BNNs such as the hyper-network approaches and the GP style hierarchical priors and posteriors of Karaletsos & Bui, NeurIPS, '20. For the latter approach, there are a few key differences besides the obvious one of their GP mapping from latent to weight space. Most importantly, they share their latents between subsequent layers, making it unclear how to generalize the approach to CNNs. We also suspect that their method is difficult to scale to larger architectures as they do not present any results beyond MNIST-like problems. We are happy to discuss this further and add an extended related work section in the appendix.
> >
> > 4e. On under-fitting issues of VI for BNNs:
> > - This is a general issue for VI and our approach does not fundamentally solve this problem. We should note that our approach is not designed to target this problem anyway -- indeed the inducing weight approach also extends to ensembles and other inference approaches.
> >
> > 4f. On your question "What is training time like?":
> > - See response to R**8V4M**. In short: all the methods in comparison, including the inducing weight methods, finish training in hours on a single GPU, rather than days or weeks. Therefore we would consider the overhead to be acceptable. We will add precise figures in revision.

---

> > ### Comment · Reviewer_pv3L · 2021-08-26
> > **Response**
> >
> > Thank you for the clarifications overall. I'm still unconvinced by the rebuttal and think that there are a couple of outstanding issues with the paper.
> >
> > First, the presentation is close to impenetrably dense in my reading, although thank you for the clarity comments (1c,2b). I really hope that the authors do end up putting a table with matrix sizes and continue to keep the introduction to matrix normal distributions around. Subjectively, I find the kronecker operators much cleaner to deal with but I think this is a personal preference. Similarly, I also strongly suggest dropping either the weight or function space parameterization to give yourself more space in the methods section.
> >
> > Second, I strongly object to not comparing across sparse approaches because "they change the probabilistic model." In general, there's no reason a priori to expect that a NN with a Gaussian prior should produce sparse solutions, and indeed, we probably don't even think that a Gaussian prior is particularly reflective of our beliefs about the problem so there's minimal reason to be tied to it. Just choose the prior that helps produce the sparsest solutions a priori (e.g. horseshoe).
> >
> > Similarly, any different type of sparse pruning approach would necessarily also change the likelihood and thus the probabilistic model. Thus, I think it's a pretty large weakness of the paper not to compare to stronger sparse pruning approaches, particularly ones that are also similarly probabilistic in nature (e.g. Deng et al, '19).
> >
> > Third (and deeply connected to the second point), there's not a lot (if any) of improvement over even ~random~ _EDIT_ _magnitude_ pruning (+retraining) as evidenced by Table 6 (or a deterministic NN as in Tables 10+), so there seems to be a bit of a lack of connection to practice. After all, we don't really need to be Bayesian for the sake of being Bayesian.
> >
> > ### Detailed Comments
> >
> > 1a,b: Thanks for the clarifications. Overall, I think the sparsity framing of this work needs to be emphasized more.
> >
> > 2a,b: Thanks for the clarifications -- not being able to follow the matrix shapes was why I asked these questions.
> >
> > 2c: Yes, I'm still confused here. It's not obvious to me why the weight and function spaces would then produces the same objective (and ultimately the same approximation) under different scalings. And furthermore, what the difference in objective in Eq 40 would do.
> >
> > 2d: Thanks for the clarifications, I agree that you're going for the non-probabilistic approach so it's fine to drop the KL.
> >
> > 3a: Thanks, on a first set of passes through the paper, this is hard to get at. I'd suggest making it more clear.
> >
> > 3b: Fair enough. Again, you probably want to compare to sparsity inducing approaches.
> >
> > 3c,e: Again, thanks for running this comparison against a basic pruning approach. I find it particularly interesting that random pruning generally seems considerably better calibrated than your approach at higher and higher levels of sparsity. This to me suggests that the comparison with other pruning based approaches (e.g. magnitude based) would go somewhat poorly. _EDIT: This comparison is done to some extent in Table 6, and is suggestive that my point is correct. It's magnitude based pruning, rather than random pruning, and there seems to be a lack of improvement over magnitude based pruning. Please make sure that this is marked clearly in the figure caption._
> >
> > 3d,e: To me, the practical goal of this paper is to demonstrate a Bayesian approach that simultaneously prunes and then maintains high performance. You seem to achieve reasonable-ish ECE after pruning (although apparently somewhat worse than standard pruning) and worse accuracy than a standard neural net (how we interpret Tables 10+). Thus, why should a practitioner prefer this Bayesian approach? Why wouldn't they either use a standard NN or a pruned standard NN?
> >
> > 3f: Thanks for the clarification.

---

> > > ### Author Response · Authors · 2021-08-26
> > > **Reply to your response (thank you for responding)**
> > >
> > > Thank you for your response, good to hear that most of the questions are cleared. On your remaining questions:
> > >
> > > Q1: on comparisons to other pruning approaches
> > > - We believe you agree with us on our technical statement about the other pruning approaches, that "they change the probabilistic model". We highlight this statement to show that this is another layer of comparison: in probabilistic deep learning, performance differences can come from differences in model design and/or differences in inference approaches.
> > > - If we were to compare with e.g. Deng et al. (2019), then this is comparing multiple differences at the same time: (a) Gaussian prior vs. sparsity prior; (b) pruning weights (thus changing the network architecture) vs pruning $Z$ (no change in network arch); and (c) SG-MCMC inference vs. inducing weight inference. Too many varying factors here to demonstrate a clear picture.
> > > - In our experiments we decided to focus on comparing inference approaches (thus presenting Ensemble-U vs Ensemble-W, and FFG-U vs FFG-W). We believe it is better to present a focused study that really demonstrate the key innovation. (This does not mean we are hiding the drawbacks -- we are upfront about the increased time complexity.)
> > > - We will add pruning ensemble in camera ready, for which we think it has less different aspects so it is closer to an "apples to apples" comparison. Still we believe our current experiments presented the most direct comparisons.
> > >
> > > Q2: on your comments "the practical goal is to demonstrate a Bayesian approach that simultaneously prunes and then maintains high performance", "why should a practitioner prefer this Bayesian approach, why not use a (pruned) standard NN"
> > > - We emphasise that our primary contribution is the inducing weight formulation. So perhaps we would like to modify your statement of our goal as "maintaining performance while being parameter efficient" -- pruning is not the only approach to achieve parameter efficiency.
> > > - Indeed our approach is orthogonal to pruning, our pruning experiments show that they can be applied together. We hope you can also take this aspect in consideration.
> > > - For the purpose of maintaining uncertainty estimate quality after network compression, we believe one needs to start from a full network with uncertainty estimation ability (e.g. BNN or deep ensemble). Starting from a deterministic NN means the pruning/compression procedure later on do not consider uncertainty.
> > > - In the above example, although it is possible to fine-tune the pruned model with Bayesian approach, this is no longer a "maintenance" of uncertainty (because there's no original uncertainty estimate to maintain). Therefore the uncertainty of many other possible function fits - which are represented by different parameter settings of the original full NN - will be lost.
> > >
> > > Q3: confusions on point (2c)
> > > - The derivation comes from the function-space view presented in appendix E.
> > > - We include this view to give an interpretation of the auxiliary variables/parameters. Indeed the function-space view shows the connection between the augmented $U$ variables to inducing output variables in sparse GP, as well as the connection between $Z$ parameters in our framework to the inducing input locations in sparse GP. The visualisation in Figure 2 crucially relies on this view.
> > > - We thank you for your comments on improving clarity; in camera-ready we will consider better presentation of the interpretation of the inducing weights.

---

> > > > ### Author Response · Authors · 2021-08-26
> > > > **Additional detailed comments to your response**
> > > >
> > > > Additional response to your further points:
> > > >
> > > > * We are unsure what you mean by “random pruning”. We are not removing weights randomly from the model in any of the experiments. For FFG-W, we drop weights based on the KL divergence of the posterior to the prior as proposed in [\(Graves, NeurIPS 2011\)](https://papers.nips.cc/paper/2011/hash/7eb3c8be3d411e8ebfab08eba5f49632-Abstract.html), which boils down to the signal-to-noise ratio making it the probabilistic equivalent of magnitude-based pruning.
> > > >
> > > > * We strongly disagree with your interpretation of the pruning results. At a pruning level where accuracy on the original test data is roughly maintained (which we would consider the practically most relevant setting), i.e. FFG-U (50%) and FFG-W (10%), FFG-U is more accurate on both CIFAR10 and CIFAR100, while the ECE of FFG-W is lower on CIFAR10, but higher on CIFAR100. Further, pruned FFG-U outperforms pruned FFG-W across almost all dataset pairs and metrics for arbitrary pruning levels. Therefore we believe that FFG-U
> > > > clearly offers a practical advantage over FFG-W. The latter is only preferable at extreme pruning levels, at which performance significantly deteriorates.
> > > >
> > > > * The technical content of our work already stretches the format of a 9-10 pages conference paper. The novelty and significance of these contributions has been recognised by all reviewers, including yourself.
> > > >
> > > > We’d kindly ask you to re-consider your assessment, and we are willing to discuss more if you have further questions.

---

> > > > > ### Comment · Reviewer_pv3L · 2021-08-31
> > > > > **Clarification**
> > > > >
> > > > > Sorry for the mis-understanding about "random pruning", I mis-interpreted the pruning procedure in lines 802-804. I see now that it's magnitude based pruning after re-training. The point I'm making still stands though.

---

> ### Author Response · Authors · 2021-08-25
> **We'd appreciate any further feedback & clarifications**
>
> Thanks again for your efforts in reviewing our paper.
>
> We would appreciate it if you can let us know whether our response has addressed your concerns. We are more than willing to clarify more if you have further questions.
>
> Best,
> Authors

---

### Official Review · Reviewer_8V4M · 2021-07-16

**Rating:** 7
**Confidence:** 3

**Summary:**

The paper proposes a parameter-efficient method for uncertanity quantification in deep neural network using per layer inducing weights. The paper develops a network weight sampling scheme based on inducing weights. The proposed method employs Matheron's rule for efficient sampling. The method requires fewer parameters than even deterministic networks, while keeping good calibration and performance, at the cost of longer compute times, when few MC weight samples are used.


**Main Review:**

Originality

The paper proposes a novel idea (best to my knowledge) of using inducing points known from sparse variational GPs in the context of Bayesian neural networks.
The paper adequately cites and describes differences with related work.

Quality

Submission is technically sound and claims are well supported through detailed derivations and exhaustive experimentation.
Methods used are appropriate and the work looks complete.
The paper is careful and honest about evaluating the strengths (smaller number of parameters) and weakness (larger computational cost) of the proposed method as well as comparison to the baseline alternatives (weight pruning).

It would be interesting to see the training times additionally to the predictions times. As far as I understand from appendix G.2, during training the method uses 1 MC sample, and this might be the regime where the method is much less effective (>10x slower).

I am not sure whether the full weight matrix W gets materiallized during training of the FFG-U method.
If this is the case, does it diminish all the gains in terms of memory requirement during training / testing?

The paper is very math heavy and while most of the calculations looks good to me, I might have missed some errors.

Clarity

The submission is clearly written and well organised. It adequately informs the reader.
The paper refers to complex mathematical objects and procedures (e..g Matheron's sampling, Matrix Normal closed form derivations), but aims to inform the reader through references and even explaining the basics of these concepts.

Significance

The paper proposes an interesting method for reducing the number of network parameters by a large factor.
However, it increases the computational cost during testing and very likely also during training.
Therefore, I expect that this method might find applicability in areas which require low memory and calibrated predictions, but the prediction time is not as important.
Nevertheless, from the theoretical standpoint, the paper develops an interesting connection between variational inference BNNs and sparse variational GPs.


**Time Spent Reviewing:**

11

---

> ### Author Response · Authors · 2021-08-08
> **Response**
>
> Thank you for your positive review and feedback, it is encouraging to read that you found the paper clear and well-organised. We are glad that you appreciate the theoretical contribution of our work as well as the care that went into the discussion of the limitations and the baseline comparisons.
>
> 1. On your comment about training time overhead:
> - we reported overhead in test time, our conclusion is that the overhead becomes less significant when using more MC samples. This indeed broadly carries over to training time, so training FFG-U with 1 MC sample is indeed slower than training a deterministic network. Note that training FFG-W with 1 MC sample is also twice as slow as deterministic networks due to local reparameterisation. All methods presented in our paper can be trained in a matter of hours on a single GPU (and not multiple days or weeks), so we would consider the overhead to be acceptable. We will add precise figures to the camera-ready revision.
>
> 2. For the implementation of the weight matrices $W$:
> - we instantiate the full weight matrices in our implementation to simplify the code base. This is common (research) practice e.g. many works on pruning methods do not take advantage of sparse linear algebra kernels. Memory efficient implementations of our method are of course possible by folding the projection from inducing space into the forward pass or e.g. only instantiating one column of the weight matrix at a time (which is ok for hardware with limited parallelisation capabilities). Ultimately we would expect a memory efficient implementation to be device- and hardware-specific.

---

### Official Review · Reviewer_tCFn · 2021-07-21

**Rating:** 7
**Confidence:** 3

**Summary:**

The paper proposes to represent uncertainty in neural networks by augmenting neural network layers with inducing weights and then assuming that the resulting vector of weights is drawn from a matrix normal distribution which is structured so that it has fewer parameters than the original network. The parameters of the distribution are learned via variational inference and the authors present a procedure to sample efficiently from this distribution for optimizing the model parameters and for inference at test time. Experiments on classification and out of distribution detection are presented to validate the approach.

**Limitations And Societal Impact:**

I have provided suggestions on addressing the limitations in the review above. I believe there is no potential negative societal impact of this work.

**Main Review:**

**Quality and Clarity**

The overall idea is clear but quite heavy on notation and hence may require multiple readings to process entirely. I would recommend adding a table of symbols either in the body of the paper or in the appendix to enable readers to keep track of the notation. I would also suggest modifying Figure 1 to illustrate how the method is memory efficient since that is the key selling point of this work.

**Originality and Significance**

While this is one of many approaches to representing uncertainty in deep learning, the ability to effectively represent uncertainty without increasing, and indeed sometimes decreasing, storage cost is certainly significant. My only concern is that the method appears quite sensitive to hyperparameter choices, especially from the discusiion in Section 4.2 (Lines 260-268) and associated plots and without any guidance on how to choose the hyperparameters, it might be difficult to translate the success of the method to datasets other than the ones considered in the paper.

**Strengths**

1. A new memory efficent approach and associated sampling procedure is proposed for representing uncertainties in deep learning.
2. Experiments on a range of task validate the efficacy of the approach for both prediction and out-of-distribution detection.

**Weaknesses**

1. The writing is a bit heavy on notation and the details may be hard to parse in one reading.
2. The approach appears quite sensitive to hyperparameters and it is not clear how to set the hyperparameters for new datasets.

**Queries and Suggestions**

1. Is it possible to give some guidance on choosing the hyperparameters $\sigma_{\max}$, $\lambda_{\max}$, $M$ in general or is an ablation study the only way to figure it out? If it is the latter then I would recommend highlighting that fact a bit more clearly in the paper and providing any insights you may have on reasonable ranges over which to search, based on your experience with the experiments. Exploring the efficacy of AutoML in finding the optimal hyperparameters might also be an interesting option.

2. Why does pruning FFG-W to 2% of the size of the deterministic network give the lowest ECE in Figure 7? That appears to be violating the observed trend quite significantly and is probably worth investigating or at least providing some, even speculative, thoughts on.

3. I would suggest including some comparison with other space efficient approaches for uncertainty estimation like rank-1 BNNs. Even if they are not as space efficient as this approach it would be instructive to see if the extra space they require gives any performance improvement on the accuracy/ECE metrics and if not then those approaches can be convincingly ruled out in favor of this one.

4. I also did not notice any wall-clock time numbers for the different methods. It would be beneficial to have them since it does appear that the space savings come at the cost of slightly higher time complexity. While that is not a big issue, it would be good to see how much longer the method takes on average to get a better sense of the tradeoff.



**Time Spent Reviewing:**

3

---

> ### Author Response · Authors · 2021-08-08
> **Response**
>
> Thank you for your encouraging review and your suggestions on clarity improvements. We will address your questions and concerns regarding the hyperparameter sensitivity below.
>
> > 1. Guidance on/sensitivity to hyperparameters?
>
> We disagree that our method is particularly sensitive to hyperparameters. All ML methods have hyperparameters; a careless choice would break the method (e.g. consider choosing learning rate $10^{-2}$ for the Adam optimiser which often does not work), and the best setting varies based on the particular task in consideration. However, what matters is to provide a common recipe for the hyperparameters that work well across a range of experiments. In our case we performed an ablation study on ResNet-18 as a comparably small architecture, and we found the selected hyperparameter setting from this ablation study to transfer well to our ResNet-50 experiments. We will clarify this further in revision.
>
> [\(Louizos & Welling, ICML 2017\)](https://arxiv.org/abs/1703.01961) uses a similar hyperparameter to control $q$ variance, but sets it to a fixed value which tends to work well in each experiment and is in a similar range to ours -- a slight restriction of the variational variance appears to be a consistent theme for good performance. More generally, variational inference for NNs is prone to underfitting, and the hyperparameters are chosen to alleviate this problem as much as possible. This issue has recently been receiving significant attention through the lens of cold posteriors [\(Wenzel et al., ICML 2020\)](https://arxiv.org/abs/2002.02405). Hence more general theoretically-grounded approaches for setting such hyperparameters require further research.
>
> > 2. Why is ECE the lowest for the most pruned FFG-W?
>
> Note that accuracy goes down as well. In our experience, underfitting models (e.g. unconstrained FFG-W) are often well-calibrated, we would hypothesise that this is due to being less confident as a result of not fitting the training data perfectly. We can add this as a speculative explanation, but had not found a reference to substantiate this claim and had therefore left it out. We believe this phenomenon would be worth studying further.
>
> > 3. Comparison to rank-1 BNN?
>
> The suggested approaches, including rank-1 BNNs, are fundamentally limited to a higher parameter count than deterministic nets. Given that they overall perform similarly to deep ensembles (see [existing comparisons on CIFAR10 with a Wide-Resnet-28-10](https://github.com/google/uncertainty-baselines/tree/main/baselines/cifar), an architecture on the same model-size scale as ResNet-50), we considered the deep ensemble sufficient as SOTA baseline for accuracy/calibration performance (see also the overall response).
>
> > 4. Time comparison
>
> See Fig. 4 for a test time comparison. The overall conclusions broadly carry over to training (with  the exception of FFG-W being about twice as slow as deterministic networks there due to local reparameterisation). We will add a table to the camera-ready paper.

---

> > ### Comment · Reviewer_tCFn · 2021-09-01
> > **Re**
> >
> > Thank you for your clarifications. I do not have any other questions and I will keep my score of Accept for this paper.

---

### Author Response · Authors · 2021-08-08
**Summary**

We thank all reviewers for their extensive and constructive feedback.

We would first like to summarise the **strengths** of the paper as per the reviewers:

**Novelty of the approach**
- Our approach is based on inference in auxiliary variable space, which is novel and orthogonal to existing memory efficient uncertainty estimation approaches based on low-rank q(W) distributions or pruning. Our approach is deeply motivated by sparse GPs and we make this connection explicit in the paper. We expect our approach to inspire future developments which translate useful techniques from the GP literature to uncertainty estimation in deep learning.

- All the reviewers agree the approach is novel: R**tCFn** _“a new memory efficient approach”_ ; R**8V4M** _“the paper proposes a novel idea”_ ; R**pv3L** _“they seem to be a new version of VI for BNNs”_ ; R**6QzC** _“the method is solid and can be seen as a 'natural' counterpart of the standard inducing weight method in function space (GPs) for weight space (BNNs)”_.

**Significance of the contribution**
- Our approach is the first uncertainty estimation method in BNN/deep ensemble literature to **achieve <25% memory of a deterministic network even before pruning**. Prior to our work, the most memory efficient approaches such as [rank-1 BNNs \(Dusenberry et al., ICML 2020\)](https://arxiv.org/abs/2005.07186) still increase the number of parameters compared to a deterministic network.

- Reviewers agree that this contribution is significant: R**tCFn** _“the ability to effectively represent uncertainty without increasing, and indeed sometimes decreasing, storage cost is certainly significant”_; R**8V4M** _“Submission is technically sound and claims are well supported through detailed derivations and exhaustive experimentation”_; R**6QzC** _“Experiments show that the resulting BNNs stay competitive to standard BNNs while enjoying up to 75% reduction in parameter size”_.

We address the two main **concerns** among the reviewers; we are keen to discuss these further if additional clarifications are needed.

**Technically dense presentation**
- The reviewers described the manuscript as generally _“well-written”_ (R**6QzC**), _“overall clear”_ (Rs **tCFn**, **8V4M**) and _“well-organised”_ (R**8V4M**). However, we are aware that our submission is a rather technical paper that will benefit from further editing based on the reviewers’ comments (e.g. typesetting, additional discussions). We also believe that R**pv3L**, who was the most critical, had a good understanding of our paper based on the depth and level of detail in their questions.

- The overall feedback assures us that at large we communicate the core ideas effectively, and R**8V4M** summarises this succinctly: _“The paper refers to complex mathematical objects ... but aims to inform the reader through references and even explaining the basics of these concepts”_. As there will be an additional page available for the camera-ready version, we are optimistic that we will be able to make the final revision of our paper easier to read.

**Comparison to other baselines**
- R**tCFn** and **pv3L** suggest comparing to additional baselines, in particular to [rank-1 BNNs \(Dusenberry et al., ICML 2020\)](https://arxiv.org/abs/2005.07186). We are open to running these comparisons, but have opted not to at submission time in the interest of preserving space in an already dense paper. This is because **the suggested baselines are fundamentally limited to a higher parameter count than a deterministic network**, even though they perform quantitatively similar to deep ensembles that are compared in our paper. Instead, **the significance of our work lies in substantially reducing the parameter count** at a marginal loss in accuracy compared to the state-of-the-art.

- In detail, rank-1 BNNs [have been reported to perform comparably to deep ensembles](https://github.com/google/uncertainty-baselines/tree/main/baselines/cifar) on CIFAR10 with a Wide-Resnet-28-10 (35M parameters for the deterministic version). We would expect these results to transfer to the Resnet-50 (25M parameters for the deterministic version) architecture used in our paper. However, rank-1 BNNs are not even close to our methods in terms of parameter efficiency, so it would only serve as an additional quantification of accuracy/calibration and fill a similar role to deep ensembles in our comparisons.
- The most closely related work to ours on the [k-tied Normal distribution \(Swiatkowski et al., ICML 2020\)](https://arxiv.org/abs/2002.02655) compares only to mean-field variational inference, so we are confident that with our additional baselines in deep ensembles and pruned variational inference we are firmly in line with experimental protocols in the recent published literature on parameter-efficient uncertainty estimation.

---

### Decision · Program_Chairs · 2021-09-27

**Decision:**

Accept (Poster)

**Comment:**

This paper comes up with a variational inference strategy for learning a factored weight matrix, that they can then sample over efficiently using Matheron’s rule.  They need <25% of the parameters of the original deep ensemble but are competitive across metrics.  Four reviews were given for this paper with scores 6, 7, 4, 7.  Thus there was a general consensus to accept with one reviewer arguing for a reject.  In general the reviewers found the paper technically strong, novel and important in terms of the problem it addresses.

One of the major criticisms of the paper is that it is too dense and the reader is overloaded with technical background.  All reviewers seem to agree with this point (i.e. in discussion: “I wholly agree .. that the presentation of this paper is its major weakness”), but differ quite significantly in how important they think this is.  Some of the reviewers applaud the authors for striving for completeness, but seem to have found it a little excessive.  In particular, the reviewers didn't think presenting both the function space and weight space perspective was necessary.  Thus the authors are strongly recommended to move one of these to the appendix to make space for other parts of the paper.

There’s some criticism of the priors used - the reviewer felt like the empirical comparison wasn’t apples to apples because of the priors were different between their method and the baselines.  This seems reasonable, but the choice of prior seems core to the proposed method (needs to be decomposable due to Matheron’s rule).  Thus it would seem acceptable to not require the authors to change the prior of their method.

The final criticism is that the method is slower than the original ensemble, so sparsity doesn’t help - though there is still (in theory) considerable memory savings.  Again, the reviewers differed in their assessment of how important this issue is.

The majority felt that this is a strong paper. Considering that the major cited weakness is in the presentation of the technical background, the recommended decision is to accept the paper with the ask that the authors rework the technical background to make it more accessible.  In particular, the reviewers recommended that the function space perspective could be moved to the appendix in the camera ready.  This would free up space to make the paper less mathematically dense and allow more space for other parts of the paper.